# Relative terrestrial exposure ages inferred from meteoric $^{10}$Be and NO$_3^-$ concentrations in soils along the Shackleton Glacier, Antarctica

Melisa A. Diaz[1,2†], Lee B. Corbett[3], Paul R. Bierman[3], Byron J. Adams[4], Diana H. Wall[5], Ian D. Hogg[6,7], Noah Fierer[8], W. Berry Lyons[1,2]

[1]School of Earth Sciences, The Ohio State University, Columbus, OH, 43210, USA
[2]Byrd Polar and Climate Research Center, The Ohio State University, Columbus, OH, 43210, USA
[3]Department of Geology, University of Vermont, Burlington, VT, 05405, USA
[4]Department of Biology, Evolutionary Ecology Laboratories, and Monte L. Bean Museum, Brigham Young University, Provo, UT, 84602, USA
[5]Department of Biology and School of Global Environmental Sustainability, Colorado State University, Fort Collins, CO, 80523, USA
[6]Canadian High Arctic Research Station, Polar Knowledge Canada, Cambridge Bay, NU, X0B0C0, Canada
[7]School of Science, University of Waikato, Hamilton, 3216, New Zealand
[8]Department of Ecology and Evolutionary Biology and Cooperative Institute for Research in Environmental Science, University of Colorado Boulder, Boulder, CO, 80309, USA
[†]Now at Geology and Geophysics, and Applied Ocean and Physics and Engineering, Woods Hole Oceanographic Institution, Woods Hole, MA, 02543, USA

*Correspondence to:* Melisa A. Diaz (diaz.237@osu.edu)

**Abstract.** Modeling studies and field mapping show that increases in ice thickness during glacial periods were not uniform across Antarctica. Rather, outlet glaciers that flow through the Transantarctic Mountains (TAM) experienced the greatest changes in ice thickness. As a result, ice-free areas that are currently exposed may have been covered by ice at various points during the Cenozoic, complicating the understanding of ecological succession in TAM soils. We collected soil surface samples and depth profiles every 5 cm to refusal (up to 30 cm) from eleven ice-free areas along the Shackleton Glacier, a major outlet glacier of the East Antarctic Ice Sheet (EAIS). We measured meteoric $^{10}$Be and NO$_3^-$ concentrations to calculate measured (using $^{10}$Be inventory), estimated (using NO$_3^-$), and inferred (using surface $^{10}$Be concentration) surface exposure ages, both with and without assuming erosion. Exposure ages ranged from 58 ka to >6.5 Ma correcting for erosion and 57 ka to 1.9 Ma without erosion, with the youngest ages near the glacier terminus and at lower relatively elevations. We correlated NO$_3^-$ concentrations with meteoric $^{10}$Be concentrations to estimate exposure ages for all locations with NO$_3^-$ depth profiles but only surface $^{10}$Be data. Our results indicated that NO$_3^-$ concentrations can be used in conjunction with few meteoric $^{10}$Be data to rapidly and efficiently estimate relative surface exposure ages. In comparing NO$_3^-$ and $^{10}$Be depth profile measurements, we find that much of the southern portion of the Shackleton Glacier region has likely developed undisturbed under a hyper-arid climate regime.

**1. Introduction**

One of the most intriguing questions in biogeography concerns the relationship between the evolution of terrestrial organisms and landscape disturbance (e.g. glacial overriding), particularly in Antarctica. Current data indicate that organism lineages have survived in some Antarctic soils for possibly millions of years, despite multiple glaciations throughout the Pleistocene (Convey et al., 2008; Fraser et al., 2012; Stevens and Hogg, 2003). It is still unclear how and where these organisms found suitable glacial refugia given the high salt concentrations in high-elevation soils (Lyons et al., 2016). The most biodiverse soils in the Ross Sea sector are at low elevations near the coast, where the Ross Ice Shelf or sea ice meet the Transantarctic Mountains (TAM) (Collins et al., 2020). These soils are also those which are most susceptible to glacial overriding during glacial maxima, though the timing of retreat and glacial extent is still unknown on local scales (Golledge et al., 2012; MacKintosh et al., 2011).

Antarctica is believed to have maintained a persistent ice sheet since at least the Eocene epoch, and the East and West Antarctic Ice Sheets (EAIS and WAIS, respectively) have waxed and waned since at least the Miocene (Gasson et al., 2016; Gulick et al., 2017). Sediment core records collected from the Ross Sea and ice cores from the Antarctic interior indicate that the EAIS and WAIS have undergone dozens of glacial and interglacial cycles throughout the Cenozoic (Augustin et al., 2004; Talarico et al., 2012). The WAIS is a marine-terminating ice sheet with a grounding line below sea level, which decreases the stability of the ice sheet and results in rapid ice sheet advance and retreat during glacial periods compared to the EAIS (Pollard and DeConto, 2009). The EAIS is grounded above sea level and is generally more stable than the WAIS. The EAIS and WAIS were at their most recent greatest extent during the Last Glacial Maximum (LGM) (~22,000 yrs. ago) (Clark et al., 2009). During the LGM, the EAIS expanded along its margins and the greatest increases in height occurred at outlet glaciers, which flow through exposed peaks of the TAM and drain into the Ross and Weddell Seas (Anderson et al., 2002; Golledge et al., 2012; Mackintosh et al., 2014). As a result, many of the currently exposed TAM soils were overrun by ice during the LGM and some may have only recently been exposed.

Much of the Antarctic continent is a polar desert and geomorphological data from ice-free soils in the McMurdo Dry Valleys indicate that some regions have likely been hyper-arid for as long as 15 Ma (Marchant et al., 1996; Valletta et al., 2015). As such, atmospherically-derived constituents, including salts and metals, can accumulate in exposed Antarctic

soils at concentrations similar to those from the Atacama and Namib Deserts (Diaz et al., 2020; Lyons et al., 2016; Reich and
Bao, 2018). Using soil nitrate concentrations from the Meyer Desert in the Beardmore Glacier region and nitrate fluxes
calculated from a Dominion Range ice core, Lyons et al. (2016) estimated that at least 750,000 years have passed since the
Meyer Desert had wide-spread soil wetting. It is likely that other high elevation and inland locations in the TAM also have
high concentrations of salts and similarly old "wetting ages".

64         We calculated relative surface soil exposure ages of ice-free areas along the Shackleton Glacier, a major outlet

glacier of the EAIS. Outlet glaciers are among the most sensitive areas to glaciological change in Antarctica, and changes in
their extents over time are recorded in nearby sedimentary deposits (Golledge et al., 2013; Jones et al., 2015; Scherer et al.,
2016; Spector et al., 2017). The Shackleton Glacier flows between several exposed peaks of the Central Transantarctic
Mountains (CTAM) and ice-free areas are present at both low and high elevations. We report concentrations of meteoric
[10]Be and nitrate ($NO_3^-$) in soils from eleven distinct ice-free areas and use these data to estimate the exposure ages using
different assumptions. The sampling methodology was designed to capture soils which have low salt concentrations due to
recent exposure from glacial retreat following the LGM and soils that were exposed since at least the last glacial period.
These age data are among few surface exposure ages in the CTAM (Ackert and Kurz, 2004; Balter-Kennedy et al., 2020),
are the only age estimates of soils from the Shackleton Glacier region, inform how the EAIS responds to changes in climate,
and are crucial in understanding Antarctic terrestrial biogeography.
**2. Background**
**2.1. Cosmogenic nuclide exposure age dating and meteoric [10]Be systematics**

77         [10]Be is a cosmogenic radionuclide with a half-life of 1.39 Ma (Korschinek et al., 2010) that is produced both in the

atmosphere (meteoric) and *in-situ* in mineral grains. In the atmosphere, N and O gases are bombarded by high energy cosmic
radiation to produce meteoric [10]Be. Particle reactive [10]BeO or [10]Be(OH)$_2$ is produced and removed from the atmosphere by
wet and dry deposition (McHargue and Damon, 1991). At Earth's surface, meteoric [10]Be sorbs onto clay particles and it is
insoluble in most natural waters of pH greater than 4 (Brown et al., 1992; You et al., 1989). The clay particles can be
redistributed to lower depths in the soil profile due to particle migration or can be transported by winds. As such, the total
number of [10]Be atoms in a soil profile, its inventory, is a function of surface exposure duration, erosion, clay particle
translocation, solubility, and sedimentation. If delivery rates can be estimated, meteoric [10]Be can be used as a tool to
understand exposure age, erosion rates, and soil residence times (see Willenbring and Von Blanckenburg, 2009 and
references within).

87        The measurement meteoric [10]Be in soil has enabled researchers to date surfaces and features in Antarctica. Previous

studies have measured meteoric [10]Be in the McMurdo Dry Valleys (MDV) and Victoria Land soils and sediments to
calculate exposure ages and to determine the onset of the current polar desert regime (Dickinson et al., 2012a; Graham et al.,
2002; Schiller et al., 2009; Valletta et al., 2015). In general, these previous studies found that high elevation, northern fringe
regions along the Ross Embayment have been hyper-arid since at least the Pliocene. Meteoric [10]Be data have yet to be
published from the CTAM which represent ice sheet dynamics and climatic conditions closer to the Polar Plateau.
**2.2. Nitrate systematics in Antarctic soils**

94        The nitrogen cycle in Antarctica differs greatly from the nitrogen cycle in temperate regions, primarily due to scarce

biomass and vascular plants (Cary et al., 2010; Michalski et al., 2005). Nitrogen in CTAM soils primarily exists as nitrate
($NO_3^-$) and is primarily sourced from the atmosphere, with varying contributions from the troposphere and stratosphere (Diaz
et al., 2020; Lyons et al., 2016; Michalski et al., 2005). Similar to meteoric [10]Be, $NO_3^-$ is deposited on exposed soils, though
in contrast to [10]Be, nitrate salts are highly water-soluble. Once deposited on the surface, nitrate salts can be dissolved and
transported to lower elevations or eluted to depth when wetted (i.e. during ice/snow melt events). However, the hyper-arid
climate of the CTAM allows $NO_3^-$ to accumulate at high concentrations in soils (Claridge and Campbell, 1968a; Diaz et al.,
2020; Lyons et al., 2016). Soil $NO_3^-$ concentrations have the potential to inform wetting history and possibly glacial history
in the CTAM due to the relatively high solubility of nitrate salts, though uncertainties regarding heterogeneous deposition
and post-depositional alteration (such as re-volatilization and photolysis) require further investigation (Diaz et al., 2020; Frey
et al., 2009; Graham et al., 2002).
**2.3. Relative exposure age dating approach**

Here, we used meteoric $^{10}$Be and $NO_3^-$ concentrations to estimate CTAM relative exposure ages, acknowledging the
widespread use of *in-situ* exposure age dating which we later use for cross-validation. *In-situ* cosmogenic nuclides, such as
$^{10}$Be, $^{26}$Al, $^{21}$Ne, and $^{3}$He, have been measured to determine surface exposure ages at several locations across Antarctica,
particularly in the MDV and other exposed surfaces in Victoria Land (e.g. Balco et al., 2019; Brook et al., 1993, 1995; Bruno
et al., 1997; Ivy-Ochs et al., 1995; Strasky et al., 2009). There are considerably fewer studies from the CTAM (Ackert and
Kurz, 2004; Balter-Kennedy et al., 2020; Bromley et al., 2010; Kaplan et al., 2017; Spector et al., 2017), and previously
reported exposure ages of CTAM moraines and boulders from these studies ranged from <10 ka to >14 Ma. We seek to
utilize $NO_3^-$ and meteoric $^{10}$Be concentrations to attain a greater number of surface exposure ages and understand the
relationship between $NO_3^-$ and $^{10}$Be in the hyper-arid environment of the CTAM. Exposure ages are determined by three
approaches: "measured" by using measured meteoric $^{10}$Be concentrations in depth profiles, "estimated" by using $NO_3^-$
concentrations to estimate $^{10}$Be concentrations, and "inferred" by using the maximum/surface concentration of $^{10}$Be in the
soil profile to infer the total number of $^{10}$Be atoms in the profile (Graly et al., 2010). These approaches are described in
Sections 4.3 and 5.3.
**3. Study sites**
Shackleton Glacier (~84.5 to 86.4°S; ~130 km long and ~10 km wide) is a major outlet glacier of the EAIS that
drains north into the Ross Embayment with other CTAM outlet glaciers to form the Ross Ice Shelf (RIS) (Fig. 1). The ice
flows between exposed surfaces of the Queen Maud Mountains, which range from elevations of ~150 m near the RIS to
>3,500 m further inland. The basement geology of the Shackleton Glacier region is comprised of igneous and metamorphic
rocks that formed from intruded and metamorphosed sedimentary and volcanic strata during the Ross Orogeny (450-520 Ma)
(Elliot and Fanning, 2008). The southern portion of the region consists of the Devonian-Triassic Beacon Supergroup and the
Jurassic Ferrar Group, while the northern portions consists of Pre-Devonian granitoids and the Early to Mid-Cambrian
Taylor Group (Elliot and Fanning, 2008; Paulsen et al., 2004). These rocks serve as primary parent material for soil
formation (Claridge and Campbell, 1968b). Deposits of the Sirius Group, the center of the stable vs. dynamic EAIS debate,
have been previously identified in the southern portion of the Shackleton Glacier region, particularly at Roberts Massif (Fig.
2) and Bennett Platform, with a small exposure at Schroeder Hill (Hambrey et al., 2003).

The valleys and other ice-free areas within the region have been modified by the advance and retreat of the Shackleton Glacier, smaller tributary glaciers, and alpine glaciers. Similar to the Beardmore Glacier region, the Shackleton Glacier region is a polar desert, which results in the high accumulation of salts in soils. The surface is comprised primarily of till, weathered primary bedrock, and scree, which ranges in size from small boulders and cobbles to sand and silt. Clay minerals have been previously identified in all samples from Roberts Massif and are likely ubiquitous throughout the region (Claridge and Campbell, 1968b). However, the clays are a mixture of those derived from sedimentary rocks and contemporaneous weathering (Claridge and Campbell, 1968b). Thin, boulder belt moraines, characteristic of cold-based glaciers, were deposited over bedrock and tills at Roberts Massif, while large moraines were deposited at Bennett Platform (Fig. 2; Balter-Kennedy et al., 2020; Claridge and Campbell, 1968). Additional information on the sample locations and surface features is provided in Tables 1 and 2.

## 4. Methods

### 4.1. Sample collection

During the 2017-2018 austral summer, we visited eleven ice-free areas along the Shackleton Glacier: Roberts Massif, Schroeder Hill, Bennett Platform, Mt. Augustana, Mt. Heekin, Thanksgiving Valley, Taylor Nunatak, Mt. Franke, Mt. Wasko, Nilsen Peak, and Mt. Speed (Fig. 1). These areas represent soils from near the head of the glacier to near the glacier terminus at the coast of the RIS. Two surface samples (Table 1) were collected at each location (except for Nilsen Peak and Mt. Wasko, represented by only one sample) with a plastic scoop and stored in Whirl-Pak™ bags. One sample was collected furthest from the Shackleton Glacier or other tributary glaciers (within ~2,000 m) to represent soils that were likely exposed during the LGM and previous recent glacial periods. A second sample was collected closer to the glacier (between ~1,500 and 200 m from the first sample) to represent soils likely to have been exposed by more recent ice margin retreat.

Soil pits were dug by hand at the sampling locations furthest from the glacier for Roberts Massif, Schroeder Hill, Mt. Augustana, Bennett Platform, Mt. Heekin, Thanksgiving Valley, and Mt. Franke. Continuous samples were collected every 5 cm until refusal (up to 30 cm) and stored frozen in Whirl-Pak™ bags. All surface (21) and depth profile (25) samples were shipped frozen to The Ohio State University and kept frozen until analyzed.

## 4.2. Analytical methods

### 4.2.1. Meteoric $^{10}$Be analysis

A total of 30 sub-samples of surface soils from all locations and depth profiles from Roberts Massif, Bennett Platform, and Thanksgiving Valley were sieved to determine the grain size at each location. The percentages of gravel (>2 mm), sand (63 µm-2 mm), and silt (<63 µm) are reported in Table S1. Since there is a strong grain size dependence of meteoric $^{10}$Be (little $^{10}$Be is carried on coarse (>2 mm) grains (Pavich et al., 1986)) the gravel portion of the sample was not included in the meteoric $^{10}$Be analysis. The remaining soil (<2 mm) was ground to fine powder using a shatterbox.

Meteoric $^{10}$Be (Table 2) was extracted and purified at the NSF/UVM Community Cosmogenic Facility following procedures originally adapted and modified from Stone (1998). First, 0.5 g of powdered soil was weighed into platinum crucibles and 0.4 g of SPEX $^{9}$Be carrier (with a concentration of 1,000 µg mL$^{-1}$) was added to each sample. The samples were fluxed with a mixture of potassium hydrogen fluoride and sodium sulfate. Perchloric acid was then added to remove potassium by precipitation and later evaporated. Samples were dissolved in nitric acid and precipitated as beryllium hydroxide (Be(OH)$_2$) gel, then packed into stainless steel cathodes for accelerator mass spectroscopy isotopic analysis at the Purdue Rare Isotope Measurement (PRIME) Laboratory. Isotopic ratios were normalized to primary standard 07KNSTD with an assumed ratio of 2.85 x 10$^{-12}$ (Nishiizumi et al., 2007). We corrected sample ratios with a $^{10}$Be/$^{9}$Be blank ratio of 8.2 ± 1.9 x 10$^{-15}$, which is the average and standard deviation of two blanks processed alongside the samples. We subtracted the blank ratio from the sample ratios and propagated uncertainties in quadrature.

### 4.2.2. Nitrate analysis

Separate, un-sieved sub-samples of soil from all locations and depth profiles were leached at a 1:5 soil to water ratio for 24 hours, then filtered through a 0.4 µm Nucleopore membrane filter. The leachate was analyzed on a Skalar San++ Automated Wet Chemistry Analyzer with an SA 1050 Random Access Auto-sampler (Lyons et al., 2016; Welch et al., 2010). Concentrations are reported as NO$_3^-$ (Table S2) with accuracy, as determined using USGS 2015 standard, and precision better than 5% (Lyons et al., 2016).

### 4.3. Exposure age model

We developed a mass balance using the fluxes of meteoric [10]Be to and from Shackleton Glacier region soils to calculate the amount of time which has passed since the soil was exposed (Pavich et al., 1984, 1986). The model assumes that soils that were overlain by glacial ice in the past and are now exposed, accumulated less [10]Be than soils that were exposed throughout the glacial periods (Fig. 3). The concentration of meteoric [10]Be at the surface ($N$, atoms g$^{-1}$) per unit of time ($dt$) is expressed as a function, where the addition of [10]Be is represented as the atmospheric flux to the surface ($Q$, atoms cm$^{-2}$ yr$^{-1}$), removal due to radioactive decay is represented by a disintegration constant ($\lambda$, yr$^{-1}$), and erosion ($E$, cm yr$^{-1}$) is with respect to soil density ($\rho$, g cm$^{-3}$) (Eq. 1). Particle mobility into the soil column is represented by a diffusion constant ($D$, cm$^2$ yr$^{-1}$) multiplied by a concentration gradient.

$$\frac{dN}{dt} = Q - \lambda N - E\frac{dN}{dz} - D\frac{d^2N}{dz^2} \tag{1}$$

However, this function is highly dependent on $dz$, which represents an unknown value of depth into the soil column which is influenced by meteoric [10]Be deposition and removal. Additionally, the soil diffusion term is unconstrained and likely varies with depth. We accounted for these uncertainties and other uncertainties regarding [10]Be migration in the soil column by calculating the inventory ($I$, atoms cm$^{-2}$) of the soil (Eq. 2), assuming that $Q$ had not changed systematically over the accumulation interval (Graly et al., 2010; Pavich et al., 1986). The inventory is the total sum of meteoric [10]Be atoms in the soil profile and the change in inventory due to deposition, decay, and surface erosion is related surface exposure age (Eq. 3).

$$I = \sum N \cdot \rho \cdot dz \tag{2}$$

$$\frac{dI}{dt} = Q - \lambda I - EN \tag{3}$$

If the inventory of meteoric [10]Be in the soil profile, the concentration at the surface, and soil density are known, and published values for erosion and [10]Be flux to the surface are used, we can combine Eqs. (1-3), and solve for time ($t$, years) (Eq. 4).

$t = -\frac{1}{\lambda} \cdot \ln\left[1 - \frac{\lambda I}{Q - E\rho N}\right]$          (4)
Equation (4) provides a maximum exposure age assuming that the soil profile did not contain meteoric [10]Be before
it was exposed to the surface ($N_0 = 0$). Since our exposure age dating technique relies on the number of [10]Be atoms within
the sediment column ($I$), any pre-existing [10]Be atoms in the soil ($N_0 \neq 0$) causes the calculated age to be an overestimate (Fig.
3c-d) (Graly et al., 2010). Meteoric [10]Be concentrations typically decrease with depth until they reach a "background" level
(Graly et al., 2010). The background is identified as the point where the concentration of meteoric [10]Be is constant with
depth ($\frac{dN}{dz} = 0$). Typically, the background values can be used to calculate an initial inventory ($I_i$, atoms cm$^{-2}$) using Eq. (5),
where $N_z$ is the [10]Be concentration (atoms g$^{-1}$) at the bottom of the profile ($z$, cm), and correct the observed total inventory
(Eq. 6). In this case, we assume that the initial concentration of meteoric [10]Be is isotropic. However, an accurate initial
inventory can only be determined for soil profiles which decrease in [10]Be concentrations to background levels due to the
downward transport of [10]Be from the surface. This may not be the case in areas of permafrost where [10]Be is restricted to the
active layer (Bierman et al., 2014).
$I_i = N_z \cdot \rho \cdot z$          (5)
$t = -\frac{1}{\lambda} \cdot \ln\left[1 - \frac{(I - I_i)\lambda}{Q - E\rho N}\right]$          (6)
Additionally, the initial inventory can be influenced by repeated glacial advance and retreat during glacial-
interglacial cycles. For this case, the soil has "inherited" [10]Be during each subsequent exposure to the atmosphere, some of
which may have been removed with eroded soil (Fig. 3c-d). For constructional landforms, such as moraines, the inheritance
is equal to the background/initial inventory. Without information on drift sequences, it is difficult to correct the measured
inventory for inheritance by distinguishing meteoric [10]Be that was deposited after the most recent ice retreat from [10]Be that
was deposited during previous interglacial periods. Instead, only ages that represent total time of exposure through glacial-
interglacial cycles, likely as overestimates, can be reported with confidence.

### 4.3.1. Model variable selection and key assumptions

The exposure age calculations are dependent on the selected values for the variables in Eq. (1-6). We chose a flux value ($Q$) of $1.3 \times 10^5$ atoms cm$^{-2}$ yr$^{-1}$ from Taylor Dome (Steig et al., 1995) due to a similar climate to that of the CTAM and an absence of local meteoric $^{10}$Be flux data. Soil density ($\rho$) across the Shackleton Glacier region was approximately 2 g cm$^{-3}$. While we did not calculate erosion rates, previous studies have estimated rates from rocks of 1 to 65 cm Ma$^{-1}$ in Victoria Land (Ivy-Ochs et al., 1995; Margerison et al., 2005; Morgan et al., 2010; Strasky et al., 2009; Summerfield et al., 1999) and 5 to 35 cm Ma$^{-1}$ further south in the Transantarctic Mountains (Ackert and Kurz, 2004; Balter-Kennedy et al., 2020; Morgan et al., 2010). Balter-Kennedy et al. (2020) determined that erosion rates for boulders at Roberts Massif which were less than 2 cm Ma$^{-1}$. However, we chose a conservative value of 5 cm Ma$^{-1}$ for our analysis of the Shackleton Glacier region.

It is important to note two key assumptions in our variable selection and model development. First, we have assumed a uniform erosion rate across the region. Given the variety of surface features at each location (Table 2), some locations on valley floors, for example, may have increased surface concentrations of meteoric $^{10}$Be due to entrapment of wind-blown fine-grained sediments. Locations on hillslopes and valley walls might have higher erosion rates (Morgan et al., 2010; Schiller et al., 2009). We assumed that deflation of fine-grained material had occurred rapidly on the flat surfaces we sampled due to strong winds over the poorly consolidated tills following soil exposure (Lancaster et al., 2010). Due to a deficit of soil erosion data in the CTAM, we calculated exposure ages (Eq. 6) with the 5 cm Ma$^{-1}$ erosion value and without the erosion/deposition term ($E$=0). Second, we attempted to estimate the background concentrations and initial inventory for each sample collected furthest from the glacier. We hypothesized that these samples were potentially exposed throughout at least the LGM and had negligible inheritance, though this was merely an assumption. With the possibility of overestimating or underestimating the exposure ages, we solved Eq. 6 both with and without estimated initial inventory terms. For all samples, including those without depth profile measurements, we utilized an empirical relationship derived between surface/maximum meteoric $^{10}$Be concentration and measured inventory to estimate surface exposure ages (see Section 5.3.3) (Graly et al., 2010). Regarding our $NO_3^-$ measurements, we assumed that aside from solubilization and salt translocation, $NO_3^-$ is preserved in the soils and any volatilization or photolysis is negligible (Diaz et al., 2020; Jackson et al., 2016).

## 5. Results

**5.1. Concentrations of meteoric [10]Be and depth profile composition**

Sediment grain size is similar among the three soil profiles collected from Roberts Massif, Bennett Platform, and Thanksgiving Valley; the soils are primarily comprised of sand-sized particles, with less silt-sized and smaller material (Fig. 4). The proportions of silt and gravel are similar at Roberts Massif, although the majority of the profile is sand-sized. Thanksgiving Valley has the least fine material, while Bennett Platform has a more even grain size distribution. The deepest profile is from Thanksgiving Valley, while the Roberts Massif and Bennett Platform profiles are half the depth. All three profiles are ice-cemented at the bottom and are shallow compared those collected from the McMurdo Dry Valleys (Dickinson et al., 2012b; Schiller et al., 2009; Valletta et al., 2015).

Surface concentrations of meteoric [10]Be span more than an order of magnitude in the Shackleton Glacier region and range from 2.9 x $10^8$ atoms g$^{-1}$ at Mount Speed to 73 x $10^8$ atoms g$^{-1}$ at Roberts Massif (Fig. 5; Table 3). At individual sites where samples were collected at two locations, concentrations are typically highest for the samples furthest from the glacier, with notable exceptions at Roberts Massif and Thanksgiving Valley (Fig. 5). This trend is expected since our sampling plan was designed to capture both recently exposed soils (near the glacier(s)) and soils which have been exposed throughout the LGM and possibly other glacial periods. The measured inventories (Eq. 2) vary from 0.57 x $10^{11}$ atoms at Bennett Platform to 1.5 x $10^{11}$ atoms at Roberts Massif (Table 4).

The meteoric [10]Be depth profiles differ between Roberts Massif, Thanksgiving Valley, and Bennett Platform. The profile from Roberts Massif has the highest overall concentrations (Fig. 6). Within the profile, the 5-10 cm sampling interval has the highest concentration, followed by the bottom of the profile, then the surface. The profile behavior for Thanksgiving Valley is similar, though the differences in concentrations within both profiles are relatively small. Bennett Platform is the only location where the surface concentration is the highest compared to the remainder of the profile, which decreases with depth (Fig. 6). Although we sampled the entirety of the active layer where particle mobility throughout the soil column occurs, no depth profiles appear to decrease to background levels needed to calculate an initial meteoric [10]Be inventory (Eq. 5). As a result, we are not able to correct the measured inventory for background [10]Be nor are we able estimate the inherited [10]Be concentration in the soil (Eq. 6).

**5.2. Relationship between meteoric $^{10}$Be and $NO_3^-$**

Measured concentrations of $NO_3^-$ span four orders of magnitude across the seven depth profiles sampled in the Shackleton Glacier region (Fig. S1; Table S2). The lowest concentration is from Mt. Franke, ~1 µg g$^{-1}$; the highest concentration is from Roberts Massif, 15 mg g$^{-1}$. In addition, similar to the meteoric $^{10}$Be profiles, the $NO_3^-$ concentrations are highest for the samples that were collected furthest from the coast and at the highest elevations (Table S2). The concentrations of $NO_3^-$ and meteoric $^{10}$Be are compared for Roberts Massif, Bennett Platform, and Thanksgiving Valley (Fig. 6b). In general, the profiles from Roberts Massif and Thanksgiving Valley are similar, where $^{10}$Be and $NO_3^-$ concentrations are highest just below the surface in the 5-10 cm interval and are fairly consistent throughout the profile. The $NO_3^-$ depth profile mirrors the $^{10}$Be profile at Bennett Platform – while $^{10}$Be concentrations decrease with depth, the $NO_3^-$ concentration increases with depth.

Since the behaviors of $NO_3^-$ and $^{10}$Be are parallel or mirrored (as in the case for Bennett Platform), we further evaluate their relationship. When regressed on log scales, $NO_3^-$ and $^{10}$Be have a strong power-law relationship with $R^2$ values ranging from 0.66 to 0.99 (Fig. 6c). The power-law slope for Roberts Massif and Thanksgiving Valley is positive, while the Bennett Platform has a negative slope. Given this regressed relationship, it is possible to estimate $^{10}$Be concentrations using $NO_3^-$ concentrations (see Section 5.3.2).

**5.3. Relative exposure age calculations and estimates**

**5.3.1 "Measured" exposure ages from Roberts Massif, Bennett Platform, and Thanksgiving Valley**

We calculated exposure ages for the samples furthest from the glacier for Roberts Massif, Bennett Platform, and Thanksgiving Valley using Eq. 4, both with and without the erosion term (Table 3). The exposure ages with erosion range from 120 ka to 4.15 Ma, and the ages without erosion range from 110 ka to 1.67 Ma for Bennett Platform and Roberts Massif, respectively. Thanksgiving Valley is intermediate with an exposure age of 540 ka with erosion and 500 ka without erosion. Since we are not able to correct for initial inventory or inheritance, the exposure ages with the erosion term represent maximum ages. The erosion rate we estimated is relatively low compared to the calculated exposure ages for most samples and would only slightly influence the measured exposure ages. Roberts Massif is an exception where the inclusion

or exclusion of erosion alters the measured age by over 50%. Moreover, the ages without erosion terms are probably
overestimates as well without inheritance corrections.

### 5.3.2 "Estimated" exposure ages using $NO_3^-$ relationship

As we suggest in Section 5.2, the power-law relationship between $NO_3^-$ and meteoric $^{10}Be$ can be used to estimate
$^{10}Be$ concentrations from $NO_3^-$ concentrations. Since we measured $NO_3^-$ concentrations in all seven depth profiles, we
compared the profile concentrations and shape from the four profiles without $^{10}Be$ depth measurements (Mt. Augustana,
Schroeder Hill, Mt. Franke, and Mt. Heekin) to the Roberts Massif, Bennett Platform, and Thanksgiving Valley profiles with
both measurements (Fig. S1). Our calculation fundamentally assumes no loss of $NO_3^-$ due to prolonged surface exposure and
that $NO_3^-$ profiles which have similar shapes among the sites might have similar $^{10}Be$ profile shapes as well. The profiles are
all fairly homogenous and most similar to the profile from Thanksgiving Valley, though Schroeder Hill is most similar to
Roberts Massif (Fig. S1). Applying the power-law relationship from Thanksgiving Valley to Mt. Augustana, Mt. Franke and
Mt. Heekin, and the relationship from Roberts Massif to Schroeder Hill, we provide estimates of meteoric $^{10}Be$
concentrations for the entire depth profile (Table S2) and use these concentrations to calculate an "estimated" inventory
using Eq. 2 (Table 4). Further, the estimated inventories are used to estimate exposure ages using Eq. 4, both with and
without the erosion term.
The estimated inventories (using the $NO_3^-$ power-law relationship) with erosion range from 0.14 x $10^{11}$ atoms at
Bennett Platform to 1.5 x $10^{11}$ atoms at Roberts Massif (Table 4). The measured and estimated inventories differ by ~3-18%.
The estimated exposure ages using the estimated inventory range from 120 ka to 4.54 Ma with erosion, and the ages without
erosion range from 110 ka to 1.74 Ma for Bennett Platform and Roberts Massif, respectively (Table 4). The measured and
$NO_3^-$ estimated exposure ages, both with and without erosion, only differ by ~4-20% for Roberts Massif, Bennett Platform,
and Thanksgiving Valley. Since we cannot calculate exposure ages using only $^{10}Be$ for the profiles from Schroeder Hill, Mt.
Augustana, Mt. Heekin, and Mt. Franke, we are not able to make similar age comparisons. However, we can compare the
estimated surface $^{10}Be$ concentrations using $NO_3^-$ to the measured $^{10}Be$ concentrations. The percent differences at Schroeder
Hill and Mt. Heekin are 4% and 7%, respectively, while Mt. Augustana and Mt. Franke have higher differences of 36% and
40%, respectively (Tables 3 and S2).

### 5.3.3 "Inferred" exposure ages using inventory relationship

Similar to our exposure age estimates using $NO_3^-$ concentrations, we used the relationship between the maximum meteoric [10]Be concentration in the soil profile and the meteoric [10]Be inventory (Graly et al., 2010) to "infer" [10]Be inventories and calculate maximum exposure ages for all eleven locations, again, with and without erosion (Fig. 7; Table 5). As is the case for Roberts Massif and Thanksgiving Valley, the highest concentrations may not always be at the surface for all locations; however, the relationship is sufficiently strong to provide an estimate of the [10]Be inventory and thus an age estimate (Fig. 7). Compared to the measured inventories from Roberts Massif, Bennett Platform, and Thanksgiving Valley, the inferred inventories differ by ~16-130%. The inferred exposure ages with erosion range from 58 ka to >6.5 Ma, and the ages without erosion range from 57 ka to 1.94 Ma for Mt. Speed and Roberts Massif, respectively (Table 4). With the exception of Roberts Massif, Thanksgiving Valley, and Mt. Speed, the oldest surfaces are those which we sampled furthest from the glacier, which is consistent with our sampling methodology to capture younger and older soils. The sample from Roberts Massif collected closest to the glacier has an estimated exposure age that is outside the model limits (>6.5 Ma). The measured exposure ages and the inferred exposure ages differ by ~49-75% with erosion and ~15-75% without erosion. The greatest differences between the ages are at Bennett Platform.

### 6. Discussion

Meteoric [10]Be concentrations and surface exposure ages vary widely across the Shackleton Glacier region and at individual locations. Although these data are only measurements from discrete points on the landscape, they constrain relative terrestrial exposure ages. These meteoric [10]Be and $NO_3^-$ data contribute to growing exposure age measurements, which can inform climate, landscape development, and biogeography. The Shackleton Glacier region soil profiles have the highest meteoric [10]Be concentrations ($\sim 10^9$ atoms $g^{-1}$) yet measured in Earth's polar regions (Fig. 6a). Though our profiles are shallower than profiles from the MDV and Victoria Land in Antarctica (Dickinson et al., 2012a; Schiller et al., 2009; Valletta et al., 2015) and Sweden and Alaska in the Arctic (Bierman et al., 2014; Ebert et al., 2012), the soils from these previous studies reached background concentrations of [10]Be within the top 40 cm, which is close to our maximum depth of 30 cm at Thanksgiving Valley. The Bennett Platform soil profile is most similar to the soil profiles from other regions in

Antarctica, as they have decreasing $^{10}$Be concentrations with depth, while Thanksgiving Valley and Roberts Massif are relatively homogenous and more similar to profiles from the Arctic.

**6.1. Calculated, estimated, and inferred exposure age validation**

Considering the novelty of our approach, we sought to test and externally validate the exposure ages. Our calculated, estimated, and inferred exposure ages are consistent with the limited *in-situ* exposure age data from the Shackleton Glacier region (http://antarctica.ice-d.org; Balco, 2020). Exposure ages from glacial erratic boulders using *in-situ* cosmogenic measurements were determined in previous studies (Balter-Kennedy et al., 2020; Balco, 2020; http://antarctica.ice-d.org) from Roberts Massif, Thanksgiving Valley, and Mt. Franke (Figs. 8 and 9). From *in-situ* $^{10}$Be, $^{26}$Al, $^{3}$He, and $^{21}$Ne data, exposure ages on the northern flank of Roberts Massif range from 1.10 Ma to 3.26 Ma (Balter-Kennedy et al., 2020; Balco, 2020; http://antarctica.ice-d.org), and our measured, estimated, and inferred ages without erosion are 1.67 Ma, 1.74 Ma, and 1.94 Ma, respectively. Our ages, which are likely overestimates due to a lack of initial inventory or inheritance corrections, are comparable to these nearby *in-situ* ages at similar elevations (Figs. 8 and 9). The ages with the erosion term are greater and outside the range from Balter-Kennedy et al. (2020). This suggests that soil erosion rates are probably low at Roberts Massif, and the initial inventory and $^{10}$Be inheritance from previous exposures are likely significantly smaller than the measured inventory. Otherwise, the corrected meteoric $^{10}$Be exposure ages would be much greater than the *in-situ* ages.

To the north, the *in-situ* ages from erratic boulders at Thanksgiving Valley vary greatly from ~4.3 ka near the glacier to 450 ka at higher elevations, though most ages appear to be around 30 ka (Figs. 8 and 9) (Balco, 2020; http://antarctica.ice-d.org). Our exposure ages are greater than most previous ages. In particular, the sample collected closest to Shackleton Glacier has an inferred age two orders of magnitude higher than the *in-situ* age from a nearby glacial erratic at the same elevation (Fig. 9). Given the location (~100 m from the glacier) and young nearby *in-situ* age (~4.3 ka), this location was likely covered during the LGM and other glacial periods. Therefore, considering the high surface concentration of meteoric $^{10}$Be for this sample, it is possible that there is an additional delivery mechanism of $^{10}$Be, such as deposition of material deflated from the valley walls or at high elevations, or an otherwise large inherited component.

Closer to the Ross Ice Shelf, the *in-situ* ages from Mt. Franke range from ~29 ka to 220 ka. Our estimated age
without erosion is at the top that range at 220 ka, though the inferred ages are considerable younger at 94 ka and 72 ka (Table
5). Similar to Roberts Massif, our ages from Mt. Franke ages are comparable to the *in-situ* ages from similar elevations (Fig.
9). Here, soil erosion, initial inventory, and inheritance likely minimally influence the measured $^{10}$Be inventory. We argue
that while the measured, estimated, and inferred ages from the Shackleton Glacier region are similar to *in-situ* ages, they are
likely an overestimate and most useful from a relative perspective in understanding which surfaces have been exposed for
longer than others.
**6.2. NO$_3^-$ as an efficient inventory and exposure age estimation tool**
This is the first study to use NO$_3^-$ concentrations to directly estimate meteoric $^{10}$Be concentrations study, but not the
first to attempt to use water-soluble NO$_3^-$ and salts to help understand glacial history. Previous studies have argued that
atmosphere-derived salt concentrations at the surface may correlate with exposure ages and wetting ages in Antarctica
(Graham et al., 2002; Graly et al., 2018; Lyons et al., 2016; Schiller et al., 2009). Graly et al. (2018) showed that, in
particular, water-soluble NO$_3^-$ and boron exhibited the strongest relationships with exposure age ($R^2 = 0.9$ and 0.99,
respectively). Lyons et al. (2016) used nitrate concentrations to estimate the amount of time since the soils were last wetted
and Graham et al. (2002) attempted to calculate exposure ages using the inventory of nitrate in the soil. Graly et al. (2018)
argue that boron is preferable to nitrate due to concerns related to nitrate mobility under sub-arid conditions (e.g. Frey et al.,
2009; Michalski et al., 2005), and given that uncertainties in local accumulation rates and ion transport can result in
inaccurate ages when using NO$_3^-$ alone (Graham et al., 2002; Schiller et al., 2009). Based on the results presented here for
hyper-arid CTAM ice-free regions and the concerns with boron mobility depending on whether the B species present in the
soils is BO$_3^{3-}$ (borate) or H$_3$BO$_3$ (boric acid), we conclude that NO$_3^-$ appears suitable for relative age dating and for
producing age estimates.
We show that the differences between measured $^{10}$Be inventories and estimated inventories using NO$_3^-$ are low (see
Section 5.3.2) and argue that the power-law relationship between meteoric $^{10}$Be and NO$_3^-$ can be used to expand our current
exposure age database for the TAM; compared to cosmogenic radionuclide analyses, NO$_3^-$ analyses are rapid and cost
effective. However, a model using NO$_3^-$ or salts alone is likely insufficient, unless the anion accumulation rates are known
(Graham et al., 2002; Schiller et al., 2009). Though the regressions between $NO_3^-$ and $^{10}Be$ are strong (Fig. 6c), each of the
three profiles from Roberts Massif, Bennett Platform, and Thanksgiving Valley have different regression coefficients and
slopes. In other words, the nature of the relationship between meteoric $^{10}Be$ and $NO_3^-$ varies across the Shackleton Glacier
region and varies depending on the location. This is likely due to differences in $NO_3^-$ and $^{10}Be$ transport and mobility in
different surface environments and under different local climates. To address these uncertainties, some $^{10}Be$ data (surface
samples for all locations and a few depth profiles) are necessary to constrain the most accurate regression and minimize the
associated error.
We tested our meteoric $^{10}Be$ – $NO_3^-$ model with data from Arena Valley in the MDV (Graham et al., 2002) and
found that our model is applicable to other TAM ice-free areas. Similar to the Shackleton Glacier region soils, the soils from
Arena Valley are hyper-arid with high concentrations of $NO_3^-$ and other salts (Graham et al., 2002). Precipitation in the
MDV is low at ~5 cm water equivalent each year (Fountain et al., 1999), though $NO_3^-$ and other water-soluble salts at the
surface can be wetted and mobilized. The highest $NO_3^-$ concentrations are at 10 cm depth, while $^{10}Be$ concentrations are
highest at the surface and decrease with depth, indicating vertical transport of $NO_3^-$ through time (Graham et al., 2002). The
power-law relationship between $^{10}Be$ and $NO_3^-$ throughout the profile is weaker for the Arena Valley samples compared to
Shackleton Glacier samples; there is a stronger power-law correlation in the top 20 cm ($R^2 = 0.61$) compared to the bottom
70 cm ($R^2 < 0.01$), though the profile is considerably deeper (110 cm). Using the power-law relationship from Bennett
Platform, which mostly closely resembles the profile behavior for Arena Valley given the negative regression slope, the
estimated inventory is $5.4 \times 10^{10}$ atoms. The measured inventory is of the same order of magnitude, $1.3 \times 10^{10}$ atoms,
indicating a moderate model fit. Applying the power-law relationship from Arena Valley, the estimated inventory is $9.2 \times$
$10^9$ atoms, which is ~27% lower than the measured inventory. These results indicate that, although the Shackleton Glacier
region is nearly 900 km from Arena Valley, the correlation between $NO_3^-$ and meteoric $^{10}Be$ is widely applicable in hyper-
arid soils. However, as stated previously, $NO_3^-$ and meteoric $^{10}Be$ data are needed to ascertain the general profile and slope
behavior within the region. Additionally, though our $NO_3^-$ estimated ages are validated using *in-situ* data from previous
studies, the $NO_3^-$ dating tool will need to be further evaluated with additional measurements and erosion, initial inventory,
and inheritance corrections.

### 6.3. Implications for paleoclimate and ice sheet dynamics

Our work demonstrates that $NO_3^-$ and $^{10}Be$ are correlated in much of the Shackleton Glacier region and this relationship has important implications for understanding landscape disturbance, either by meltwater or glacier overriding. Exposure age data from across Antarctica show that a polar desert regime began in the mid-Miocene and has persisted into modern time (Lewis et al., 2008; Marchant et al., 1996; Spector and Balco, 2020; Valletta et al., 2015). Additionally, Barrett (2013) provides a detailed review of studies focused on Antarctic glacial history, particularly centered around the "stabilist vs. dynamicist" debate concerning the overall stability of the EAIS. Interpreting 40+ years of data from published literature, they conclude that the EAIS is stabile in the interior with retreat occurring along the margins, including at outlet glaciers (Golledge et al., 2012). Given these findings, we would expect $NO_3^-$ and meteoric $^{10}Be$ concentrations to be correlated in hyper-arid Antarctic soils, such as those from the Shackleton Glacier region, as both constituents are derived from atmospheric deposition with minimal alteration at the surface. The major differences between the two concern transport mechanisms. Meteoric $^{10}Be$ transport is limited by clay particle mobility and $NO_3^-$ is mobile upon soil wetting. Deviations in the expected relationship between $^{10}Be$ and $NO_3^-$ can inform knowledge of surface processes in the TAM.

If we assume an "ideal" situation where an undisturbed hyper-arid soil has accumulated meteoric $^{10}Be$ (Fig. 3a-b), $^{10}Be$ concentrations would be highest at the surface and decrease to background levels at depth. None of the profiles we sampled and measured for meteoric $^{10}Be$ and $NO_3^-$ reached background concentrations. All profiles were sampled until frozen soil was reached (or bedrock at Schroeder Hill) (Fig. S1), demonstrating an active layer much shallower than those from the MDV (Graham et al., 2002; Schiller et al., 2009; Valletta et al., 2015). This suggests that $^{10}Be$-laden particles were able to migrate deeper in the past and mobility has been relatively recently (within the $^{10}Be$ half-life) limited to the top ~20 cm for most the Shackleton Glacier region. Though clay particle translocation by percolating water can explain the correlated behavior of $^{10}Be$ and $NO_3^-$ at Roberts Massif and Thanksgiving Valley, it is unlikely that the region had sufficient precipitation for significant percolation over the last 14 Ma (Menzies et al., 2006). The concentrations of fine particles in the soil profiles also do not change significantly with depth, as would be expected if large precipitation or melt events were frequent (Fig. 4).

Similar to Arena Valley and Wright Valley in the MDV (Graham et al., 2002; Schiller et al., 2009), $NO_3^-$
concentrations are highest just beneath the surface at Roberts Massif, indicating shallow salt migration under an arid climate.
These data suggest that the samples furthest inland at Roberts Massif and Thanksgiving Valley have been undisturbed since
at least the middle to late Pleistocene given the soil exposure ages. Although meteoric $^{10}$Be and $NO_3^-$ at Bennett Platform are
mirrored with a negative power-law slope, we argue that the difference is not due to $NO_3^-$ mobility, but instead $^{10}$Be
deposition. Bennett Platform was the only location we sampled on a large moraine (Fig. 2c), and as such, we would expect
minimal inheritance with $^{10}$Be decreasing at depth. This is generally the observed behavior, with significantly higher surface
concentrations. The $NO_3^-$ profile behavior is similar to those throughout the Shackleton Glacier region, though the
concentrations continue to increase with depth, possibly indicating minor percolation of $NO_3^-$ rich brine. What may be
considered the "anomalous" data point is the surface concentration of meteoric $^{10}$Be. Even though we sampled a
constructional landform, the sample was collected between two boulder lines in a small, local depression (~1 m) (Table 2). It
is probably no coincidence that this location also has the greatest proportion of fine-grained material in the soil profile. The
two boulder lines impede wind flow and act as a sediment and snow trap, resulting in a higher concentration of meteoric $^{10}$Be
than expected simply from atmospheric deposition. In this case, an additional deposition term (superseding any erosion)
needs to be considered to accurately date the moraine, and the current exposure age we measured may be an overestimate.
The youngest surfaces we sampled are those from the lowest elevations and closest to the Ross Ice Shelf (Fig. 10).
This is generally consistent with previous glacial modeling studies which show that the greatest fluctuations in glacier height
during the LGM were along outlet glacier and ice shelf margins (Golledge et al., 2012; MacKintosh et al., 2011; Mackintosh
et al., 2014). However, erosion rates are low throughout Antarctica (Balter-Kennedy et al., 2020; Ivy-Ochs et al., 1995;
Morgan et al., 2010) and would not drastically impact our relatively young inferred ages (Fig. 10). Additionally, background
concentrations of meteoric $^{10}$Be in other Antarctic soil profiles are often approximately one to two order of magnitude lower
than surface concentrations (Fig. 6). With these considerations, the Mt. Speed, Mt. Wasko, and Mt. Franke samples were all
likely covered by the Shackleton Glacier during the LGM, as well as the lower elevation, closest to the glacier samples from
Mt. Heekin, Bennett Platform, and Mt. Augustana. The samples we collected near the head of Shackleton Glacier encompass
a range of ages, where lower elevation soils are relatively younger, though the soils from Schroeder Hill and Roberts Massif
have likely been exposed since the early Pleistocene (Fig. 10).
Sirius Group deposits were observed at Roberts Massif and were deposited as the Shackleton Glacier retreated in
this region (Fig. 2a). Evidence for a dynamic EAIS is derived primarily from the diamictite rocks (tills) of the Sirius Group,
which are found throughout the TAM and include well-documented outcrops in the Shackleton Glacier region, but their age
is unknown (Hambrey et al., 2003). Some of the deposits contain pieces of shrubby vegetation, suggesting that the Sirius
Group formed under conditions warmer than present with trees occupying inland portions of Antarctica (Webb et al., 1984,
1996; Webb and Harwood, 1991). Sparse marine diatoms found in the sediments were initially interpreted as evidence for
formation of the Sirius Group via glacial over riding of the TAM during the warmer Pliocene (Barrett et al., 1992), though it
is now argued that the marine diatoms were wind-derived contamination, indicating that the Sirius Group is older (Scherer et
al., 2016; Stroeven et al., 1996). We document a large diamictite at site RM2-8 that is underlain by soils with an inferred age
of at least 1.9 Ma, possibly greater than 6.5 Ma. These exposure ages suggest that the loose Sirius Group diamict was
deposited at Roberts Massif some point after the Pliocene. While these data cannot constrain the age of the formation, we
suggest that the diamict could have formed prior to the Pliocene and was transported during the Pleistocene glaciations.
**7. Conclusions**
We measured concentrations of meteoric $^{10}$Be and $NO_3^-$ in soils from eleven ice-free areas along the Shackleton
Glacier, Antarctica, which include the highest measured meteoric $^{10}$Be concentrations from the polar regions. Measured
(using meteoric $^{10}$Be inventories), estimated (using the power-law relationship between $NO_3^-$ and $^{10}$Be), and inferred (using
the relationship between maximum $^{10}$Be and total inventory) exposure ages were calculated and ranged from 58 ka to >6.5
Ma with an estimated erosion component and 57 ka to 1.9 Ma without erosion. In general, there is good agreement between
the three techniques.
The estimated and inferred ages without erosion at Roberts Massif, Thanksgiving Valley, and Mt. Frank are similar
to nearby *in-situ* ages from previous studies. In particular, relating $NO_3^-$ concentrations to $^{10}$Be measurements results an
efficient method to attain a greater number of exposure ages in the CTAM, a region with currently sparse meteoric $^{10}$Be data.

However, the power-law relationship between $NO_3^-$ and $^{10}Be$ had either a positive or negative slope depending on the location, therefore the widespread applicability of this tool needs to be further evaluated. Additionally, though we assumed an erosion rate for the region, some soils in local topographic lows probably have a positive particle flux.

Since $NO_3^-$ and $^{10}Be$ are both derived from atmospheric deposition, we expect the shape of their accumulation profiles to be similar at depth in hyper-arid soils. In general, this was true for Roberts Massif and Thanksgiving Valley, while $NO_3^-$ and $^{10}Be$ concentrations were mirrored at Bennett Platform. We conclude that much of the southern Shackleton Glacier region has maintained persistent arid conditions since at least the Pleistocene, though the region was warmer and wetter in the past, as evidenced by frozen soil at the bottom of our depth profiles. The onset of aridity is particularly important in understanding refugia and ecological succession in TAM soils. Since the region has remained hyper-arid and undisturbed for upwards of a few million years, prolonged exposure has resulted in the accumulation of salts at high concentrations in the soils. As such, it is an enigma how soil organisms have persisted throughout glacial-interglacial cycles. However, it is possible that organisms have survived near the glacier at locations like Mt. Augustana, where glacial advance appears to have been minimal during the LGM, but seasonal summer melt has the potential to solubilize salts.

Overall, our data show that the relatively youngest soils we sampled were at lower elevations near the Shackleton Glacier terminus and lower elevations further inland (typically near the glacier). Our sampling scheme was successful in capturing a range of surface exposure ages which contribute to growing archives in the CTAM. We hope that future studies will address the outstanding issues regarding inheritance dynamics of meteoric $^{10}Be$ in disturbed environments and particle erosion/deposition rates.

**Author Contributions**

The project was designed and funded by BJA, DHW, IDH, NF, and WBL. Fieldwork was conducted by BJA, DHW, IDH, NF, and MAD. LBC, PRB, and MAD prepared the samples for meteoric $^{10}$Be analysis and MAD analyzed the samples for $NO_3^-$. MAD wrote the article with contributions and edits from all authors.

**Data Availability Statement**

The datasets generated for this study are included in the article or supplementary materials.

**Competing Interests**

The authors declare that they have no conflict of interest.

**Acknowledgments**

We thank the United States Antarctic Program (USAP), Antarctic Science Contractors (ASC), Petroleum Helicopters Inc. (PHI), and Marci Shaver-Adams for logistical and field support. We especially thank Dr. Marc Caffe and the Purdue University PRIME Lab for their assistance with AMS measurements. Additionally, we thank Dr. Andrew Christ at University of Vermont for thoughtful discussions and Dr. Sue Welch and Daniel Gilbert at The Ohio State University for help with initial laboratory analyses. We appreciate the detailed and thoughtful suggestions and edits from Dr. Brent Goehring and an anonymous reviewer which have greatly improved this manuscript. This work was supported by NSF OPP grants 1341631 (WBL), 1341618 (DHW), 1341629 (NF), 1341736 (BJA), NSF GRFP fellowship 60041697 (MAD), and a PRIME Lab seed proposal (MAD). Sample preparation and LBC's time supported by NSF EAR 1735676. Geospatial support for this work provided by the Polar Geospatial Center under NSF OPP grants 1043681 and 1559691.

**Figures:**

**Figure 1:** Overview map of the Shackleton Glacier region, located in the Queen Maud Mountains of the Central Transantarctic Mountains. The red circles represent our eleven sampling locations, with an emphasis on Roberts Massif (orange), Bennett Platform (green), and Thanksgiving Valley (blue), which have the most comprehensive dataset in this study. The bedrock serves as primary weathering product for soil formation (Elliot and Fanning, 2008; Paulsen et al., 2004). Base maps provided by the Polar Geospatial Center.

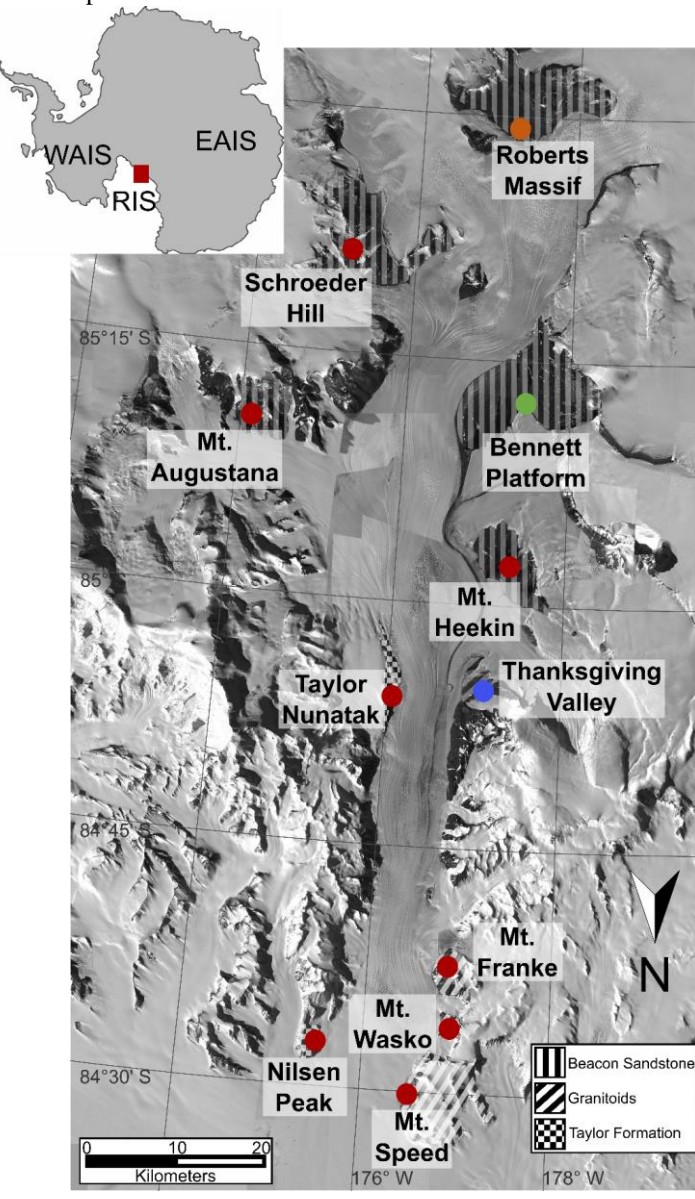

537

**Figure 2:** The Sirius Group was documented at Roberts Massif near the RM2-8 sampling location (a). Small moraines were observed at Roberts Massif (b) and large moraines at Bennett Platform (c).

540

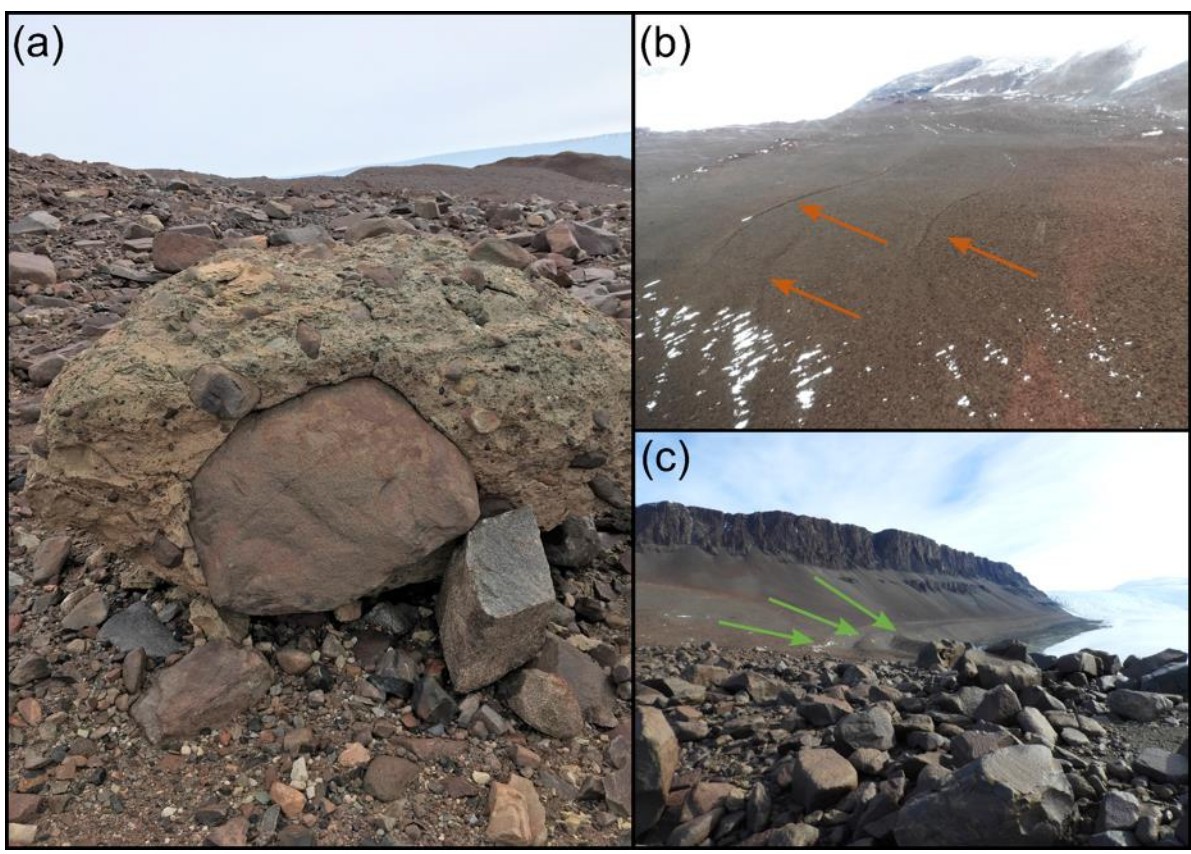

541

**Figure 3:** Conceptual diagram of meteoric [10]Be accumulation in soils during glacial advance and retreat. In "ideal" conditions, [10]Be accumulates in exposed soils and [10]Be concentrations beneath the glacier are negligible at background levels (a). As the glacier retreats, [10]Be can begin accumulating in the recently exposed soil and an inventory can be measured to calculate exposure ages. In the case where the glacier has waxed and waned numerous times and the soils already contain a non-negligible "inheritance" concentration of [10]Be, the inventories need to be corrected for [10]Be inheritance (c-d) to accurately determine exposure ages.

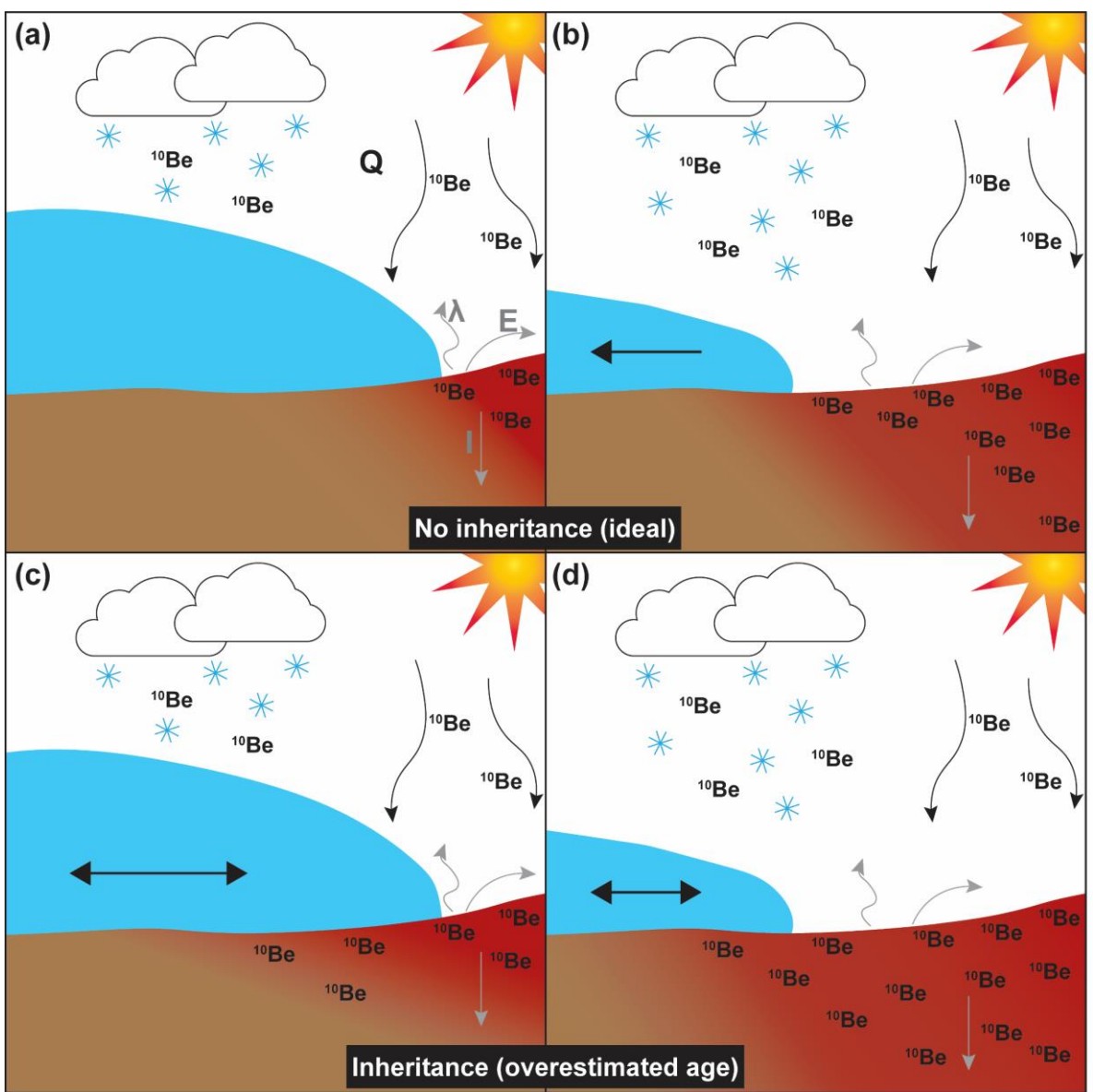

**Figure 4:** The grain size composition of soil profiles collected from Roberts Massif (a, orange), Bennett Platform (b, green),
and Thanksgiving Valley (c, blue). The soil pits from Bennett Platform and Thanksgiving Valley are also shown with
distinct soil horizons.

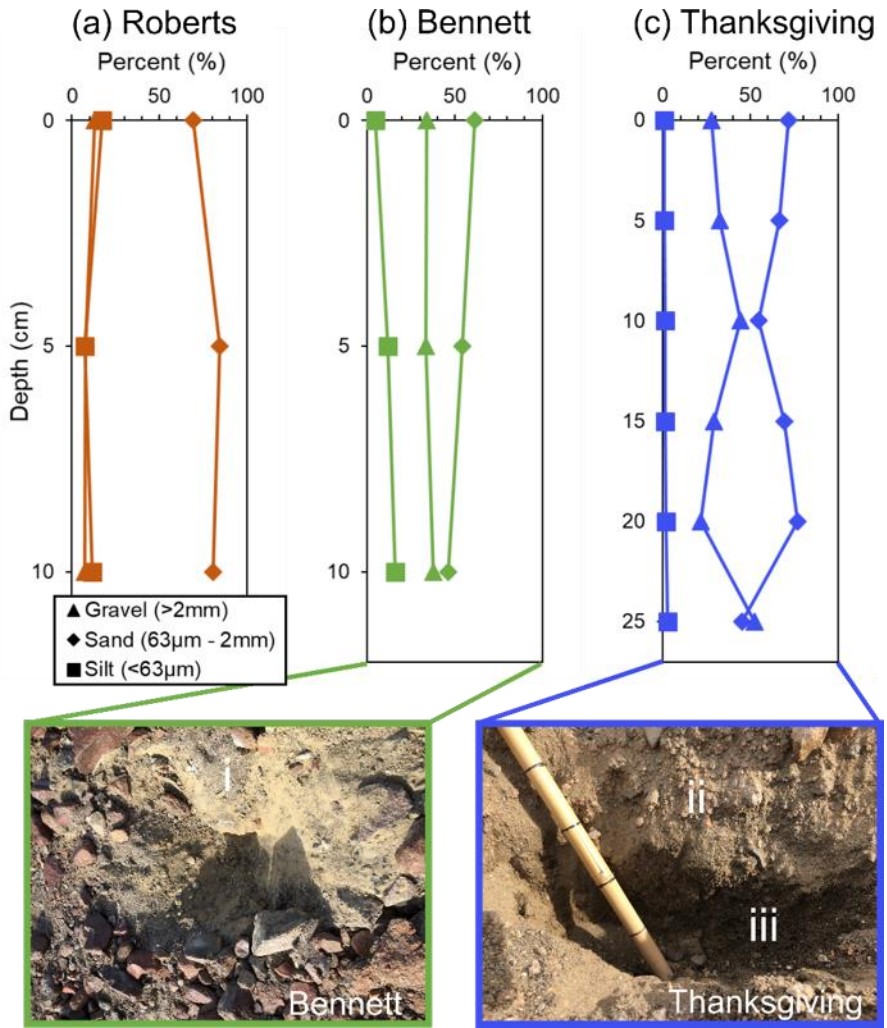


**Figure 5:** Spatial distribution of surface meteoric $^{10}$Be concentrations in the Shackleton Glacier region (a). Where possible, two samples were collected at each location to represent surfaces closest to the glacier, which might have been glaciated during recent glacial periods, and samples furthest from the glacier that are likely to have been exposed during recent glacial periods. Insets of Roberts Massif (b), Bennett Platform (c), and Thanksgiving Valley (d) are included, as these locations serve as the basis for our relative exposure age models. Base maps provided by the Polar Geospatial Center.

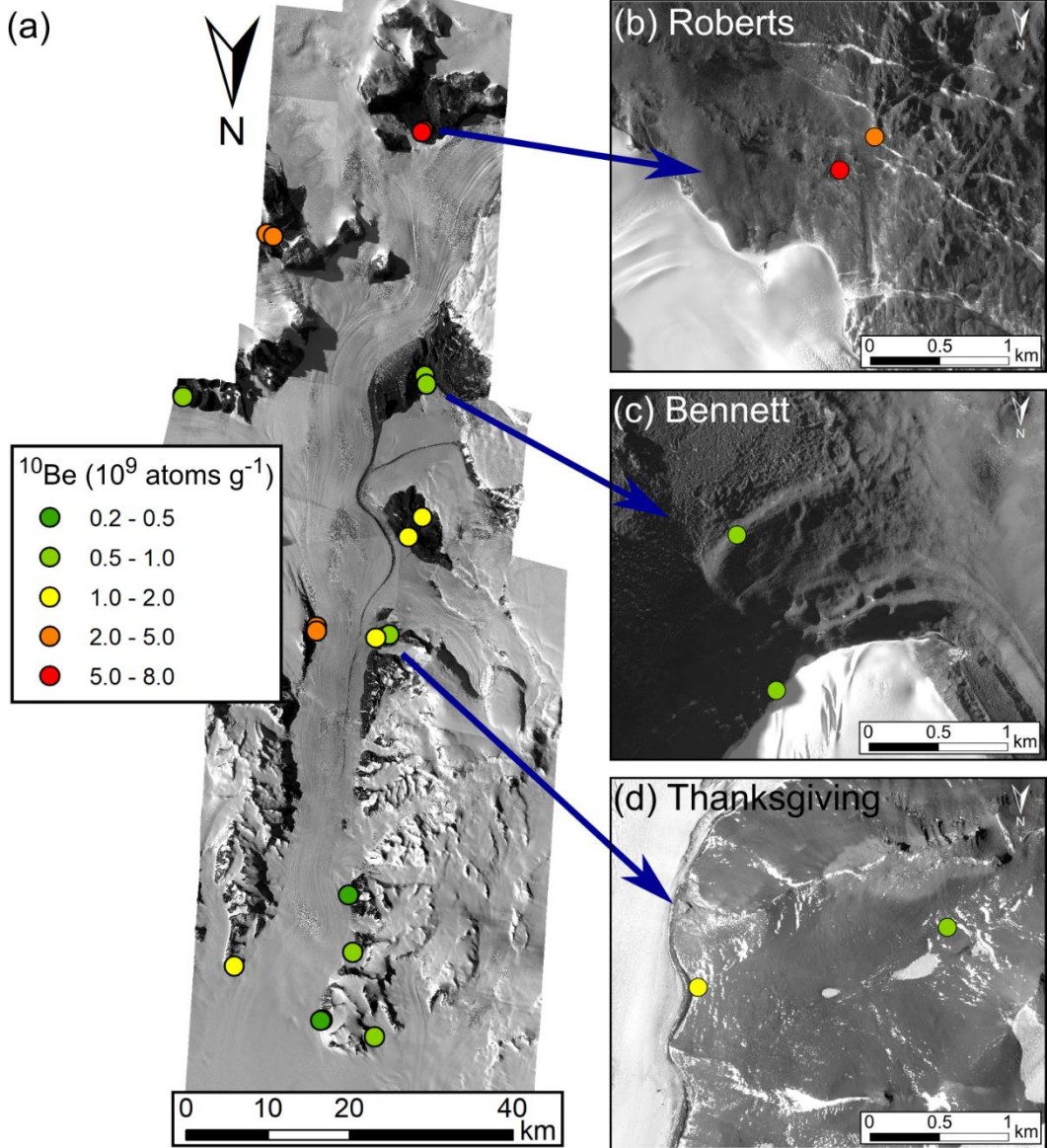

**Figure 6:** Soil profiles of meteoric [10]Be concentrations for Roberts Massif (orange), Bennett Platform (green), and
Thanksgiving Valley (blue) compared to profiles from the Antarctic (Dickinson et al., 2012[*]; Schiller et al., 2009[†]; Valletta
et al., 2015[‡]) and Arctic (Bierman et al., 2014[¶]; Ebert et al., 2012[§]) (a). The [10]Be concentration profiles were also compared
to $NO_3^-$ concentration profiles (b) and a power function was fit to the data (c).

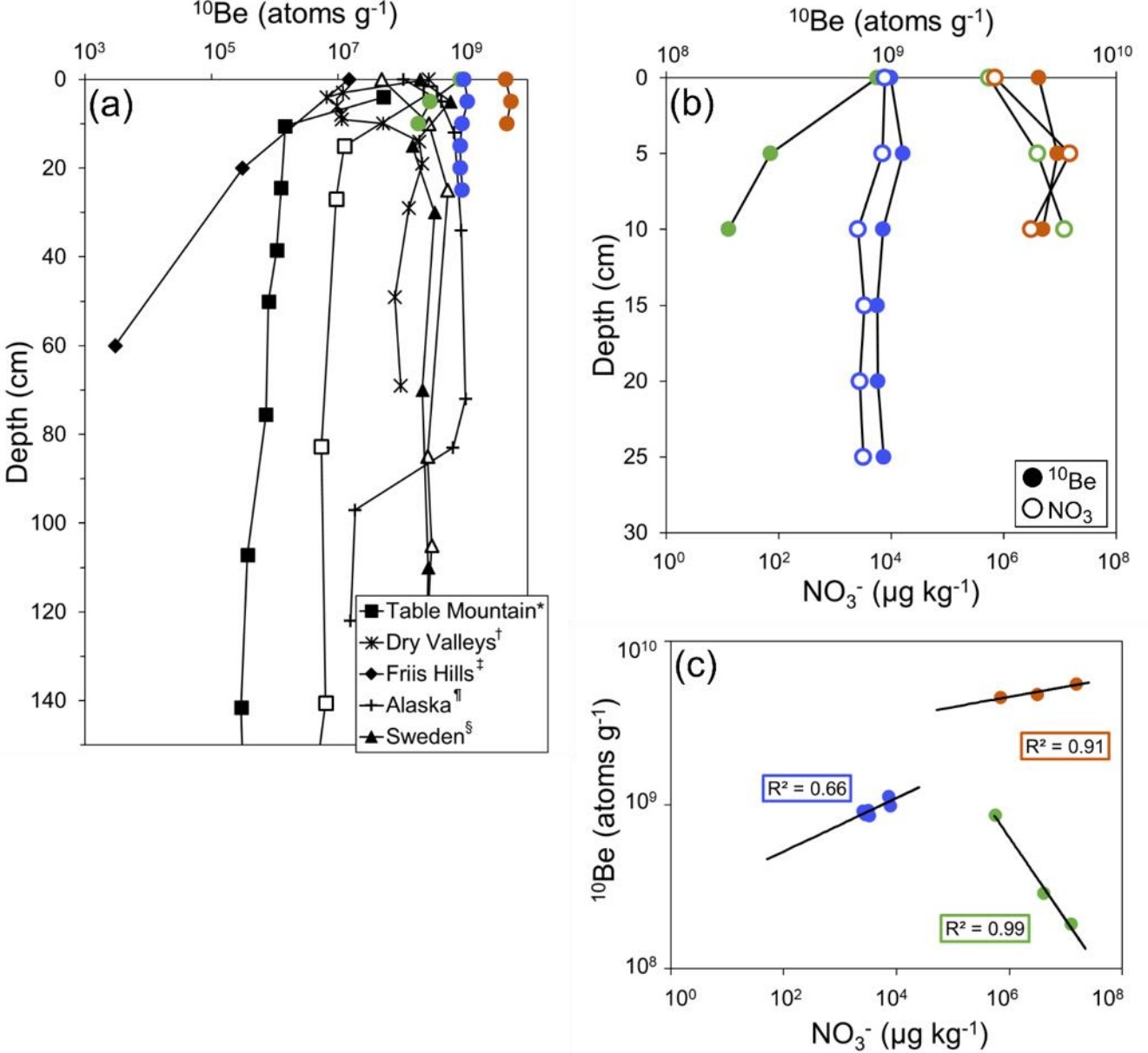

**Figure 7:** Relationship between the measured maximum (or surface) meteoric [10]Be concentration and the calculated
inventory (Eq. 2). This relationship is used to infer [10]Be inventories given a maximum or surface concentration (Graly et al.,
2010). The solid black line is the power relationship between concentration and inventory, while the dashed grey line is the
regression from Graly et al. (2010).

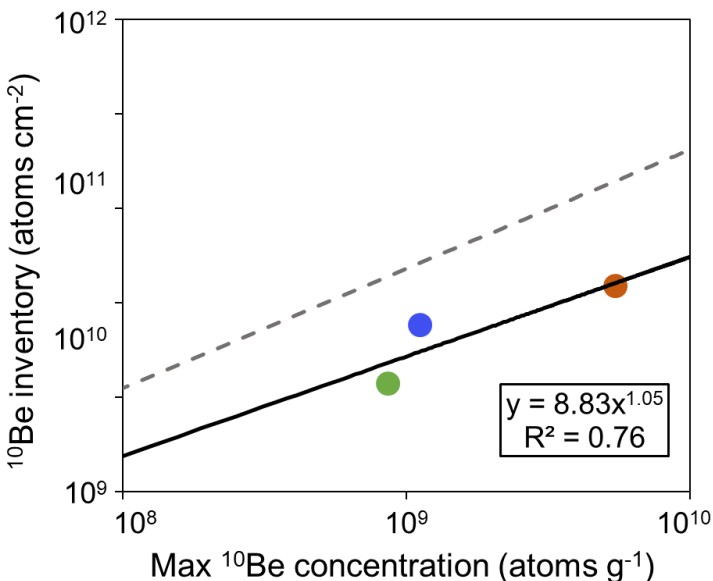



Figure 8: *In-situ* exposure age measurements from glacial erratic boulders (black filled triangles) (http://antarctica.ice-d.org;
Balco, 2020; Balter-Kennedy et al., 2020) in relation to the meteoric [10]Be sample locations from Roberts Massif (a, orange),
Thanksgiving Valley (b, blue), and Mt. Franke (c, grey). Pleistocene-age moraines described by Balter-Kennedy et al. (2020)
are labeled at Roberts Massif in green. We identified moraines (green) of an unknown age at Mt. Franke.

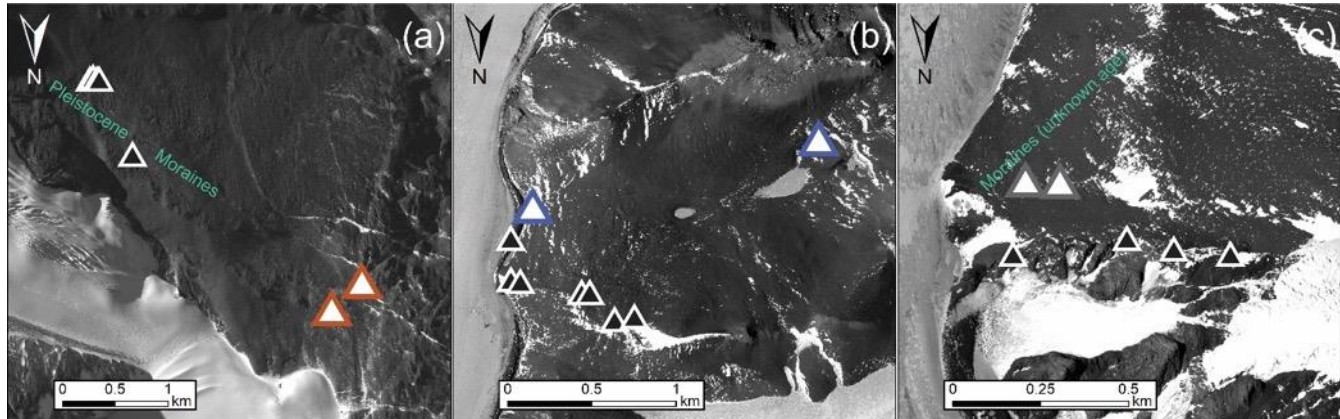


**Figure 9:** Estimated (using $NO_3^-$) meteoric $^{10}Be$ exposure ages (open colored triangles) and inferred (using maximum $^{10}Be$
concentration) exposure ages (closed colored triangles) without erosion compared to *in-situ* ages from ICE-D (Balco, 2020)
and Balter-Kennedy et al. (2020) (solid triangles) against elevation. All *in-situ* ages were measured from glacial erratic
boulders.

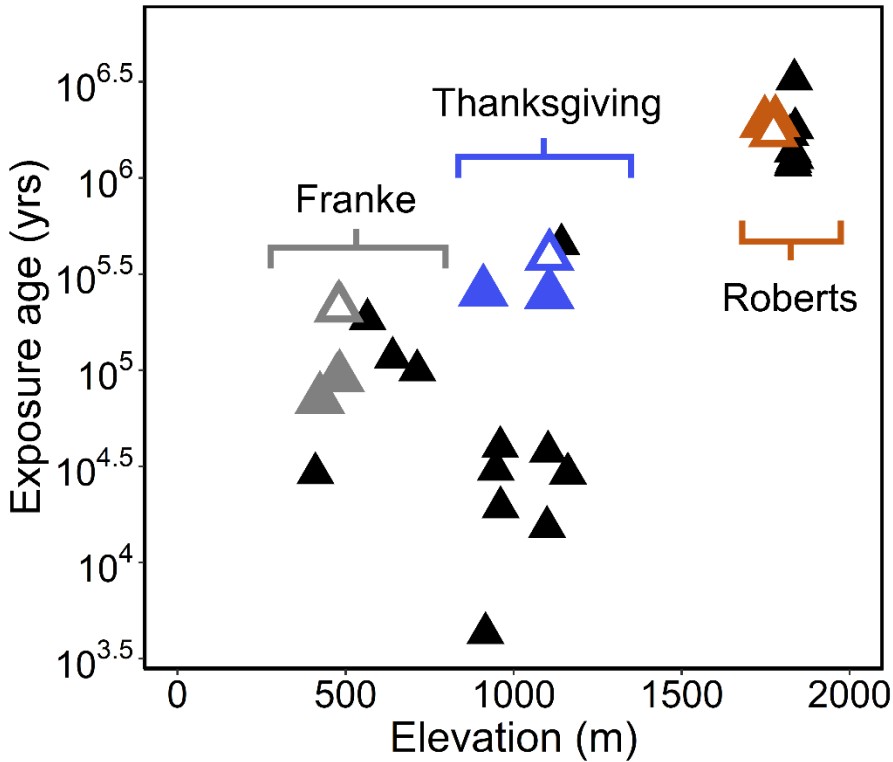


**Figure 10:** Inferred surface exposure ages versus distance from the coast (a) and elevation (b), with (blue) and without
(black) an assumed erosion term. Upward facing triangles are samples collected furthest from the glacier, while downward
triangles are samples collected closest to the glacier. The estimated surface exposure ages using $NO_3^-$ concentrations are
included in panel (c). Values with asterisks (*) are ages calculated using the measured meteoric $^{10}$Be concentrations in depth
profiles.

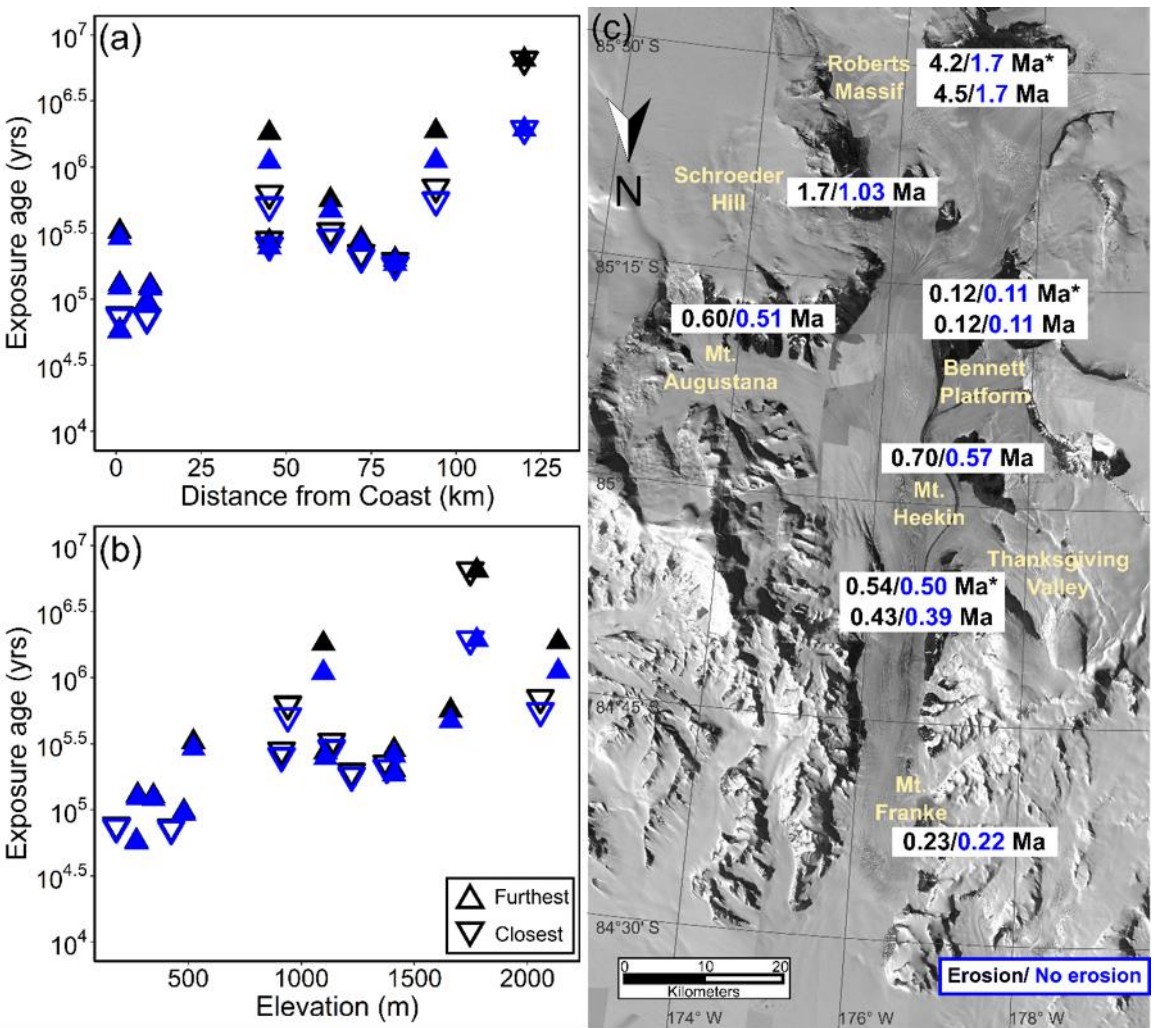


**Tables:**

**Table 1:** Geographic data of samples collected from eleven ice-free areas along the Shackleton Glacier. Distance from the coast (aerial) was measured post-collection using ArcMap 10.3 software. Samples of the format "X-1" are samples collected furthest from the glacier in the transect.

| Location | Sample name | Latitude | Longitude | Elevation (m) | Distance from coast (km) |
|---|---|---|---|---|---|
| Mt. Augustana | AV2-1 | -85.1706 | -174.1338 | 1410 | 72 |
| Mt. Augustana | AV2-8 | -85.1676 | -174.1393 | 1378 | 72 |
| Bennett Platform | BP2-1 | -85.2121 | -177.3576 | 1410 | 82 |
| Bennett Platform | BP2-8 | -85.2024 | -177.3907 | 1222 | 82 |
| Mt. Franke | MF2-1 | -84.6236 | -176.7353 | 480 | 9 |
| Mt. Franke | MF2-4 | -84.6237 | -176.7252 | 424 | 9 |
| Mt. Heekin | MH2-1 | -85.0299 | -177.2405 | 1660 | 63 |
| Mt. Heekin | MH2-8 | -85.0528 | -177.4099 | 1134 | 63 |
| Mt. Speed | MSP2-1 | -84.4819 | -176.5070 | 270 | 0 |
| Mt. Speed | MSP2-4 | -84.4811 | -176.4864 | 181 | 0 |
| Mt. Speed | MSP4-1 | -84.4661 | -177.1224 | 276 | 0 |
| Mt. Wasko | MW4-1 | -84.5600 | -176.8177 | 345 | 10 |
| Nilsen Peak | NP2-5 | -84.6227 | -176.7501 | 670 | 0 |
| Roberts Massif | RM2-1 | -85.4879 | -177.1844 | 1776 | 120 |
| Roberts Massif | RM2-8 | -85.4857 | -177.1549 | 1747 | 120 |
| Schroeder Hill | SH3-2 | -85.3597 | -175.0693 | 2137 | 94 |
| Schroeder Hill | SH3-8 | -85.3569 | -175.1621 | 2057 | 94 |
| Thanksgiving Valley | TGV2-1 | -84.9190 | -177.0603 | 1107 | 45 |
| Thanksgiving Valley | TGV2-8 | -84.9145 | -176.8860 | 912 | 45 |
| Taylor Nunatak | TN3-1 | -84.9227 | -176.1242 | 1097 | 45 |
| Taylor Nunatak | TN3-5 | -84.9182 | -176.1282 | 940 | 45 |

**Table 2:** Surface features of the sample locations from the Shackleton Glacier region.

| Location | Sample name | Sample description |
|---|---|---|
| Mt. Augustana | AV2-1 | Up valley from Gallup Glacier (tributary glacier); at valley floor; surface covered by cobbles and pebbles; red-stained sandstones nearby; frozen ground at bottom of depth profile |
| Mt. Augustana | AV2-8 | At toe of Gallup Glacier; surface covered primarily by boulders; mainly sand between boulders |
| Bennett Platform | BP2-1 | On larger moraine; local depression between two boulder lines, up valley from McGregor Glacier (tributary glacier); at valley floor |
| Bennett Platform | BP2-8 | At toe of McGregor Glacier (tributary glacier); surface covered primarily by boulders; mainly sand between boulders |
| Mt. Franke | MF2-1 | Bottom of wide valley floor; near small moraine; frozen soil at bottom of depth profile |
| Mt. Franke | MF2-4 | Bottom of wide valley floor; near small moraine |
| Mt. Heekin | MH2-1 | On high-elevation saddle; surface covered by sparse small boulders, cobbles, and pebbles; poorly consolidated till; frozen ground at bottom of profile |
| Mt. Heekin | MH2-8 | At toe of Baldwin Glacier (alpine glacier) on valley floor; two ponds nearby; surface covered by loose rocks and sand; poorly consolidated till; possible polygonal surface nearby |
| Mt. Speed | MSP2-1 | Steep slope; large granite boulders; scree |
| Mt. Speed | MSP2-4 | Near cliff by Shackleton Glacier; large granite boulders; scree |
| Mt. Speed | MSP4-1 | Spur on level with glacier; frozen soil near 5 cm depth |
| Mt. Wasko | MW4-1 | Steep slope; large granite boulders; scree; nearby snowpack |
| Nilsen Peak | NP2-5 | On ridge; near large snow patch |
| Roberts Massif | RM2-1 | Near thin moraine; red-stained sandstones nearby with etches; frozen ground at bottom of depth profile |
| Roberts Massif | RM2-8 | Near thin moraine and Sirius Group diamict; large boulders nearby with unconsolidated sediment |
| Schroeder Hill | SH3-2 | Red-stained sandstone; poorly consolidated till; bedrock at bottom of profile |
| Schroeder Hill | SH3-8 | Red-stained sandstone; poorly consolidated till; |
| Thanksgiving Valley | TGV2-1 | Lightly uphill on valley wall; poorly consolidated till; frozen ground at bottom of depth profile; polygonal surface nearby |
| Thanksgiving Valley | TGV2-8 | At the toe of Shackleton Glacier; near thin moraines, surface covered primarily large boulders |
| Taylor Nunatak | TN3-1 | On ridge; surface covered by small boulders with underlaying silt; frozen ground at bottom of depth profile |
| Taylor Nunatak | TN3-5 | Valley floor; nearby snow patches; few glacial erratics; surface covered primarily by small boulders and cobbles with underlaying silt |


**Table 3:** Concentration of meteoric $^{10}$Be in Shackleton Glacier region surface soils and depth profiles from Roberts Massif, Bennett Platform, and Thanksgiving
Valley.

| Sample name | Sample mass (g) | Mass of $^9$Be added (µg)* | AMS Cathode Number | Uncorrected $^{10}$Be/$^9$Be ratio $(10^{-11})$** | Uncorrected $^{10}$Be/$^9$Be ratio uncertainty $(10^{-13})$** | Background-corrected $^{10}$Be/$^9$Be ratio $(10^{-11})$*** | Background-corrected $^{10}$Be/$^9$Be ratio uncertainty $(10^{-13})$*** | $^{10}$Be concentration ($10^9$ atoms g$^{-1}$) | $^{10}$Be concentration uncertainty ($10^7$ atoms g$^{-1}$) |
|---|---|---|---|---|---|---|---|---|---|
| AV2-1 | 0.499 | 394.3 | 151135 | 2.201 | 1.143 | 2.201 | 1.143 | 1.162 | 0.604 |
| AV2-8 | 0.500 | 400.2 | 151137 | 1.786 | 1.067 | 1.785 | 1.067 | 0.955 | 0.571 |
| BP2-1, 0-5 | 0.499 | 401.2 | 151147 | 1.616 | 1.055 | 1.615 | 1.055 | 0.868 | 0.567 |
| BP2-1, 5-10 | 0.499 | 399.2 | 151148 | 0.353 | 0.748 | 0.352 | 0.748 | 0.188 | 0.400 |
| BP2-1, 10-15 | 0.496 | 400.2 | 151149 | 1.573 | 1.894 | 1.573 | 1.894 | 0.848 | 1.021 |
| BP2-8 | 0.498 | 400.2 | 151550 | 0.542 | 0.448 | 0.541 | 0.448 | 0.291 | 0.241 |
| MF2-1 | 0.505 | 398.2 | 151554 | 3.713 | 3.444 | 3.712 | 3.444 | 1.956 | 1.815 |
| MF2-4 | 0.501 | 398.2 | 151555 | 2.448 | 1.395 | 2.447 | 1.396 | 1.300 | 0.741 |
| MH2-1 | 0.498 | 399.2 | 151138 | 0.864 | 0.820 | 0.863 | 0.820 | 0.462 | 0.439 |
| MH2-8 | 0.499 | 395.3 | 151139 | 0.681 | 0.847 | 0.680 | 0.847 | 0.360 | 0.449 |
| MSP2-1 | 0.499 | 403.2 | 151556 | 0.539 | 0.464 | 0.538 | 0.464 | 0.291 | 0.250 |
| MSP2-4 | 0.502 | 402.2 | 151557 | 0.693 | 0.673 | 0.692 | 0.674 | 0.370 | 0.361 |
| MSP4-1 | 0.499 | 400.2 | 151566 | 1.112 | 1.117 | 1.111 | 1.117 | 0.596 | 0.598 |
| MW4-1 | 0.498 | 400.2 | 151564 | 1.093 | 0.662 | 1.092 | 0.662 | 0.586 | 0.356 |
| NP2-5 | 0.496 | 402.2 | 151565 | 2.391 | 1.200 | 2.391 | 1.200 | 1.295 | 0.650 |
| RM2-1, 0-5 | 0.502 | 399.2 | 151558 | 8.541 | 4.116 | 8.541 | 4.116 | 4.538 | 2.187 |
| RM2-1, 5-10 | 0.499 | 398.2 | 151559 | 8.853 | 8.411 | 8.852 | 8.411 | 4.721 | 4.485 |
| RM2-1, 10-15 | 0.500 | 400.2 | 151560 | 13.70 | 8.460 | 13.70 | 8.460 | 7.327 | 4.524 |
| RM2-8 | 0.498 | 401.2 | 151561 | 10.17 | 15.27 | 10.17 | 15.27 | 5.475 | 8.221 |
| SH3-2 | 0.497 | 398.2 | 151551 | 7.191 | 3.129 | 7.190 | 3.129 | 3.850 | 1.675 |
| SH3-8 | 0.501 | 398.2 | 151552 | 4.270 | 3.351 | 4.269 | 3.351 | 2.267 | 1.780 |
| TGV2-1, 0-5 | 0.498 | 398.2 | 151140 | 1.860 | 2.431 | 1.859 | 2.431 | 0.993 | 1.299 |

| | | | | | | | | | |
|---|---|---|---|---|---|---|---|---|---|
| TGV2-1, 5-10 | 0.500 | 398.2 | 151141 | 1.731 | 1.589 | 1.731 | 1.589 | 0.921 | 0.846 |
| TGV2-1, 10-15 | 0.497 | 393.3 | 151142 | 1.635 | 1.377 | 1.634 | 1.377 | 0.864 | 0.728 |
| TGV2-1, 15-20 | 0.502 | 399.2 | 151143 | 1.645 | 1.776 | 1.645 | 1.777 | 0.874 | 0.944 |
| TGV2-1, 20-25 | 0.498 | 403.2 | 151144 | 1.711 | 0.852 | 1.710 | 0.852 | 0.925 | 0.461 |
| TGV2-1, 25-30 | 0.497 | 399.2 | 151145 | 2.148 | 2.071 | 2.147 | 2.071 | 1.152 | 1.112 |
| TGV2-8 | 0.499 | 399.2 | 151146 | 2.106 | 2.185 | 2.105 | 2.185 | 1.125 | 1.168 |
| TN3-1 | 0.500 | 401.2 | 151562 | 7.092 | 5.903 | 7.091 | 5.903 | 3.802 | 3.165 |
| TN3-5 | 0.500 | 401.2 | 151563 | 3.926 | 5.694 | 3.925 | 5.694 | 2.105 | 3.053 |

*$^9Be$ was added through commercial SPEX carrier with a concentration of 1000 μg mL$^{-1}$.

**Isotopic analysis was conducted at PRIME Laboratory; ratios were normalized against standard 07KNSTD3110 with an assumed ratio of 2850 x 10$^{-15}$ (Nishiizumi et al., 2007). Blank $^{10}Be/^9Be$ ratio values averaged 8.152 ± 1.884 x 10$^{-15}$.



Table 4: Exposure ages calculated from Eq. (1-6) and estimated ages using $NO_3^-$ concentration data.

| Location | Measured inventory ($10^{11}$ atoms) | Measured exposure age with $E$ (Ma) | Measured exposure age without $E$ (Ma) | Estimated inventory ($10^{11}$ atoms)* | Estimated exposure age with $E$ (Ma)* | Estimated exposure age without $E$ (Ma)* |
|---|---|---|---|---|---|---|
| Augustana | - | - | - | 0.580 | 0.601 | 0.505 |
| Bennett | 0.135 | 0.115 | 0.106 | 0.143 | 0.122 | 0.113 |
| Franke | - | - | - | 0.268 | 0.232 | 0.217 |
| Heekin | - | - | - | 0.646 | 0.703 | 0.571 |
| Roberts | 1.47 | 4.15 | 1.67 | 1.51 | 4.54 | 1.74 |
| Schroeder | - | - | - | 1.05 | 1.66 | 1.03 |
| Thanksgiving | 0.570 | 0.535 | 0.495 | 0.465 | 0.426 | 0.394 |
| *Estimations derived from linear relationship between NO3- concentration and meteoric 10Be concentration | | | | | | |


**Table 5:** Estimated exposure ages using relationship between maximum $^{10}$Be concentration and inventory in Figure
S1 (Bierman et al., 2014).

| Sample name | Inferred inventory ($10^{11}$ atoms) | Inferred exposure age with $E$ (Ma) | Inferred exposure age without $E$ (Ma) |
|---|---|---|---|
| AV2-1 | 0.38 | 0.285 | 0.258 |
| AV2-8 | 0.33 | 0.224 | 0.207 |
| BP2-1 | 0.31 | 0.200 | 0.186 |
| BP2-8 | 0.31 | 0.195 | 0.181 |
| MF2-1 | 0.21 | 0.097 | 0.094 |
| MF2-4 | 0.18 | 0.074 | 0.072 |
| MH2-1 | 0.59 | 0.565 | 0.469 |
| MH2-8 | 0.42 | 0.328 | 0.292 |
| MSP2-1 | 0.16 | 0.058 | 0.057 |
| MSP2-4 | 0.18 | 0.076 | 0.074 |
| MSP4-1 | 0.24 | 0.129 | 0.123 |
| MW4-1 | 0.24 | 0.127 | 0.121 |
| NP2-5 | 0.42 | 0.326 | 0.291 |
| RM2-1 | 1.24 | >6.5* | 1.93 |
| RM2-8 | 1.50 | >6.5* | 1.94 |
| SH3-2 | 1.07 | 1.87 | 1.11 |
| SH3-8 | 0.67 | 0.702 | 0.560 |
| TGV2-1 | 0.34 | 0.274 | 0.248 |
| TGV2-8 | 0.38 | 0.282 | 0.255 |
| TN3-1 | 1.06 | 1.81 | 1.09 |
| TN3-5 | 0.62 | 0.628 | 0.512 |
| *Outside of model range | | | |

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
