# Peer review of "Relative terrestrial exposure ages inferred from meteoric 10Be and NO3-1"

_Earth Surface Dynamics, 2020_

## Referee Comment (RC1) · Brent Goehring (Referee) · 14 Aug 2020

General Comments

Diaz et al. present a compelling study showing the utility of combining measurements of meteoric 10Be with soluble nitrate as a means to determine surface exposure ages. In this case, they apply their new method to soils adjacent to Shackleton Glacier, Antarctica. However, their new methodology, particularly the combined use of nitrate and 10Be is not well-enough described. Additionally, and as noted below, there needs to be a rigorous uncertainty analysis completed. All that being said, I will very much enjoy seeing this paper published, but for now it needs revision. The methods and results are interesting from an applied sense in that it could be used elsewhere, but their work also adds to the glacial history of the Transantarctic Mountains. Below I present general comments and then further below I present a number of detailed comments and suggest changes.

The one supplementary figure showing the relationship between max 10Be concentration and total 10Be inventory should not be buried in the supplement.

I find that the introduction reads too much like a thesis introduction. All of the content is very good, but I think it could use a bit of streamlining that will help motivate the rest of the paper a bit better, as I think you need to also address the limitations of in situ exposure dating, as you mention later on, but it could benefit from being a bit earlier. Bear in mind this is purely a stylistic opinion can certainly be ignored.

Throughout the manuscript, anywhere there is a reference to an age, rather than a duration, need to use Ma instead of Myr.

There is overall a lack of uncertainty analysis that needs to be completed, particularly exploring the sensitivity of your various age determination models to parameter variance. The measurement uncertainties in this case are tiny compared to other uncertainties. A full error analysis will greatly strengthen the conclusions made in the paper and really needs to be done before publication. A bootstrap approach should be sufficient.

There is far too much framing of the study around Pliocene glacier dynamics, and particularly the Sirius formation. I'd much prefer to see the expansion of the possible newish and important approach that can be implemented combining 10Be with nitrate as a measure of surface exposure duration. Figure 8 demonstrates very nicely a coherent pattern of ice thinning/retreat. This needs to be played up, and the return late in the manuscript to the Sirius Group detracts from the novelness of the work.

Detailed Comments

Line 37: Please provide a citation or two for the first part of the sentence. There is actually quite sparse direct evidence for smaller interglacial extents relative to the Holocene and much is largely inferred from distal evidence or modeling. Additionally, the Ross Embayment is a large area and thus this statement is somewhat vague.

Line 51: How are calculated and estimated exposure ages any different from each other? I know this seems nit-picky, but it is somewhat strange wording as your estimated exposure age had to be calculated first.

Line 62: Unsure what "these studies" are. Are you referring to those cited at the end of the sentence or the sentence prior? If the sentence prior, why do you have a new set of citations?

Section 2.1 Should be worked more into the introduction in my view.

Line 78: Nishiizumi et al., 2007 is not actually a half-life study, an outcome of the standardization is that a different half-life than had been used must be used. Recommend citing: - Korschinek, G., Bergmaier, A., Faestermann, T., Gerstmann, U., Knie, K., Rugel, G., Wallner, A., Dillmann, I., Dollinger, G., Gostomski, C., Gostomski, C., Kossert, K., Maiti, M., Poutivtsev, M., Remmert, A. (2010). A new value for the half-life of 10Be by Heavy-Ion Elastic Recoil Detection and liquid scintillation counting Nuclear Instruments & Methods In Physics Research Section B-Beam Interactions With Materials And Atoms 268(2), 187 - 191. https://dx.doi.org/10.1016/j.nimb.2009.09.020

- Chmeleff, J., Blanckenburg, F., Blanckenburg, F., Kossert, K., Jakob, D. (2010). Determination of the 10Be half-life by multicollector ICP-MS and liquid scintillation counting Nuclear Instruments & Methods In Physics Research Section B-Beam Interactions With Materials And Atoms 268(2), 192 - 199. https://dx.doi.org/10.1016/j.nimb.2009.09.012

Line 101: Given the general absence of anything resembling soils or till in most of Antarctica, one could argue that applying meteoric 10Be is far more spatially limited,

e.g. to regions of the Dry Valley, for example. Thus, I am not sure I would argue for your method by arguing that in situ exposure dating is limited, but instead argue that they are complementary.

Starting line 107: I am not sure the bedrock lithology is all that relevant. I understand you want to show the protolith for weathering products, but I think it could be said more concisely. I think the geologic setting paragraphs could be combined.

Line 123: Suggest changing "glacial dynamics" to "glaciers"

Line 128: By two samples, do you mean two surface samples? Suggest clarifying the text here, especially since you have depth profiles samples from elsewhere.

Line 130: In your reference to sample distance from the glacier, are you largely referring to further away as controlled by elevation, or by horizontal distance? I think some clarification of this could be useful, as depending on the valley geometry, changes in ice thickness might not be significantly further away from the glacier, or vice versa. It might be more constructive and more generalizable to perhaps say that two samples were collected, one adjacent to the glacier, characteristic of times similar to the current extent and one further away representative of significant changes in glacier size (larger). A useful column in your table and the way most Antarctic glacier change is expressed is as change in ice thickness.

Line 142: Why not report the fraction between 2mm and 425 microns? Was none present? Sand usually extends to 2 mm.

Line 170: Suggest not starting paragraph with "However...." I suggest that when laying out your calculation methods, that the equations flow more within the paragraph, rather than being at the end of each paragraph. I found it somewhat hard to ready.

Line 179: Suggest adding "any" before "have meteoric"

Line 197: Delete "which"

Line 202: Confused because didn't you calculate two samples from every location, only profiles from only a few?

Line 206: The lack of an expected concentration based on regressions against distance and elevation might just be spurious and making predictions from these regressions very tenuous. I suggest removing this sentence.

Line 222: The ages are not necessarily minimum ages, as while you may be overcorrecting for inheritance because you don't know the background inventory, you also do not a priori know the erosion rates of the soils, even though you make assumptions. I suggest that rather than couching the ages as minimum, as they are only minimum relative to your max limiting no inheritance ages, you just present them as best estimate given knowledge of the parameters.

Section 5.3.1 This section is very confusing in terms of what you did and is not represented in the methods at all, thus the results presented here come out of nowhere. There needs to be a clearer explanation of what was done. I think the approach is really neat and valuable, but right now it just isn't explained well-enough. I am also very confused upon the first and second read as to what was done with what profile, as the second paragraph mixes results from sites with both measurements and sites without.

Section 5.3.2 Like the prior section, where there are a number of inferred methodological requirements, more expansion of the discussion is needed to aid the reader that may only have casual knowledge of meteoric 10Be knowledge as I can see many readers being most interested in the inferred ice history. I think one thing that will help immensely is that this and the prior section are more traditionally considered as part of the discussion and the results purely your 10Be and NO3- measurements. Now, if you were to present the calculation methods using nitrate and the inventory vs max concentration analyses in the methods, then you could keep in the results. At present, there is just a bit too much mixing and overall not enough time dedicated to these important sections that you then use extensively in the discussion below. Also, best I

can tell Figure 8 does not show the relationship between max concentration and total inventory, please investigate, or do you mean to only present the max exposure ages.

Line 247: Please elaborate or define what the model limits are, as this is not defined. Presumably just the influence of the time scale to 10Be saturation given an erosion rate. I also wish there were different terminologies used with regards to calculated vs estimated. Perhaps refer to one as the apparent max limiting age and the other a model age?

Line 260: The correspondence with in situ ages is quite remarkable. What is lacking though is a clear representation of the two different data sets. This is why I suggested that perhaps you determine the elevation above modern ice surface and thus you can then make age vs elevation plots for your data and the in situ data. I think will drive home much more clearly the correspondence. Or you could consider maps showing the various bits of data, but I think they will get very busy very quickly. While the correspondence in many scenarios is striking, one thing to consider and make sure you make clear is whether the in situ data are from bedrock or from erratics, as they will have quite different exposure ages and thus your soil ages might always be older than nearby in situ erratic exposure ages. The fact that your meteoric ages, including nitrate corrected, agree so much with in situ erratic ages suggests some mechanism for resetting and flushing of 10Be or that your model is determining the pre-LGM inherited concentration quite clearly. I think this needs further discussion and is important to highlight more.

Line 272: Need a reference for exposure dating results from Beardmore Glacier.

Line 276: Unclear if you are referring to your ages or in situ exposure ages. Please clarify.

Line 280: Need to insert "to" after "first"

Line 288: The arguments about the suitability seem out of place and kind of come

out of nowhere and seem to set up a strawman for no apparent reason. I suggest removing and focusing on the apparent success of the nitrate correction given the good agreement with in situ exposure dating. Starting line 292: The first few sentences of this paragraph read too much like a conclusions section. Suggest revision.

Line 303: As mentioned above, the nitrate regression models needs further description and elaboration, particularly since this really is the first major combined use of these two measures. Line 306: Wouldn't a lack of correlation be expected given the exponential fall off of a 10Be profiles, so that below a certain depth there will be little to no variance in the 10Be concentration and presumably the same in nitrate?

Line 313: Missing "was"

Line 327: The referencing choices are confusing. Are you referring to the start timing of the last glacial cycle and thus referring the Blunier and Brook and the other refs?

Line 340: Unclear as to which exposure age you are referring to. Bennett Platform?

Line 352: Suggest rather than saying delayed response that you more generalize it and just say different response from Ross Ice Shelf confluent outlet glaciers, or something to that effect.

Line 358: This conclusion is spot on and is a major finding of the paper, however its use, the details, etc. are not elaborated on enough earlier in the manuscript.

Line 365: The broader question then becomes, how do we differentiate between a site with inherited meteoric 10Be that was covered by LGM ice from a site that was never covered during the LGM and more recent glaciations. This is a question that the in situ community has struggled with. We are only starting to get clarity from a focus on erratic exposure dating with long-lived nuclides or application of in situ 14C to erratics and bedrock. Recent work in the Weddell Embayment with very old erratic and bedrock in situ ages were clearly covered by LGM ice as shown by in situ 14C, including preservation of delicate features like moraines (e.g., Nichols et al., 2019). Thus, during

a say 10 kyr long ice cover period, how much of a reduction in the meteoric 10Be signal can be expected? What about reduction in nitrate? Presumably unless the ice is wet based, neither will be mobilized and then you need the correct pH conditions. These thoughts are briefly touched on, but the manuscript could use a bit more elaboration on the long-term interpretation of the signal recorded by your methods and what its implications are for interpreting surface processes in Antarctica. Thus, it could be useful to elaborate on the presence of polythermal moraines, why are some areas reset for the meteoric and in situ methods.

Figure 1: Not sure if this is supposed to be this way of if some strange PDF artifact, but the exposed rock areas are banded. I also think you could make the overview map larger scale to give readers a better context of the Shackleton Glacier.

Figure 3: A similar figure thinking about the fate of nitrate during ice cover would be informative.

Figure 4: Add panel labels please. Also, it is confusing that in the Shackleton glacier map, the coloring represents concentration, but you then use the same colors for the different sites, or is it only the arrows? This is somewhat confusing, and I suggest not using colored arrows that are the same as the color scaled points for concentration. Here the figure is trying to show too much.

Figure 5: This figure and all figures. Are uncertainties shown, but smaller than the symbol? Please note this or add uncertainties if need be.

Figure 6: Suggest removing the lines connecting the points, as it implies that there is a trend in grain size % between the points. The measurements are point measurements.

Figure 7c: Please provide equations for the fits along with uncertainties on the fit parameters. These uncertainties then need to be used for error analysis on the resulting ages.

Table 2: I suggest presenting uncertainties using the same exponent for the measured

value and uncertainty.

---

## Referee Comment (RC2) · Anonymous Referee #2 · 15 Aug 2020

**I. Summary.**

The summary of this review is that the data collected in this paper are useful, interesting, and valuable to publish. In general, the idea that accumulation of atmospheric constituents in Antarctic soils is useful for estimating soil ages and residence times is important from many perspectives, including glacier change, paleoclimate, and biology, and this paper contains a lot of data that are relevant to this topic. However, I don't think the paper is ready for publication at the moment, because many sections of the paper are incomplete, have a weak relationship to what I think are the important points of the paper, or were written in too simplified or simplistic a way. Perhaps some of the

oversimplifications are only a consequence of the practice of charging open access publication fees on a per-page basis, but they are a serious problem for this paper. At the moment, this paper contains interesting and useful observations, but is not in a condition that will lead readers to understand this.

As will be immediately evident, I spent a lot of time reviewing this paper and looking at the data, again because I think the data are worthwhile and I'd like to see them published. In fact, this review may be longer than the paper. Thus, I hope the authors take this in a positive way as an effort to help the paper live up to its potential. All the issues I've noted below can be fixed – although fixing some of them will require abandoning some major parts of the paper as written – to make a good paper, and I hope the authors will do this.

One problem with this review is that my concerns with the paper mostly relate to fairly large-scale aspects of the paper organization and data analysis, and cannot be addressed with a few line edits. They will require some reorganization of the paper. Thus, the organization of this review is that I have covered what I think are the major issues in several sections at the beginning, and then at the end suggested an improved organization for the paper. A few minor comments are also added at the end.

**II. Overall motivation of paper.**

The overall motivation of this paper as it is written now is that the measurements are presented as having been made for the purpose of characterizing glacier change in the Shackleton Glacier area. This was a surprise to me, because most of the authors are associated with ecological and microbiological research in Antarctica and I had some recollection of hearing about this project from the authors at scientific meetings. Thus, I looked up the funding source for this project and found that my recollection was correct: the project is not focused on glacier change per se, but instead on learning about the relation between microbial ecology and soil exposure duration by investigating biological communities in soils that were and were not covered by ice during the

last glacial maximum. As I understand it, this is extremely interesting: soils that are exposed for extremely long periods of time in the TAM without disturbance build up high concentrations of salts which limit the survival of microbial communities, but then on the other hand, recently disturbed soils with low salt concentrations that are more habitable were recently covered by ice. I am sure I am oversimplifying this, but the facts that the microbial communities can't survive in the old soils, and might not survive glaciation either, leads to a compelling mystery about how they recolonize and move around as the ice advances and retreats.

This research question is really interesting. In my view it is much more compelling than the motivation given in the paper to learn about glacier changes in a few places in the Transantarctic Mountains. A lot is already known about that from previous research, and there have been several other projects that are in progress or recently completed that were specifically designed to learn about past glacier change in the Shackleton Glacier area in much more detail than is possible for this study. Learning about glacier change is certainly important, but fairly routine. The refugia-and-recolonization question is much more interesting and exciting. However, it is not mentioned at all in this paper.

From the perspective of this paper, this is important for two reasons.

*II.1. The way the paper is motivated makes the experimental design look bad when, in fact, it is not.*

The experimental design of this study is very well designed from the perspective of a biological survey. The use of atmospheric fallout constituents of soils to rapidly get an approximate idea of the soil age, and distinguish soils that were ice-covered during the LGM from soils that have not been ice-covered for millions of years, is a smart, well-designed approach that is likely to be effective for its intended purpose. On the other hand, the study is not well designed for the purpose of reconstructing past glacier change. A well-designed study aimed at quantifying glacier change in some

ice-marginal area from exposure-age measurements of some sort would involve geomorphic mapping of glacial deposits and determination of their relative age, followed by collection of a large number of exposure-age samples from a range of stratigraphically ordered glacial deposits in each ice-marginal area, including replicate sampling of each landform to test for inheritance and recycling effects. A good example of such a study is the Balter paper that focuses on Roberts Massif and is cited here – that study involved only one of the ice-free areas discussed in the present paper, but included extensive mapping followed by several hundred exposure age measurements, including many replicates from each landform as well as sampling from landforms with a known relative age relationship, providing many opportunities to test the assumptions of their methods. The present paper does not include any of these elements. There are also similar examples from the southern TAM, including research by Bromley and Todd at sites slightly farther away near Scott and Reedy Glaciers.

The point here is that if the present study was motivated by the original objectives of collecting geological information needed to study ecosystem succession, it would be perceived by readers as well-conceived and well-designed. If motivated as a study of glacier change as in this paper, on the other hand, the experimental design appears weak and inadequate by comparison to other studies. By extension, it causes the conclusions of the paper to appear to be based on substandard data.

I very strongly urge the authors to change this emphasis. They should clearly explain the purpose of the overall project that led them to the experimental design used here. It is true that the data collected for this purpose also have value in quantifying glacier change, so there is nothing wrong with focusing additional discussion on that later in the paper, but motivating the entire paper from this perspective makes the paper much weaker than it should be.

*II.2. The way the paper is motivated leads the paper off into vague theories that can't be addressed by the data.*

The second reason that motivating the paper as a glacier change study has negative consequences for the paper is that it leads the authors to a number of broader motivations and conclusions that are not well addressed by the data that are actually presented here. Specifically, the authors spend a lot of time in the introduction discussing the issue of East Antarctic Ice Sheet "stability." However, as written, this discussion is very vague and it is not clearly related to the actual observations. In general, if you have some broad, continental-scale hypothesis and you want to test it with a few local observations, you need some sort of clear, quantitative prediction that follows from the big hypothesis and can be tested or falsified with your observations. This sort of connection is not present in the paper. This issue does not affect my conclusion that the observations in the paper are valuable and should be published, but I think setting up the paper in this way is a poor decision by the authors that makes the paper look less valuable than it really is. In reading this paper it almost seemed that the authors did not think that their own observations were valuable or interesting, so they felt obligated to add a number of unrelated things that sounded more important. Unfortunately, this had the opposite effect for me: I felt like the authors wrote a whole bunch of checks in the introduction that they had no way to eventually cash. In the next paragraphs I'll specifically highlight the sections that I thought were overbroad and acted to reduce, rather than increase, the impact of the paper. I encourage the authors to remove all of these sections.

The most problematic part of the paper from this perspective is the first two paragraphs of the introduction (lines 33-45) and section 2.1 ("Stability of the EAIS"), lines 55-76. The introduction discusses the fact that the Antarctic ice sheets are proposed to have been a lot smaller during some warm periods in the past. While it is certainly true that this has been hypothesized and that in a very general sense this is a strong motivation for studying past changes in the size of the Antarctic ice sheets, there is almost no connection between this overall idea and the specific observations described in this paper. As discussed above, if this is the motivation for the work, the work looks inadequate.

Section 2.1 is much more problematic. First, the use of "dynamic" in this section refers to a dispute in the Antarctic ice sheet change literature during the 1990's having to do with whether or not the EAIS collapsed during the Pliocene, which makes this section quite difficult to understand for readers who are not already familiar with the 1990's literature. It would be clearer to simply state that it is not yet known whether or not the East Antarctic Ice Sheet was significantly smaller during past warm climates. The second problem in this section has to do with confusion between ice sheet change and climate change. The references in this section include both model simulations that show that the EAIS could have been smaller during warm periods and also observational studies arguing that deposits in the TAM require uninterrupted polar desert conditions since the Miocene. These two things are not comparable: the presence or absence of polar desert conditions is not a proxy for ice sheet size. The discussion of how long polar desert conditions have prevailed in the TAM is important in this paper because it gives context for one potential application of salt deposition in soils, i.e. the idea of a "wetting age" in which the amount of salt that has accumulated can give information on when liquid water was last present. However, this important implication of the idea is not at all mentioned here.

To me the overall effect of this section was mainly to confuse things by introducing a vague digest of older literature without clear indications as to how it is relevant to the paper. There is no pathway for the reader to relate this background information to the study. The reader is left to think that certainly what happened to the EAIS during past warm periods is important, but more thinking will lead to the observation that there are already several thousand exposure-age measurements from around Antarctica that have not answered this question. How are the additional handful of measurements here, which are mostly more complex to interpret than the existing data because they are meteoric rather than in-situ-produced Be-10, going to help?

To summarize, it seems to me that setting up very broad questions but giving the reader no pathway for how the authors' observations are going to help answer them

**ESurfD**
nearly ensures that the reader will be disappointed. In fact, when the paper gets to the conclusions, the reader is disappointed, or at least I was. Specifically, the section on line 321 "Our data support....that EAIS was not synchronous.." is particularly disappointing from this perspective. To begin with, nowhere have the authors defined "synchronous." Synchronous with what? What is the relation between "stable" and "dynamic" in the introduction and "synchronous" here? What would the observations in the paper look like if they were not "synchronous"? Again, there is no pathway for the reader to understand exactly what "synchronous" means, and how the observations here could distinguish it from the alternative (which is also not defined).

**III. Oversimplified explanation of atmospherically produced Be-10.**

This part of the review mostly focuses on section 2.2 and 4.3, which explain how meteoric Be-10 can be used to estimate the exposure age of a soil. Although nothing in these sections is specifically incorrect, this part of the text is hard to understand and in some areas is oversimplified, which I think later leads the authors into oversimplified or weak conclusions.

With regard to section 2.2, the main thing the authors need to get across here is that meteoric Be-10 builds up in soils, so the total amount of Be-10 present in a soil profile is related to the age of the soil. This information is here, but it is missing some important context and mixed up with other confusing things. For example, it is not true that the purpose of measuring meteoric instead of in-situ-produced Be-10 is because quartz is absent. In all the example studies given here, plenty of quartz was present. So that is very confusing to readers. This section is also missing two critical points. One, the authors should clearly state that meteoric Be-10 is mobile in the soil, so it is not the concentration at any particular location that is proportional to the exposure age, but instead the total inventory in the entire soil profile. Two, the behaviour of meteoric Be-10 and salts in soils may be quite different, for example because Be-10 remains bound to particles even when the soil is wet, whereas salts are mostly mobile in water.

The other important area here that needs to be either here or in the section on study sites is a discussion of exactly what landforms were sampled and how that relates to meteoric Be-10 systematics. At most of these sites one could sample from either a constructional landform deposited in an ice sheet advance, typically some kind of a moraine, or from a surface between moraines. Sediment in a moraine could be subglacially derived, would most likely have been emplaced all at once or in a short period, and could therefore be emplaced with a fairly low bulk Be-10 concentration. In this case the Be-10 concentration measured now would likely reflect the age of the moraine. Inter-moraine surfaces, on the other hand, may have been covered and un-covered by ice repeatedly, perhaps with gradual addition of small amounts of sediment each time. In this case the bulk Be-10 inventory would be unlikely to reflect the amount of time since the site was most recently uncovered, but instead the total amount of time spent ice-free during a long succession of glacial-interglacial cycles. Interpreting Be-10 data from these two sorts of sites might be quite different.

Section 4.3 is about how to quantitatively interpret Be-10 concentrations as an exposure age of the soil. This section would benefit from several improvements. Specifically,

Equation (1) seems to be missing important elements. This equation is intended to indicate that the Be-10 concentration at the soil surface increases due to deposition ($Q$), and decreases because of radioactive decay ($\lambda N$) and erosion. However, it is missing the equally important process of downward transport of meteoric Be-10 into the soil. Depending on the process that is moving Be-10 around in the soil, this could be quite complicated, but if you think of it as diffusion it would be a partial differential equation looking something like this:

$$\frac{dN}{dt} = Q - \lambda N - E\frac{dN}{dz} + D\frac{d^2N}{dz^2} \tag{1}$$

A complete solution for this can be potentially very complicated, especially as diffusion is probably not isotropic and also variable with depth in the soil profile. Thus, a com-

**ESurfD**
plete advection-diffusion equation is not generally used in this application. Regardless, Equation (1) is at least incomplete and also confusing for the reader, because as it is written it does not include any process that can move Be-10 below the immediate surface layer.

A common approach in the meteoric Be-10 literature to simplify this relationship and make it more useful is to write the governing equation for the soil inventory $I$ (atoms per cm2, vertically integrated) instead of the concentration, like:

$$\frac{dI}{dt} = Q - \lambda I - EN_s \qquad (2)$$

where $N_s$ is the surface concentration (atoms/g) and E is the erosion rate in mass per area units. This representation also highlights the fundamental problem in interpreting single measurements of the surface concentration: relating the inventory (which is the quantity that scales monotonically with the exposure age) to the surface concentration (which may or may not always be proportional to the exposure age). Using this equation instead of Equation (1) would make this paper much clearer. Alternatively, this paper could simply refer to other literature that describes meteoric Be-10 systematics in detail – it is not necessary to reinvent the wheel here.

Finally, an important point for these sites is that it is not even clear that erosion is taking place throughout the ice-free areas at all. In flat areas covered by unconsolidated glacial diamicts, after deflation of fine-grained material takes place (which is probably shortly after deposition) and leaves a bouldery lag covering the surface, there are really not any processes that can cause erosion. Perhaps the only process that can bring new sediment to the surface and permit deflation would be periglacial disturbance of the soil. This issue reminds me that an important thing that needs to be added to section 3 is some discussion of the surface characteristics of each site, including presence or absence of boulder pavements and periglacial features like cracks and polygons, because these features are relevant to interpreting the Be-10 data. In

addition, if the authors observed inflationary silt layers beneath gravel pavements at any of these sites, they should make note of it – interpreting Be-10 concentrations from an inflationary layer would be quite different from an eroding matrix. In any case, observations of long-term soil erosion in the Dry Valleys are mostly on hillslopes, and there is some evidence that flat, valley-bottom areas in the lee of glacier tongues are actually sediment sinks, where fine-grained sediment removed by wind deflation from hillslopes and surrounding rock areas can accumulate. Thus, it is quite possible that the Be-10 inventory at these sites is increasing due to fine sediment deposition, not decreasing due to surface erosion. The overall point of this section is that it is not at all clear to me that erosion should even be included in the relationship between inventory and age for these sites. For this paper, I think it might make the most sense to simply relate inventory to exposure age by $dI/dt = Q - \lambda I$, i.e. disregarding erosion and deposition, and accept that this approach might be either under- or over-estimating exposure ages.

In my view, the important things that need to be in this section, some of which are already here in part but in an incomplete way, are as follows.

First, this section has to define what the inventory is. That is Equation (2) as written, but it would be much simpler to just write that in integral form.

Second, this section has to relate the inventory to the exposure age. This needs to be accompanied by a discussion of the field evidence for or against the existence of erosional or depositional processes at the sites. The model for relating age to inventory has to be based on the physical observations of the processes that are happening at these sites.

Third, this section has to clearly explain how one measures the Be-10 inventory. As already discussed in the paper, this can be done in two ways, either by measuring a complete depth profile and integrating, or using an empirical relation between surface concentration and inventory as in the Graly paper.

An additional problem with this section is that "inheritance" is not clearly defined, which

is confusing. There are two possible interpretations of "inheritance" here which are quite different. First, if the site is a constructional landform that was deposited at a particular time by a glacier advance, then "inheritance" is the Be-10 that was present in the glacial sediment at the time of deposition. In this case, one would expect it to be constant with depth, and it would be the same thing as the "background" Be-10 discussed later. The second possibility applies to the situation where the same soil surface is repeatedly exposed and covered by ice during multiple glacial-interglacial cycles. In this case, the "inheritance" is not well defined. Is it the Be-10 that was there before the last period of ice cover? This would not be expected to be constant with depth, so it would not be the same as "background" and it might be impossible to distinguish from Be-10 that was deposited after the last ice retreat. Alternatively, it could be Be-10 that was deposited in the soil parent material at some long distant past time before all the ice cover events, which might be the same as "background." A final possibility, as discussed above, is that some of the soils may be inflationary, and then "inheritance" would be the Be-10 concentration in silt at the time it was added to the soil column. The point is that it is important to clearly define what "inheritance" means and whether it is or is not the same thing as "background".

Finally, a clear definition of "background" in the context of a depth profile is needed here. The basic concept (that the concentration is supposed to decrease with depth until you reach a depth where the concentration becomes invariant with depth) is correctly described near line 182, but what is missing is a clear statement of how one knows that one has observed this. One cannot say that it is possible to estimate the background from a depth profile unless the depth profile shows two things: a decrease in concentration between the near-surface and deep parts of the profile, and then at least two samples at the bottom of the profile that show the same concentration. In this paper, these two criteria for whether or not the background can be estimated leads to the problem that none of the depth profiles in this paper satisfy the criteria, so it is not possible to say that the background concentration has been measured for any of these profiles (also see discussion of this below). Overall, what I suggest doing here is

noting that in principle the depth profile method is one possible way to estimate $I$, but it can't be used in this application because insufficient data were collected – and then move on to discussing the approach of using an empirical correlation between $N$ and $I$ to estimate $I$.

To summarize, in my opinion these sections of the paper dealing with relating Be-10 concentrations to age need to be thoroughly revised to make these five points, clearly, in order.

**IV. Data analysis.**

This section of the review focuses mostly on section 5 and highlights three areas that I thought were incomplete or oversimplified and need improvement.

The first one of these is the section in lines 200-209 that deals with the regressions in Figure 5. I did not understand what the purpose of these regressions is. It seems that the basic sample collection design was to go to each ice-free area and collect a pair of samples, one from a site thought to have been covered by LGM ice, and a second from a more ice-distal area that was probably not covered. Existing exposure-age data show that many areas in the TAM that are outside the LGM limit have been ice-free for hundreds of thousands to millions of years. As Be-10 concentrations should be proportional to exposure age, this implies that we should expect to see order-of-magnitude differences in surface exposure ages between pairs of samples collected from the same area – potentially within meters of each other if a pair was collected just inside and outside the LGM ice limit. In this context, I don't understand why one would want to regress Be-10 concentrations against elevation and distance from the ice shelf. Is the goal here to identify differences in fallout flux with elevation? If so, that effect would be expected to be orders of magnitude smaller than the localized variation in concentration attributable to soil age, and asking this question would not make any sense unless one could identify soils at a range of elevations that were independently known to have exactly the same exposure age.

Because I don't see any basic physical relationship that would support linear regression of concentration against elevation/distance, as a reader I am left with the impression that the authors simply felt that there should be some linear regressions in the paper. I am not sure this is the impression that the authors want to give the reader. It makes the paper seem weak and confused, and I urge them to remove this section of the paper.

The second area that seems problematic to me in this section of the paper is how the authors approach estimating the Be-10 inventories in section 5.2. As discussed above, none of the depth profiles collected in this paper allow a background concentration to be estimated. One profile decreases, but does not at any time stop decreasing and become constant with depth. The others start out at a high concentration and do not decrease in the interval sampled. None of these data meet criteria for identifying a background concentration. Thus, it is not possible to use the depth profile method for estimating postdepositional Be-10 inventories for any of the sites in this paper (although of course one could assume zero background and add up all the measurements in the depth profile to compute a minimum limit on the total inventory, which may be useful to compare with the empirical $N$-$I$ transfer function). The implication of this is that section 5.2, as well as any age estimates based on a background subtraction, are not valid and should be removed.

What I suggest doing here is removing section 5.2, noting that the depth profile data do not allow estimating $I$ accurately, and rely entirely on the empirical-correlation-between-I-and-N approach for estimating $I$, which is already clearly covered in section 5.3.2. This is not really a major substantive change to the paper, because at most of the sites there are only surface data in any case.

The third area that I think needs additional discussion in this section is the discussion of the relation between Be-10 and nitrate concentrations. Both are atmospheric constituents that are deposited in soils and accumulate over time. Thus, at first order one would expect them to be positively correlated. However, in one depth profile, they are inversely correlated (Fig. 7, lower right). Of course the reason for this is that

**ESurfD**
one doesn't necessarily expect the two measurements to be correlated within a depth profile because the mobility of the two species might be different, but without further discussion here, a statement such as "we used the relationship between Be-10 and NO3 to estimate Be-10 concentrations..." (line 229) cannot make any sense. Does the relationship that is referred to have a positive or a negative slope? To summarize, this section needs to be made much more clear so that the reader can understand when concentrations, surface concentrations, and inventories are being discussed, and what differences in behaviour of Be and NO3 could lead to positive or negative correlation. This may require making this section substantially longer in order to explain the reasoning step by step so that the reader can follow it.

**V. Discussion and interpretation areas.**

As I read through the discussion and conclusions sections of this paper, my overall impression was that they contain a lot of statements that may well be true, but are not clearly related to the observations in the paper. This is a serious problem, because to the reader this makes it look like the discussion section is focused on a series of unsupported claims. This distracts attention from the important observational data and makes the paper look weaker than it should. I strongly encourage the authors to significantly revise this section to clearly link the observations to the conclusions, and make this section as long as it needs to be so that there is a clear chain of reasoning behind each of the conclusions. As it is, too many steps are skipped and it is not possible for readers to understand how the authors got to their conclusions.

The first aspect of the discussion that needs additional work is that the most basic prediction of the experimental design is that, first, Be-10 inventories and/or concentrations should increase with distance from the ice margin at each site, and, second, Be-10 inventories/concentrations for the ice-proximal samples that are supposed to have been exposed after the LGM should have magnitudes that are appropriate to post-LGM exposure, i.e. 10-15,000 years of surface exposure. As discussed earlier in this review, an initial problem here is that the authors have not clearly explained that at most of the

study sites, they attempted to sample LGM and non-LGM-aged surfaces. This needs to be clearly explained earlier in the sampling section. Regardless, the discussion section should lead off with a clear explanation of whether these basic predictions are or are not satisfied. I would do this with a figure for each site showing distance from the nearest ice margin on the x-axis, and Be-10 and NO3 concentrations on the y-axis. I think the reader first needs to know if this basic concept works at all if they are to believe any of the additional conclusions later.

The second aspect of the discussion that is incomplete/too abbreviated is the section beginning on line 260 that compares the results to existing exposure-age data from glacially transported boulders. Personally, what I would view as minimally adequate here is a map view of each site where there are existing/published exposure age data, showing the location of the soil pits described here, the location of any moraines or drift boundaries including any hypothesized LGM ice limit, and also the location of the independent exposure-age data, which will be mostly boulders dated by some in-situ-produced nuclide. Alternatively, instead of maps, these could take the form of plots with distance from the ice margin on the x-axis, and exposure ages calculated from the various data on the y-axis. Because there are only three sites where both types of data exist, this shouldn't be too hard, and without this, there is really no way for the reader to figure out whether or not the claim that the data are consistent is at all justified. A second issue here is that some of the other exposure-age data (e.g., Thanksgiving Point, Mt. Franke) appear to be available in online databases but not yet published in journal articles. I am sure the data are fine, but this may cause some citation problems. I refer that issue to the editors.

In addition, some of the text in this section gives the impression that the authors have a misunderstanding of the existing exposure-age data set. For example, consider the remark in line 273-ish about exposure ages from the Beardmore Glacier region, which states that exposure ages become younger downglacier for Shackleton and Beardmore Glaciers. This is misleading, because in both situations different glacial deposits have

been dated at different locations along the glacier. At Beardmore, LGM-age glacial deposits have been dated at several places along the glacier, but pre-LGM deposits have only been mapped or dated at one site at the top of the glacier. If site selection had been the opposite, i.e., past researchers had targeted pre-LGM deposits near the ice shelf and post-LGM deposits at the top of the glacier, the age relationship would also be the opposite. In other words, the distribution of ages reflects selection bias by researchers and cannot be used by itself to establish the existence of deposits of different ages, or any variation in the ages of particular deposits. Thus, this section of the present paper gives the impression that the authors have an overly simplistic understanding of this situation. In principle, it is possible that pre-LGM deposits are less common at low elevations, but that would have to be established via systematic mapping of these deposits. Thus, this section of the paper needs to be significantly reworked to focus on a comparison between specific mapped deposits of known or estimated ages, and not on a broad geographic analysis of a set of ages that is probably the result of selection bias.

The third aspect of this part of the review is that I could not understand the paragraph in lines 292-302. This mixes observations that the relationship between Be-10 and NO3 concentrations in depth profiles is complicated (which is true) with statements that have no clear connection to this observation such as "through a coupled approach...we developed a useful model for estimating soil exposure ages." I don't understand the connection between these two statements and others in this paragraph. I suggest starting again with this paragraph and trying to lead more clearly from observations to conclusions.

Finally, the last important thing here is that I found the disconnect between observations and conclusions to be most serious in section 6.3 ('Implications for ice sheet dynamics.'). This section contains several very broad statements. Only one of them (the discussion of the Sirius Fm.) is clearly related to the observations. This observation is interesting, but unfortunately doesn't help very much with the age of the Sirius

as a whole because it appears to be a loose clast of the Sirius that overlies the soil. Although these data show that the clast cannot have been dropped at this site until after 14 Ma, that doesn't directly constrain the age of the source material.

The other conclusions here are not related to the observations, and I think this area of the paper needs work. For example, "Our data support models...suggesting that EAIS advance and retreat was not synchronous..." (line 321). This is not correct. First of all, nowhere does the paper define what "synchronous" means and what the expected results of this study would be for "synchronous" and "asynchronous" options. Second, to me, the fact that old ages were not observed close to the ice margin at coastal locations simply indicates that the glacier thickens more nearer the grounding line, which is expected from basic glaciological principles and is not in conflict with a model in which thickening occurs at the same time everywhere. The fact that higher-Be-10 concentration soils are only found at more inland sites only shows that the authors were able to locate older deposits at inland sites, but did not find them at lower-elevation sites.

The discussion around line 333 also appears oversimplified and to not take into account basic glaciological principles. As noted above, there is no reason that there should be any relation between location and Be-10 concentration unless we know we are sampling a deposit of the same age at all sites. Therefore, a sample that diverges from this relationship also has unclear significance. The simplest explanation for this sample is just that you sampled a younger deposit. Whether that has any significance depends on the distribution of the deposits – are multiple deposits present at all sites? The other important point here is that relating the Be-10 concentration in surface samples to exposure ages relies on an empirical concentration-inventory relationship, which is quite scattered and not expected to be exact. Note that the relevant figure in Graly (2010) is on a log-log plot and displays quite a lot of scatter. Thus, even given a number of sites in the same deposit that were deposited at the same time and have the same vertically integrated inventory, significant variations in the surface Be-10 concentration

are expected to occur. To conclude that one site has a younger exposure age than another should involve showing that the difference between measured concentrations is significantly larger than we expect based on the scatter of the data used in the concentration-inventory transfer function. My overall point is that the oversimplified nature of this discussion gives the impression that the authors have not thought very hard about this. To get from the actual observations in this paper to a conclusion about glacier change, I would expect to the following steps: first, clearly describe, map, and identify glacial deposits that have been sampled; second, show whether or not samples from the same deposits are the same age, and then, third, conclude whether or not each mapped deposit is synchronous or time-transgressive. Many of these steps are absent here.

**VI. Suggested reorganization.**

This section makes some suggestions for how I would rewrite this paper to make it better. Mainly, I suggest significantly simplifiying the paper, focusing much more on the data that were actually collected in this study and not on broader topics that may seem more important but lack a clear relation to the data, and also being much more clear on the chain of reasoning between observations and conclusions. I suggest an outline that looks like the following:

1. Begin the paper by describing why the study was designed and conducted in the way that it was – as a means of estimating surface age for biological survey purposes – and then pointing out that the purpose of this paper is to describe the soil age data, which may also be useful for understanding geomorphology and glacier change in this area. I would remove the claim in the introduction that these data are likely to provide significant information as to the stability of the Antarctic ice sheets in warm periods.

2. Describe the sample sites and the approach of sampling a likely-post-LGM and likely-pre-LGM site in each area. Discuss in detail the physical and geomorphic characteristics of the site as well as any evidence for the mode of deposition of the parent

material and also whether the soil is inflationary or deflationary.

3. Explain how meteoric Be-10 in soils works in a way that is simpler and clearer than it is in the present paper, by removing Equation 1 and focusing on the relationship between inventory and age and the need to relate concentration to inventory to make an estimate of the age from one surface sample. Explain both ways of relating $N$ to $I$. Be clear about what "inheritance" is.

4. Explain the expected relationship between Be-10 and NO3.

5. In the data analysis section, begin by establishing whether the basic premises of the study (ice-distal sites should have more Be-10, and LGM-age sites should have the amount of Be-10 expected to have accumulated since the LGM) are true. Note that the depth profile data are not adequate to estimate background concentrations, and remove this section of the discussion. After addressing the basic validation of the approach, move on to secondary questions such as whether presumed LGM-age sites have similar Be-10/NO3 inventories up and down the glacier, and differences in Be-10/NO3 inventories among pre-LGM sites.

6. Convert concentrations to exposure ages and compare these to the expected distribution of LGM deposits as well as other exposure age data for the sites where there are some data. Use maps of these sites to clearly show the geographic relationship between your and other data.

7. With regard to the implications of these results for larger-scale issues having to do with ice sheet change during warm periods, I don't think the exposure age aspect of these results significantly changes the overall picture that previous research has derived from the existing several thousand exposure ages from Antarctica. On the other hand, the idea that salt accumulations can give some information on past warm climates (was it warm enough for liquid water to be present in soils, and if so, when?) could be very significant. Unfortunately, there is very little discussion of this in the paper. From first principles, I would expect NO3 and Be-10 to be correlated in dry

soils, because both would accumulate and not be removed. But as soon as water is present and leaching of NO3 can occur, one would expect a lack of correlation. Thus, the relationship between these two soil age proxies could be quite valuable for paleoclimate. I would give this more attention in a revised paper.

In general, in rewriting this paper, I very strongly urge the authors to focus much more on the specific things that they measured and observed. As discussed above, I got the very strong sense in reading this paper that the authors were unsure whether or not their data set would be perceived as a significant contribution by itself, so they felt like they had to add discussion of larger-scale issues about Antarctic ice sheet change to increase the perceived interest of the paper. I think this does the paper a disservice. The observations in this paper are, in fact, relevant and of interest by themselves. There are not enough data in this paper to solve any major problems having to do with glacial chronology that have not been solved by the much larger data set of pre-existing exposure-age data. However, the observations here are very valuable in generally understanding the residence time and disturbance frequency of Antarctic soils, which is necessary both for the original ecological-succession aspect of this study and also to make exposure-dating and soil development studies better in the future. The observations in this paper can stand by themselves without the need to bring in broader, but largely unrelated motivations.

**VII. Minor comments, by line number.**

Line 37 (The WAIS has been drastically reduced in size) and line 52 (A growing body of work that suggests...susceptible....). These areas incompletely describe the evidence for ice sheet change during warm periods. There exist model simulations that show that deglaciation of very large marine-based areas of the ice sheets is possible during warm climates. These are not evidence, but hypotheses that the model simulations show are physically possible. There is some indirect evidence (e.g., marine oxygen isotope data) that, given several assumptions, may be consistent with this hypothesis, but is also consistent with the hypothesis that minimal deglaciation occurred. There

is one piece of direct evidence (Be-10 in Siple Coast subglacial till; see Scherer and others) showing that the WAIS was smaller by an unknown amount sometime during the later Pleistocene. There is no direct evidence that hypothetical collapses simulated by ice sheet models took place. In fact, the best effort so far to test this hypothesis by subglacial bedrock recovery drilling in West Antarctica (Stone and others, recent WAIS meeting abstracts describing bedrock recovery drilling at Pirrit Hills) did not show any evidence for WAIS collapse. Thus, ice sheet collapses during warm periods need to be presented as a hypothesis and not as an accepted fact.

Note that the text around line 75 is much more clear in this regard and correctly distinguishes evidence and model predictions.

Near Line 100 . The authors should not mix up evidence for sustained aridity in ice-free areas with evidence for changes in the size of the ice sheet. Aridity does not necessarily require a large ice sheet, and ice sheet collapses due to marine ice margin instabilities could have occurred during cold, arid conditions. These two lines of reasoning should be kept separate.

Line 101-102. I did not understand these sentences.

Line 117. "High rates" is incorrect. Because this area is extremely arid by global standards, salt is delivered at a very low rate when compared to normal places. What is different here is not a high rate of supply but a low or zero rate of removal.

Line 122-3. This discussion gives the impression of not being well founded in glacial-geological observations. The critical difference between moraines deposited by frozen-based and wet-based ice is not their size, but rather their sedimentology. I looked at imagery of the Bennett Platform moraines and although they are large, they appear to be mostly composed of large boulders. No evidence is given in this paper that they include a fine-grained, matrix-supported till with striated clasts that would indicate formation by wet-based ice. If the authors did observe this, they should certainly describe it, with pictures, because matrix-supported tills near the ice margin in this region

would be very surprising. It seems more likely that these moraines are typical boulder moraines deposited by frozen-based ice, and their anomalous size may simply be related to the supply of boulders from large overhanging cliffs.

Line 140-ish. I think this could be stated more clearly simply by saying "We collected surface samples at all sites and 3-sample depth profiles at three sites."

Line 198ish. Because the sites you are sampling are soils and not rocks, I don't think these rock surface erosion rates are relevant. I suggest looking at papers by Dan Morgan and Jaakko Putkonen about the Dry Valleys to get an idea of the expected range for erosion rates of unconsolidated material. However, as noted above, most of these data are from hillslopes (although not all) and it's very possible that sediment deposition, rather than erosion, is taking place at some of the sites in the present paper.

line 204. What is the "coast"? It appears that the "coast" here is where the glacier flows into the ice shelf, but that makes very little sense in this context if one is thinking of the ocean as the source of salts. Open ocean is much farther away.

Line 269. The amount of time that soils are ice free must be longer for sites that are farther away from the glacier simply because of geometry. The ice sheet cannot cover more ice-distal sites unless it has already covered the ice-proximal sites. Thus, for any ice advance-retreat history, ice-distal sites will always be exposed longer. My point is that this is not a conclusion of the study (which is what this text sounds like), but it must be true under any circumstances no matter what the results.

---

## Author Comment (AC1) · 1 Oct 2020

Dear Dr. Koppes,

We are grateful to Dr. Goehring and an anonymous second referee for their thoughtful and detailed reviews of our manuscript, "Relative terrestrial exposure ages inferred from meteoric 10Be and NO3- concentrations in soils along the Shackleton Glacier, Antarctica." We have addressed the two reviews in detail with pertinent questions, comments, and concerns distilled below. A PDF is also attached with built-in organization.

[Figure]

To summarize, we agree with both Dr. Goehring and Referee #2 that the manuscript would greatly benefit from re-framing and clarification, particularly in the introduction, methods and discussion. The manuscript in its current form is staged as a geomorphologic study. Although the measurements, data, and interpretations we present are useful and of interest to the glaciological community, the original design of the study was to support a biological survey. The goal of the study is still the same – to calculate relative surface exposure ages – but the original purpose in determining these ages was to better understand ecological succession and refugia following glacier advance and retreat. As Referee #2 points out, this is not mentioned in the manuscript. Additionally, much of the current text is focused on the broader interpretations of the data, as opposed to the data themselves. As this is the first work to relate meteoric 10Be and nitrate concentrations in this manner, we agree that there needs to be a greater emphasis on method/proxy development and application.

For the revision, we will focus more on the points mentioned above and suggested by the reviewers. Although the suggested revisions are major/substantial, particularly for the introduction and discussion, with the framework developed from the referees' comments, we believe the manuscript and its impact with be much stronger. Thank you for soliciting these useful reviews.

Best regards, Melisa Diaz Postdoctoral Scholar Woods Hole Oceanographic Institution

———————————————————————————————————————— 

Brent Goehring (Referee) bgoehrin@tulane.edu

General Comments Diaz et al. present a compelling study showing the utility of combining measurements of meteoric 10Be with soluble nitrate as a means to determine surface exposure ages. In this case, they apply their new method to soils adjacent to Shackleton Glacier, Antarctica. However, their new methodology, particularly the combined use of nitrate and 10Be is not well-enough described. Additionally, and as noted

below, there needs to be a rigorous uncertainty analysis completed. All that being said, I will very much enjoy seeing this paper published, but for now it needs revision. The methods and results are interesting from an applied sense in that it could be used elsewhere, but their work also adds to the glacial history of the Transantarctic Mountains. Below I present general comments and then further below I present a number of detailed comments and suggest changes.

——As detailed in our response to Referee #2, we believe the manuscript will significantly benefit from the suggested re-framing. We will also greatly expand and describe our meteoric 10Be and nitrate methodology, particularly regarding mobility and wetting history.

The one supplementary figure showing the relationship between max 10Be concentration and total 10Be inventory should not be buried in the supplement.

——We will bring this figure into the main text.

I find that the introduction reads too much like a thesis introduction. All of the content is very good, but I think it could use a bit of streamlining that will help motivate the rest of the paper a bit better, as I think you need to also address the limitations of in situ exposure dating, as you mention later on, but it could benefit from being a bit earlier.

——As per Referee #2's suggestions, we have re-framed and rewritten the introduction to focus on the original goals behind collecting and interpreting these data – to understand relative surface soil ages for biological survey purposes. We believe that with the re-framing, the manuscript be more streamlined and focused.

Bear in mind this is purely a stylistic opinion can certainly be ignored. Throughout the manuscript, anywhere there is a reference to an age, rather than a duration, need to use Ma instead of Myr.

——We will make these changes to be in compliance with journal format.

There is overall a lack of uncertainty analysis that needs to be completed, particularly exploring the sensitivity of your various age determination models to parameter variance. The measurement uncertainties in this case are tiny compared to other uncertainties. A full error analysis will greatly strengthen the conclusions made in the paper and really needs to be done before publication. A bootstrap approach should be sufficient.

——The models that we have used in this work have been described and tested in great detail in previous studies, which include sensitivity analyses (e.g. Willenbring and von Blanckenburg, 2010; Graly et al., 2010). In general, the exposure age estimates using equations 1-4 are particularly sensitive to erosion and deposition rates. Since these values could not be determined for each sampling location, we chose to refer to our ages in a relative framework. We believe this will be more evident in the revision.

There is far too much framing of the study around Pliocene glacier dynamics, and particularly the Sirius formation. I'd much prefer to see the expansion of the possible newish and important approach that can be implemented combining 10Be with nitrate as a measure of surface exposure duration.

——We agree with Dr. Goehring and Referee #2. We are now focusing on estimating surface exposure ages and the use of atmospherically derived salts in estimating wetting history and exposure ages. This is detailed further in our responses to Referee #2.

Figure 8 demonstrates very nicely a coherent pattern of ice thinning/retreat. This needs to be played up, and the return late in the manuscript to the Sirius Group detracts from the novelness of the work.

——We now focus on our novel approach to estimating relative exposure ages and how these data contribute to our understanding of ecological succession and glacier change.

Detailed Comments Line 37: Please provide a citation or two for the first part of the

sentence. There is actually quite sparse direct evidence for smaller interglacial extents relative to the Holocene and much is largely inferred from distal evidence or modeling. Additionally, the Ross Embayment is a large area and thus this statement is somewhat vague.

——We will better clarify and support these points.

Line 51: How are calculated and estimated exposure ages any different from each other? I know this seems nit-picky, but it is somewhat strange wording as your estimated exposure age had to be calculated first.

——We will expand and clarify our methodology and terminology.

Line 62: Unsure what "these studies" are. Are you referring to those cited at the end of the sentence or the sentence prior? If the sentence prior, why do you have a new set of citations?

Section 2.1 Should be worked more into the introduction in my view.

——With the re-framing of this manuscript to focus more on the data present and their specific implications, much of the introduction will be re-written. We will be sure to clarify throughout.

Line 78: Nishiizumi et al., 2007 is not actually a half-life study, an outcome of the standardization is that a different half-life than had been used must be used. Recommend citing: âËŸAʹc Korschinek, G., Bergmaier, A., Faestermann, T., Gerstmann, U., Knie, K., Rugel, G., Wallner, A., Dillmann, I., Dollinger, G., Gostomski, C., Gostomski, C., Kossert, K., Maiti, M., Poutivtsev, M., Remmert, A. (2010). A new value for the half-life of 10Be by Heavy-Ion Elastic Recoil Detection and liquid scintillation counting Nuclear Instruments & Methods In Physics Research Section B-Beam Interactions With Materials And Atoms 268(2), 187 - 191. https://dx.doi.org/10.1016/j.nimb.2009.09.020 âËŸA ʹc Chmeleff, J., Blanckenburg, F., Blanckenburg, F., Kossert, K., Jakob, D. (2010). Determination of the 10Be halflife by

multicollector ICP-MS and liquid scintillation counting Nuclear Instruments & Methods In Physics Research Section B-Beam Interactions With Materials And Atoms 268(2), 192 - 199. https://dx.doi.org/10.1016/j.nimb.2009.09.012

——We thank Dr. Goehring for the reference and will update our citations.

Line 101: Given the general absence of anything resembling soils or till in most of Antarctica, one could argue that applying meteoric 10Be is far more spatially limited, e.g. to regions of the Dry Valley, for example. Thus, I am not sure I would argue for your method by arguing that in situ exposure dating is limited, but instead argue that they are complementary.

——We will be sure to clarify our methodology in the revision.

Starting line 107: I am not sure the bedrock lithology is all that relevant. I understand you want to show the protolith for weathering products, but I think it could be said more concisely. I think the geologic setting paragraphs could be combined.

——We will make the geologic overview more concise and focus on soil properties and landscape features.

Line 123: Suggest changing "glacial dynamics" to "glaciers"

——We will make this change.

Line 128: By two samples, do you mean two surface samples? Suggest clarifying the text here, especially since you have depth profiles samples from elsewhere.

Line 130: In your reference to sample distance from the glacier, are you largely referring to further away as controlled by elevation, or by horizontal distance? I think some clarification of this could be useful, as depending on the valley geometry, changes in ice thickness might not be significantly further away from the glacier, or vice versa. It might be more constructive and more generalizable to perhaps say that two samples were collected, one adjacent to the glacier, characteristic of times similar to the current extent

and one further away representative of significant changes in glacier size (larger). A useful column in your table and the way most Antarctic glacier change is expressed is as change in ice thickness.

——We will clarify and expand upon our sampling methodology.

Line 142: Why not report the fraction between 2mm and 425 microns? Was none present? Sand usually extends to 2 mm.

——We set our limit to medium sand size and will clarify in the text.

Line 170: Suggest not starting paragraph with "However. . .." I suggest that when laying out your calculation methods, that the equations flow more within the paragraph, rather than being at the end of each paragraph. I found it somewhat hard to ready.

——We will re-organize this section.

Line 179: Suggest adding "any" before "have meteoric" Line 197: Delete "which"

——We will make the correction.

Line 202: Confused because didn't you calculate two samples from every location, only profiles from only a few?

——We measured meteoric 10Be and nitrate concentrations from at least two samples (generally near glacier and furthest away) at all sites. We measured one profile at each site for nitrate and profiles from Roberts Massif, Bennett Platform, and Thanksgiving Valley for 10Be. We will make this clearer.

Line 206: The lack of an expected concentration based on regressions against distance and elevation might just be spurious and making predictions from these regressions very tenuous. I suggest removing this sentence.

——We will remove this sentence.

Line 222: The ages are not necessarily minimum ages, as while you may be overcorrecting for inheritance because you don't know the background inventory, you also do not a priori know the erosion rates of the soils, even though you make assumptions. I suggest that rather than couching the ages as minimum, as they are only minimum relative to your max limiting no inheritance ages, you just present them as best estimate given knowledge of the parameters.

——We thank Dr. Goehring for the suggestion and will follow his recommendation.

Section 5.3.1 This section is very confusing in terms of what you did and is not represented in the methods at all, thus the results presented here come out of nowhere. There needs to be a clearer explanation of what was done. I think the approach is really neat and valuable, but right now it just isn't explained well-enough. I am also very confused upon the first and second read as to what was done with what profile, as the second paragraph mixes results from sites with both measurements and sites without. Section 5.3.2 Like the prior section, where there are a number of inferred methodological requirements, more expansion of the discussion is needed to aid the reader that may only have casual knowledge of meteoric 10Be knowledge as I can see many readers being most interested in the inferred ice history. I think one thing that will help immensely is that this and the prior section are more traditionally considered as part of the discussion and the results purely your 10Be and NO3- measurements. Now, if you were to present the calculation methods using nitrate and the inventory vs max concentration analyses in the methods, then you could keep in the results. At present, there is just a bit too much mixing and overall not enough time dedicated to these important sections that you then use extensively in the discussion below. Also, best I can tell Figure 8 does not show the relationship between max concentration and total inventory, please investigate, or do you mean to only present the max exposure ages.

——We will reorganize these sections to present our results in a more logical manner. We will expand the nitrate and 10Be methodology, which should help clarify our results and discussion. Figure 8 includes both the max exposure ages from the "inventory

method" and the estimated ages using the "nitrate method". We will make this clear.

Line 247: Please elaborate or define what the model limits are, as this is not defined. Presumably just the influence of the time scale to 10Be saturation given an erosion rate. I also wish there were different terminologies used with regards to calculated vs estimated. Perhaps refer to one as the apparent max limiting age and the other a model age?

——We briefly mention that the maximum age the model can calculate is ~14 Ma and will make the model limit clear. We will also change and define our terminology for clarity.

Line 260: The correspondence with in situ ages is quite remarkable. What is lacking though is a clear representation of the two different data sets. This is why I suggested that perhaps you determine the elevation above modern ice surface and thus you can then make age vs elevation plots for your data and the in situ data. I think will drive home much more clearly the correspondence. Or you could consider maps showing the various bits of data, but I think they will get very busy very quickly. While the correspondence in many scenarios is striking, one thing to consider and make sure you make clear is whether the in situ data are from bedrock or from erratics, as they will have quite different exposure ages and thus your soil ages might always be older than nearby in situ erratic exposure ages. The fact that your meteoric ages, including nitrate corrected, agree so much with in situ erratic ages suggests some mechanism for resetting and flushing of 10Be or that your model is determining the pre-LGM inherited concentration quite clearly. I think this needs further discussion and is important to highlight more.

——We agree with Dr. Goehring and Referee #2 that our data need to be better compared to the in-situ ages from previous studies. We will plot the previously published ages alongside ours and indicate which were sampled from moraines and boulders. We are also expanding our interpretation of the relationship between nitrate and 10Be

and possible implications for disturbance history.

Line 272: Need a reference for exposure dating results from Beardmore Glacier.

——We will move the reference up so that it is clear.

Line 288: The arguments about the suitability seem out of place and kind of come out of nowhere and seem to set up a strawman for no apparent rea- son. I suggest removing and focusing on the apparent success of the nitrate correction given the good agreement with in situ exposure dating.

——We will move the text regarding the suitability of nitrate as an indicator of relative surface exposure age to the introduction. We believe it is important to indicate why we chose this atmospherically-derived constituent for our study. The discussion will focus again on testing and validation of our data.

Starting line 292: The first few sentences of this paragraph read too much like a con- clusions section. Suggest revision.

——We will revise.

Line 303: As mentioned above, the nitrate regression models needs further description and elaboration, particularly since this really is the first major combined use of these two measures.

——We will elaborate the nitrate model throughout the text.

Line 306: Wouldn't a lack of correlation be expected given the exponential fall off of a 10Be profiles, so that below a certain depth there will be little to no variance in the 10Be concentration and presumably the same in nitrate? Yes, a lack of correlation would be expected. We will clarify our assumptions and hypotheses in the text.

Line 352: Suggest rather than saying delayed response that you more generalize it and just say different response from Ross Ice Shelf confluent outlet glaciers, or something to that effect.

**ESurfD**
––We will edit this text.

Line 358: This conclusion is spot on and is a major finding of the paper, however its use, the details, etc. are not elaborated on enough earlier in the manuscript.

––With the proposed re-structuring and re-framing, there is much more emphasis on our nitrate and meteoric 10Be data.

Line 365: The broader question then becomes, how do we differentiate between a site with inherited meteoric 10Be that was covered by LGM ice from a site that was never covered during the LGM and more recent glaciations. This is a question that the in situ community has struggled with. We are only starting to get clarity from a focus on erratic exposure dating with long-lived nuclides or application of in situ 14C to erratics and bedrock. Recent work in the Weddell Embayment with very old erratic and bedrock in situ ages were clearly covered by LGM ice as shown by in situ 14C, including preservation of delicate features like moraines (e.g., Nichols et al., 2019). Thus, during a say 10 kyr long ice cover period, how much of a reduction in the meteoric 10Be signal can be expected? What about reduction in nitrate? Presumably unless the ice is wet based, neither will be mobilized and then you need the correct pH conditions. These thoughts are briefly touched on, but the manuscript could use a bit more elaboration on the long-term interpretation of the signal recorded by your methods and what its implications are for interpreting surface processes in Antarctica. Thus, it could be useful to elaborate on the presence of polythermal moraines, why are some areas reset for the meteoric and in situ methods.

––Dr. Goehring brings up some very important questions. However, the answer to many of these questions are unknown. Due to uncertainties with sediment transport, both modern and in the past, it is unclear how meteoric 10Be and nitrate would be affected over extended periods of time. Under persistent arid conditions, we expect nitrate to be largely conserved. As stated previously, these concerns will be addressed in the revision.

Figure 1: Not sure if this is supposed to be this way of if some strange PDF artifact, but the exposed rock areas are banded. I also think you could make the overview map larger scale to give readers a better context of the Shackleton Glacier.

——The exposed rock areas where we samples are indeed banded, hashed, and checkered in the figure to indicate lithology as per the key. We will make the overview map larger.

Figure 3: A similar figure thinking about the fate of nitrate during ice cover would be informative.

——We hope that the expanded text will suffice instead.

Figure 4: Add panel labels please. Also, it is confusing that in the Shackleton glacier map, the coloring represents concentration, but you then use the same colors for the different sites, or is it only the arrows? This is somewhat confusing, and I suggest not using colored arrows that are the same as the color scaled points for concentration. Here the figure is trying to show too much.

——We will update this figure.

Figure 5: This figure and all figures. Are uncertainties shown, but smaller than the symbol? Please note this or add uncertainties if need be.

——Due to the log scale, the measurement uncertainties are small, as indicated in Table 1.

Figure 6: Suggest removing the lines connecting the points, as it implies that there is a trend in grain size % between the points. The measurements are point measurements.

——We will update this figure.

Figure 7c: Please provide equations for the fits along with uncertainties on the fit parameters. These uncertainties then need to be used for error analysis on the resulting ages.

1——We will add these elements.

Table 2: I suggest presenting uncertainties using the same exponent for the measured value and Uncertainty.

——We will update this table.  
* * *
Anonymous Referee #2 I. Summary. The summary of this review is that the data collected in this paper are useful, interesting, and valuable to publish. In general, the idea that accumulation of atmospheric constituents in Antarctic soils is useful for estimating soil ages and residence times is important from many perspectives, including glacier change, paleoclimate, and biology, and this paper contains a lot of data that are relevant to this topic.

II. Overall motivation of paper. II.1. The way the paper is motivated makes the experimental design look bad when, in fact, it is not. The experimental design of this study is very well designed from the perspective of a biological survey. The use of atmospheric fallout constituents of soils to rapidly get an approximate idea of the soil age, and distinguish soils that were ice-covered during the LGM from soils that have not been ice-covered for millions of years, is a smart, well-designed approach that is likely to be effective for its intended purpose. On the other hand, the study is not well designed for the purpose of reconstructing past glacier change. The point here is that if the present study was motivated by the original objectives of collecting geological information needed to study ecosystem succession, it would be perceived by readers as well-conceived and well-designed. If motivated as a study of glacier change as in this paper, on the other hand, the experimental design appears weak and inadequate by comparison to other studies. I very strongly urge the authors to change this emphasis. They should clearly explain the purpose of the overall project that led them to the experimental design used here. It is true that the data collected for this purpose also have value in quantifying glacier change, so there is nothing wrong with focusing

additional discussion on that later in the paper, but motivating the entire paper from this perspective makes the paper much weaker than it should be.

—–Referee #2 is indeed correct that the samples collected for this study and for this analysis were for a larger study on ecosystem succession following changes in climate – in this case, glacial advance and retreat. The goal of this smaller study remains the same. We sought to determine relative surface exposure ages of ice-free areas along the Shackleton Glacier. Though these data can be used in understanding glacial change, we agree that the introduction and discussion should be refocused to emphasize our broader goals and significance to ecological refugia.

II.2. The way the paper is motivated leads the paper off into vague theories that can't be addressed by the data. The most problematic part of the paper from this perspective is the first two paragraphs of the introduction (lines 33-45) and section 2.1 ("Stability of the EAIS"), lines 55-76. The introduction discusses the fact that the Antarctic ice sheets are proposed to have been a lot smaller during some warm periods in the past. While it is certainly true that this has been hypothesized and that in a very general sense this is a strong motivation for studying past changes in the size of the Antarctic ice sheets, there is almost no connection between this overall idea and the specific observations described in this paper. As discussed above, if this is the motivation for the work, the work looks inadequate.

Section 2.1 is much more problematic. It would be clearer to simply state that it is not yet known whether or not the East Antarctic Ice Sheet was significantly smaller during past warm climates. The second problem in this section has to do with confusion between ice sheet change and climate change. The discussion of how long polar desert conditions have prevailed in the TAM is important in this paper because it gives context for one potential application of salt deposition in soils, i.e. the idea of a "wetting age" in which the amount of salt that has accumulated can give information on when liquid water was last present. However, this important implication of the idea is not at all mentioned here.

——We are changing the focus of the introduction to discuss ecological dispersal and refugia during glacial periods, the overall glacial history of Antarctica, the need to understand exposure ages in this region, the goals of this study to understand soil ages, and the applications both to ecology and geomorphology. We will remove the text and section(s) on East Antarctic Ice Sheet stability and instead shift the focus to persistent arid conditions, as the desert climate is particularly important for salt accumulation and the development of our nitrate proxy.

III. Oversimplified explanation of atmospherically produced Be-10.

With regard to section 2.2, the main thing the authors need to get across here is that meteoric Be-10 builds up in soils, so the total amount of Be-10 present in a soil profile is related to the age of the soil. This information is here, but it is missing some important context and mixed up with other confusing things. One, the authors should clearly state that meteoric Be-10 is mobile in the soil, so it is not the concentration at any particular location that is proportional to the exposure age, but instead the total inventory in the entire soil profile. Two, the behaviour of meteoric Be- 10 and salts in soils may be quite different, for example because Be-10 remains bound to particles even when the soil is wet, whereas salts are mostly mobile in water.

——While we do discuss meteoric 10Be systematics later in the text, we agree that it would be beneficial to better describe the system in more detail here and expand upon salt accumulation/mobility.

The other important area here that needs to be either here or in the section on study sites is a discussion of exactly what landforms were sampled and how that relates to meteoric Be-10 systematics.

——We will add a table listing on the landforms and features we sampled at each location and any notable features, such as nearby ponds, polygonal ground, etc. We will also include additional overview text in the study sites section. Mapped geomorphologic features, such as drifts and moraines, are poorly documented in this region.

Though we did not focus on identifying such features, we agree that the sample location descriptions will be informative for both this study and future studies.

Section 4.3 is about how to quantitatively interpret Be-10 concentrations as an exposure age of the soil. This section would benefit from several improvements. Specifically, Equation (1) seems to be missing important elements. A common approach in the meteoric Be-10 literature to simplify this relationship and make it more useful is to write the governing equation for the soil inventory I (atoms per cm2, vertically integrated) instead of the concentration, like: dI/dt = Q − _I − ENs (2) where Ns is the surface concentration (atoms/g) and E is the erosion rate in mass per area units. Using this equation instead of Equation (1) would make this paper much clearer. Alternatively, this paper could simply refer to other literature that describes meteoric Be-10 systematics in detail – it is not necessary to reinvent the wheel here.

––We understand that the simplicity of Eq. 1 may be misleading. We will remove the equation and replace it with a more comprehensive equation.

Finally, an important point for these sites is that it is not even clear that erosion is taking place throughout the ice-free at areas all. Perhaps the only process that can bring new sediment to the surface and permit deflation would be periglacial disturbance of the soil. This issue reminds me that an important thing that needs to be added to section 3 is some discussion of the surface characteristics of each site, including presence or absence of boulder pavements and periglacial features like cracks and polygons, because these features are relevant to interpreting the Be-10 data.

The overall point of this section is that it is not at all clear to me that erosion should even be included in the relationship between inventory and age for these sites. For this paper, I think it might make the most sense to simply relate inventory to exposure age by dI/dt = Q − _I, i.e. disregarding erosion and deposition, and accept that this approach might be either under- or over-estimating exposure ages.

––As mentioned previously, we are adding a table describing the surface features of

each sample location, including whether the samples were collected on valley floors or hillslopes. While we did not sample features such as polygons and boulder pavements, it is crucial to indicate such. Once the samples are further described, we believe the inclusion of erosion rates will become more clear.

[T]his section has to clearly explain how one measures the Be-10 inventory. As already discussed in the paper, this can be done in two ways, either by measuring a complete depth profile and integrating, or using an empirical relation between surface concentration and inventory as in the Graly paper. An additional problem with this section is that "inheritance" is not clearly defined, which is confusing. Finally, a clear definition of "background" in the context of a depth profile is needed here. The basic concept (that the concentration is supposed to decrease with depth until you reach a depth where the concentration becomes invariant with depth) is correctly described near line 182, but what is missing is a clear statement of how one knows that one has observed this. Overall, what I suggest doing here is noting that in principle the depth profile method is one possible way to estimate I, but it can't be used in this application because insufficient data were collected – and then move on to discussing the approach of using an empirical correlation between N and I to estimate I.

——Though Referee #2 acknowledges that we have introduced and described inheritance, we will clearly define both inheritance and background in the context of our study. In our study, we provided two estimates of inheritance: 1) integrating the lowest concentration at the bottom of the depth profile and 2) an empirical correlation between surface N and I. Referee #2 correctly mentions that we have not satisfied the typically criteria for attaining background measurements of meteoric 10Be using method #1. We will better emphasize the uncertainty of these calculations/estimates and focus on method #2.

IV. Data analysis.

I did not understand what the purpose of these regressions is [Fig. 5]. Because I

don't see any basic physical relationship that would support linear regression of concentration against elevation/distance, as a reader I am left with the impression that the authors simply felt that there should be some linear regressions in the paper. I am not sure this is the impression that the authors want to give the reader. It makes the paper seem weak and confused, and I urge them to remove this section of the paper.

——The purpose in including the regressions between meteoric 10Be concentration and elevation and distance from the coast was to demonstrate that there is a geographic component to 10Be concentration, probably related to glacial history. While Referee #2 correctly mentions that different drift sequences in a single sampling site would yield different 10Be concentrations, we argue that the potentially different drift sequences are due to differing glacial histories. Samples at lower elevations near the glacier were likely exposed to more periglacial processes than samples collected further inland and at higher elevations. This is demonstrated in our regressions, and we will de-emphasize this section and make these points more clear in the text.

The second area that seems problematic to me in this section of the paper is how the authors approach estimating the Be-10 inventories in section 5.2. What I suggest doing here is removing section 5.2, noting that the depth profile data do not allow estimating I accurately, and rely entirely on the empirical-correlationbetween- I-and-N approach for estimating I, which is already clearly covered in section 5.3.2. This is not really a major substantive change to the paper, because at most of the sites there are only surface data in any case.

——As stated in a previous comment, we will shift away from calculating I though integration and instead focus on our values estimated from the empirical correlation between N and I.

The third area that I think needs additional discussion in this section is the discussion of the relation between Be-10 and nitrate concentrations. To summarize, this section needs to be made much more clear so that the reader can understand when concen-

**ESurfD**
trations, surface concentrations, and inventories are being discussed, and what differ-
ences in behaviour of Be and NO3 could lead to positive or negative correlation. This
may require making this section substantially longer in order to explain the reasoning
step by step so that the reader can follow it.

——We agree with Referee #2 that this section can and should be greatly expanded
upon. Additional text will be added describing the relationship between 10Be and ni-
trate for each of the three soil profiles and the factors which have likely contributed to
the observed concentration behavior.

V. Discussion and interpretation areas.

The first aspect of the discussion that needs additional work is that the most basic pre-
diction of the experimental design is that, first, Be-10 inventories and/or concentrations
should increase with distance from the ice margin at each site, and, second, Be-10 in-
ventories/concentrations for the ice-proximal samples that are supposed to have been
exposed after the LGM should have magnitudes that are appropriate to post-LGM ex-
posure, i.e. 10-15,000 years of surface exposure. I would do this with a figure for each
site showing distance from the nearest ice margin on the x-axis, and Be-10 and NO3
concentrations on the y-axis.

——We agree that an additional figure showing 10Be and nitrate concentration versus
distance from glacier would be beneficial in supporting the overall experimental design.

The second aspect of the discussion that is incomplete/too abbreviated is the section
beginning on line 260 that compares the results to existing exposure-age data from
glacially transported boulders. Personally, what I would view as minimally adequate
here is a map view of each site where there are existing/published exposure age data,
showing the location of the soil pits described here, the location of any moraines or
drift boundaries including any hypothesized LGM ice limit, and also the location of the
independent exposure-age data, which will be mostly boulders dated by some in-situ
produced nuclide. Alternatively, instead of maps, these could take the form of plots

with distance from the ice margin on the x-axis, and exposure ages calculated from the various data on the y-axis.

A second issue here is that some of the other exposure-age data (e.g., Thanksgiving Point, Mt. Franke) appear to be available in online databases but not yet published in journal articles. I am sure the data are fine, but this may cause some citation problems. I refer that issue to the editors.

——Though there are only published data from Roberts Massif, we agree that it would be helpful to plot the in-situ data from previous studies and ICE-D alongside our data to support our comparisons. Confident estimates of the LGM trimline and mapped drifts for the other sites and features we sampled in the Shackleton Glacier region do not currently exist. Regarding the citations, we will cite Spector and Balco, 2020, which include the ICE-D dataset.

In addition, some of the text in this section gives the impression that the authors have a misunderstanding of the existing exposure-age data set. For example, consider the remark in line 273-ish about exposure ages from the Beardmore Glacier region, which states that exposure ages become younger downglacier for Shackleton and Beardmore Glaciers. In principle, it is possible that pre-LGM deposits are less common at low elevations, but that would have to be established via systematic mapping of these deposits. Thus, this section of the paper needs to be significantly reworked to focus on a comparison between specific mapped deposits of known or estimated ages, and not on a broad geographic analysis of a set of ages that is probably the result of selection bias.

——Considering the concerns Referee #2 raised regarding this section, we have decided to largely remove it.

The third aspect of this part of the review is that I could not understand the paragraph in lines 292-302. This mixes observations that the relationship between Be-10 and NO3 concentrations in depth profiles is complicated (which is true) with statements that

have no clear connection to this observation such as "through a coupled approach...we developed a useful model for estimating soil exposure ages." I suggest starting again with this paragraph and trying to lead more clearly from observations to conclusions.

——Given the overall manuscript reframing and editing of the discussion, we will improve clarity throughout.

Finally, the last important thing here is that I found the disconnect between observations and conclusions to be most serious in section 6.3 ('Implications for ice sheet dynamics.'). This section contains several very broad statements. Only one of them (the discussion of the Sirius Fm.) is clearly related to the observations. The other conclusions here are not related to the observations, and I think this area of the paper needs work. For example, "Our data support models...suggesting that EAIS advance and retreat was not synchronous..." (line 321). The fact that higher-Be-10 concentration soils are only found at more inland sites only shows that the authors were able to locate older deposits at inland sites, but did not find them at lower-elevation sites. The discussion around line 333 also appears oversimplified and to not take into account basic glaciological principles. To conclude that one site has a younger exposure age than another should involve showing that the difference between measured concentrations is significantly larger than we expect based on the scatter of the data used in the concentration-inventory transfer function. My overall point is that the oversimplified nature of this discussion gives the impression that the authors have not thought very hard about this. To get from the actual observations in this paper to a conclusion about glacier change, I would expect to the following steps: first, clearly describe, map, and identify glacial deposits that have been sampled; second, show whether or not samples from the same deposits are the same age, and then, third, conclude whether or not each mapped deposit is synchronous or time-transgressive. Many of these steps are absent here.

——These are all valid points. Given the other suggestions and changes throughout the manuscript, the revisions should rectify these concerns. Instead of focusing on EAIS

behavior, the revised manuscript focuses on the coupling of meteoric 10Be and nitrate to estimate relative ages. Since there are few, if any, data from many of the ice-free areas we sampled, we believe our data and measurements are still important. Additionally, by focusing on smaller-scale processes, we can make inferences regarding arid conditions in the CTAM. As we and Referee #2 point out, nitrate and 10Be profiles should appear and behave similarly in static persistent arid conditions since both constituents are atmospherically derived. Deviations from this expected relationship can indicate wetting or possibly erosion/deposition, which have particularly important implications for ecological succession. The points will be expanded and will primarily constitute the discussion and conclusions.

VI. Suggested reorganization. This section makes some suggestions for how I would rewrite this paper to make it better. Mainly, I suggest significantly simplifiying the paper, focusing much more on the data that were actually collected in this study and not on broader topics that may seem more important but lack a clear relation to the data, and also being much more clear on the chain of reasoning between observations and conclusions. I suggest an outline that looks like the following: 1. Begin the paper by describing why the study was designed and conducted in the way that it was – as a means of estimating surface age for biological survey purposes – and then pointing out that the purpose of this paper is to describe the soil age data, which may also be useful for understanding geomorphology and glacier change in this area. I would remove the claim in the introduction that these data are likely to provide significant information as to the stability of the Antarctic ice sheets in warm periods. 2. Describe the sample sites and the approach of sampling a likely-post-LGM and likely-pre-LGM site in each area. Discuss in detail the physical and geomorphic characteristics of the site as well as any evidence for the mode of deposition of the parent material and also whether the soil is inflationary or deflationary. 3. Explain how meteoric Be-10 in soils works in a way that is simpler and clearer than it is in the present paper, by removing Equation 1 and focusing on the relationship between inventory and age and the need to relate concentration to inventory to make an estimate of the age from one

surface sample. Explain both ways of relating N to I. Be clear about what "inheritance" is. 4. Explain the expected relationship between Be-10 and NO3. 5. In the data analysis section, begin by establishing whether the basic premises of the study (ice-distal sites should have more Be-10, and LGM-age sites should have the amount of Be-10 expected to have accumulated since the LGM) are true. Note that the depth profile data are not adequate to estimate background concentrations, and remove this section of the discussion. After addressing the basic validation of the approach, move on to secondary questions such as whether presumed LGM-age sites have similar Be-10/NO3 inventories up and down the glacier, and differences in Be-10/NO3 inventories among pre-LGM sites. 6. Convert concentrations to exposure ages and compare these to the expected distribution of LGM deposits as well as other exposure age data for the sites where there are some data. Use maps of these sites to clearly show the geographic relationship between your and other data. 7. With regard to the implications of these results for larger-scale issues having to do with ice sheet change during warm periods, I don't think the exposure age aspect of these results significantly changes the overall picture that previous research has derived from the existing several thousand exposure ages from Antarctica. On the other hand, the idea that salt accumulations can give some information on past warm climates (was it warm enough for liquid water to be present in soils, and if so, when?) could be very significant. Unfortunately, there is very little discussion of this in the paper. From first principles, I would expect NO3 and Be-10 to be correlated in dry soils, because both would accumulate and not be removed. But as soon as water is present and leaching of NO3 can occur, one would expect a lack of correlation. Thus, the relationship between these two soil age proxies could be quite valuable for paleoclimate. I would give this more attention in a revised paper. In general, in rewriting this paper, I very strongly urge the authors to focus much more on the specific things that they measured and observed.

——We are grateful to Referee #2 for such deep thinking and such a detailed review and have used their suggested organization as a guide for our revisions.

[Figure]

VII. Minor comments, by line number. Line 37 (The WAIS has been drastically reduced in size) and line 52 (A growing body of work that suggests...susceptible....). These areas incompletely describe the evidence for ice sheet change during warm periods. There exist model simulations that show that deglaciation of very large marine-based areas of the ice sheets is possible during warm climates. These are not evidence, but hypotheses that the model simulations show are physically possible. There is some indirect evidence (e.g., marine oxygen isotope data) that, given several assumptions, may be consistent with this hypothesis, but is also consistent with the hypothesis that minimal deglaciation occurred. There is one piece of direct evidence (Be-10 in Siple Coast subglacial till; see Scherer and others) showing that the WAIS was smaller by an unknown amount sometime during the later Pleistocene. There is no direct evidence that hypothetical collapses simulated by ice sheet models took place. In fact, the best effort so far to test this hypothesis by subglacial bedrock recovery drilling in West Antarctica (Stone and others, recent WAIS meeting abstracts describing bedrock recovery drilling at Pirrit Hills) did not show any evidence for WAIS collapse. Thus, ice sheet collapses during warm periods need to be presented as a hypothesis and not as an accepted fact. Note that the text around line 75 is much more clear in this regard and correctly distinguishes evidence and model predictions.

——We will be sure to make these distinctions regarding WAIS stability and collapse in the revised manuscript.

Near Line 100 . The authors should not mix up evidence for sustained aridity in ice-free areas with evidence for changes in the size of the ice sheet. Aridity does not necessarily require a large ice sheet, and ice sheet collapses due to marine ice margin instabilities could have occurred during cold, arid conditions. These two lines of reasoning should be kept separate.

——We will make these distinctions in the revised manuscript.

Line 101-102. I did not understand these sentences.

——We will revise and clarify.

Line 117. "High rates" is incorrect. Because this area is extremely arid by global standards, salt is delivered at a very low rate when compared to normal places. What is different here is not a high rate of supply but a low or zero rate of removal.

——We will make this correction.

Line 122-3. This discussion gives the impression of not being well founded in glacial geological observations. The critical difference between moraines deposited by frozen-based and wet-based ice is not their size, but rather their sedimentology. I looked at imagery of the Bennett Platform moraines and although they are large, they appear to be mostly composed of large boulders. No evidence is given in this paper that they include a fine-grained, matrix-supported till with striated clasts that would indicate formation by wet-based ice. If the authors did observe this, they should certainly describe it, with pictures, because matrix-supported tills near the ice margin in this region would be very surprising. It seems more likely that these moraines are typical boulder moraines deposited by frozen-based ice, and their anomalous size may simply be related to the supply of boulders from large overhanging cliffs.

——We agree with Referee #2 and will make this correction.

Line 140-ish. I think this could be stated more clearly simply by saying "We collected surface samples at all sites and 3-sample depth profiles at three sites." We will clarify the sampling procedure.

Line 198ish. Because the sites you are sampling are soils and not rocks, I don't think these rock surface erosion rates are relevant. I suggest looking at papers by Dan Morgan and Jaakko Putkonen about the Dry Valleys to get an idea of the expected range for erosion rates of unconsolidated material. However, as noted above, most of these data are from hillslopes (although not all) and it's very possible that sediment deposition, rather than erosion, is taking place at some of the sites in the present paper.

——Though it is well documented that ash layers and hillslopes have relatively high erosion rates, likely much higher than expected for soils in the CTAM, we will re-evaluate our erosion rates and overall usage.

line 204. What is the "coast"? It appears that the "coast" here is where the glacier flows into the ice shelf, but that makes very little sense in this context if one is thinking of the ocean as the source of salts. Open ocean is much farther away.

——Coast in this context represents the point where the glacier is no longer constrained by the TAM and flows into the ice shelf. We do not rely on distance to open ocean due to seasonal and yearly changes in this distance from sea ice extent. We will clarify in the text.

Line 269. The amount of time that soils are ice free must be longer for sites that are farther away from the glacier simply because of geometry. The ice sheet cannot cover more ice-distal sites unless it has already covered the ice-proximal sites. Thus, for any ice advance-retreat history, ice-distal sites will always be exposed longer. My point is that this is not a conclusion of the study (which is what this text sounds like), but it must be true under any circumstances no matter what the results.

——We agree and will clarify these points in the text.

Please also note the supplement to this comment:
https://esurf.copernicus.org/preprints/esurf-2020-50/esurf-2020-50-AC1-supplement.pdf

---

## Author Response (AR1)

Dear Dr. Koppes,

We are pleased to submit our revised manuscript, "Relative terrestrial exposure ages inferred
from meteoric $^{10}$Be and $NO_3^-$ concentrations in soils along the Shackleton Glacier, Antarctica." We are
very thankful to both Dr. Goehring and an anonymous reviewer for providing thoughtful comments,
suggestions, and edits, which have guided this revision. We have made the corrections proposed by Dr.
Goehring and Referee #2 in the tracked-changes manuscript at the end of our responses. However, due
to the substantial nature of the revisions, our edits are more clearly distilled and described below. In
addressing the concerns and questions raised in the two reviews, the manuscript has substantially
improved for publication.

To summarize, we agreed with both Dr. Goehring and Referee #2 that the manuscript would
greatly benefit from re-framing and clarification, particularly in the introduction, methods and
discussion. We have modified the narrative to focus on the distributions of meteoric $^{10}$Be and $NO_3^-$ and
how these data inform biogeography, climate, and glacial history for discrete points along the
Shackleton Glacier. As Referee #2 points out, we originally did not discuss the study design, which is
that of a biological survey. This is now included and strengthens the narrative. Lastly, we have clarified
our exposure age techniques, especially the relationship between $NO_3^-$ and meteoric $^{10}$Be since this is a
new method. Our data and results offer some of the only surface exposure ages in the Shackleton
Glacier region and suggest that much of the southern portion region has remained hyper-arid since at
least the Pleistocene. These findings are particularly important in understanding ecological succession
and glacial history in the Transantarctic Mountains.

Best regards,

Melisa Diaz (on behalf of all authors)

Postdoctoral Scholar
Woods Hole Oceanographic Institution
The Ohio State University
Byrd Polar and Climate Research Center

**Brent Goehring (Referee)**
bgoehrin@tulane.edu
*General Comments Diaz et al. present a compelling study showing the utility of combining*
*measurements of meteoric 10Be with soluble nitrate as a means to determine*
*surface exposure ages. In this case, they apply their new method to soils adjacent*
*to Shackleton Glacier, Antarctica. However, their new methodology, particularly the*
*combined use of nitrate and 10Be is not well-enough described. Additionally, and as*
*noted below, there needs to be a rigorous uncertainty analysis completed. All that*
*being said, I will very much enjoy seeing this paper published, but for now it needs*
*revision. The methods and results are interesting from an applied sense in that it could*
*be used elsewhere, but their work also adds to the glacial history of the Transantarctic*
*Mountains. Below I present general comments and then further below I present a*
*number of detailed comments and suggest changes.*
As detailed in our response to Referee #2, we believe that the narrative re-framing to include wetting
history and biogeography has significantly improved the manuscript. We have greatly expanded our
$NO_3^-$ and meteoric $^{10}Be$ methodology in Sections 2.2 and 5.3.2.
*The one supplementary figure showing the relationship between max 10Be concentration and*
*total 10Be inventory should not be buried in the supplement.*
This is now included in the main text as Figure 7.
*I find that the introduction reads too much like a thesis introduction. All of the content is very*
*good, but I think it could use a bit of streamlining that will help motivate the rest of the paper a*
*bit better, as I think you need to also address the limitations of in situ exposure dating, as you*
*mention later on, but it could benefit from being a bit earlier.*
As per Referee #2's proposed manuscript structure, we have re-framed and rewritten the introduction to
focus on the original goals behind collecting and interpreting these data – to understand relative surface
soil ages for biological survey purposes.
*Bear in mind this is purely a stylistic opinion can certainly be ignored. Throughout the*
*manuscript, anywhere there is a reference to an age, rather than a duration, need to use Ma*
*instead of Myr.*
We have made these changes to follow discipline formatting.

*There is overall a lack of uncertainty analysis that needs to be completed, particularly exploring*
*the sensitivity of your various age determination models to parameter variance. The*
*measurement uncertainties in this case are tiny compared to other uncertainties. A*
*full error analysis will greatly strengthen the conclusions made in the paper and really*
*needs to be done before publication. A bootstrap approach should be sufficient.*
The models that we have used in this work have been described and tested in great detail in previous
studies, which include sensitivity analyses (e.g. Willenbring and von Blanckenburg, 2010; Graly et al.,
2010). In general, the exposure age estimates using equations 1-6 are particularly sensitive to erosion,
deposition rates, and inheritance. Since these values could not be determined for each sampling
location, we chose to refer to our ages in a relative framework. Additionally, our ages are not absolute
due to the inability to correct for initial inventory and/or inheritance. These uncertainties are further
described in Sections 4.3.1 and 5.3.
*There is far too much framing of the study around Pliocene glacier dynamics, and particularly*
*the Sirius formation. I'd much prefer to see the expansion of the possible newish*
*and important approach that can be implemented combining 10Be with nitrate as a*
*measure of surface exposure duration.*
We have shifted the focus away from Pliocene glacier dynamics towards the description and application
of our analyses. We now focus on estimating surface exposure ages and the use of atmospherically
derived salts in estimating wetting history and exposure ages. This is detailed in Sections 1 and 2.
*Figure 8 demonstrates very nicely a coherent pattern of ice thinning/retreat. This needs to be*
*played up, and the return late in the manuscript to the Sirius Group detracts from the novelness*
*of the work.*
We now focus on our novel approach to estimating relative exposure ages and how these data contribute
to our understanding of ecological succession and glacier change.
**Detailed Comments**
*Line 37: Please provide a citation or two for the first part of the sentence.*
*There is actually quite sparse direct evidence for smaller interglacial extents relative*
*to the Holocene and much is largely inferred from distal evidence or modeling. Additionally,*
*the Ross Embayment is a large area and thus this statement is somewhat vague.*
We have better clarified our introduction on glacial history.
*Line 51: How are calculated and estimated exposure ages any different from*
*each other? I know this seems nit-picky, but it is somewhat strange wording as your*
*estimated exposure age had to be calculated first.*

We have explicitly defined our terminology in Sections 2.3 and 5.3. In brief, "measured" ages are ages we calculated based on the meteoric $^{10}$Be profiles we measured (Robert Massif, Bennett Platform, and Thanksgiving Valley); "estimated" ages are those we calculated based on the estimated $^{10}$Be concentrations from the power-law relationship between $NO_3^-$ and $^{10}$Be; "inferred" ages are those we calculated based on the inferred relationship between maximum $^{10}$Be concentration and inventory.

*Line 62: Unsure what "these studies" are. Are you referring to those cited at the end of the sentence or the sentence prior? If the sentence prior, why do you have a new set of citations?*

*Section 2.1 Should be worked more into the introduction in my view.*

We have clarified the introduction.

*Line 78: Nishiizumi et al., 2007 is not actually a half-life study, an outcome of the standardization is that a different half-life than had been used must be used. Recommend citing: ǎˊc Korschinek, G., Bergmaier, A., Faestermann, T., Gerstmann, U., Knie, K., Rugel, G., Wallner, A., Dillmann, I., Dollinger, G., Gostomski, C., Gostomski, C., Kossert, K., Maiti, M., Poutivtsev, M., Remmert, A. (2010). A new value for the half-life of 10Be by Heavy-Ion Elastic Recoil Detection and liquid scintillation counting Nuclear Instruments & Methods In Physics Research Section B-Beam Interactions With Materials And Atoms 268(2), 187 - 191. https://dx.doi.org/10.1016/j.nimb.2009.09.020 ǎˊ ´c Chmeleff, J., Blanckenburg, F., Blanckenburg, F., Kossert, K., Jakob, D. (2010). Determination of the 10Be halflife by multicollector ICP-MS and liquid scintillation counting Nuclear Instruments & Methods In Physics Research Section B-Beam Interactions With Materials And Atoms 268(2), 192 - 199. https://dx.doi.org/10.1016/j.nimb.2009.09.012*

We thank Dr. Goehring for the reference and have updated our citation.

*Line 101: Given the general absence of anything resembling soils or till in most of Antarctica, one could argue that applying meteoric 10Be is far more spatially limited, e.g. to regions of the Dry Valley, for example. Thus, I am not sure I would argue for your method by arguing that in situ exposure dating is limited, but instead argue that they are complementary.*

We have revised this point in Section 2.1.

*Starting line 107: I am not sure the bedrock lithology is all that relevant. I understand you want to show the protolith for weathering products, but I think it could be said more concisely. I think the geologic setting paragraphs could be combined.*

As Dr. Goehring has mentioned, the lithology is important in understanding weathering products and
material source. We believe this information is particularly important now that we have added more
detail on the sample site descriptions in Table 2.
*Line 123: Suggest changing "glacial dynamics" to "glaciers"*
We have changed this terminology throughout.
*Line 128: By two samples, do you mean two surface samples? Suggest clarifying the text here,*
*especially since you have depth profiles samples from elsewhere.*
*Line 130: In your reference to sample distance from*
*the glacier, are you largely referring to further away as controlled by elevation, or by*
*horizontal distance? I think some clarification of this could be useful, as depending on*
*the valley geometry, changes in ice thickness might not be significantly further away*
*from the glacier, or vice versa. It might be more constructive and more generalizable*
*to perhaps say that two samples were collected, one adjacent to the glacier, characteristic*
*of times similar to the current extent and one further away representative of*
*significant changes in glacier size (larger). A useful column in your table and the way*
*most Antarctic glacier change is expressed is as change in ice thickness.*
We have clarified our sampling procedure and terminology throughout the text.
*Line 142: Why not report the fraction between 2mm and 425 microns? Was none present? Sand*
*usually extends to 2 mm.*
We thank Dr. Goehring for identifying this error and have corrected the text and figures.
*Line 170: Suggest not starting paragraph with "However. . .."*
*I suggest that when laying out your calculation methods, that the equations flow more*
*within the paragraph, rather than being at the end of each paragraph. I found it somewhat*
*hard to ready.*
We have attempted to re-integrate the equations in Section 4.3.
*Line 179: Suggest adding "any" before "have meteoric" Line 197:*
*Delete "which"*
We have clarified the text.
*Line 202: Confused because didn't you calculate two samples from every location, only profiles*
*from only a few?*

We measured meteoric $^{10}$Be and $NO_3^-$ concentrations from at least two samples (generally near glacier
and furthest away) at all sites. We measured one profile at each site for nitrate and profiles from Roberts
Massif, Bennett Platform, and Thanksgiving Valley for $^{10}$Be. We have clarified this throughout the text.
*Line 206: The lack of an expected concentration*
*based on regressions against distance and elevation might just be spurious*
*and making predictions from these regressions very tenuous. I suggest removing this*
*sentence.*
We have removed this portion and the associated figure (formally Figure 5).
*Line 222: The ages are not necessarily minimum ages, as while you may*
*be overcorrecting for inheritance because you don't know the background inventory,*
*you also do not a priori know the erosion rates of the soils, even though you make*
*assumptions. I suggest that rather than couching the ages as minimum, as they are*
*only minimum relative to your max limiting no inheritance ages, you just present them*
*as best estimate given knowledge of the parameters.*
Our inheritance corrections in the original text were estimates since our depth profile concentrations of
$^{10}$Be did not reach background levels; we could not assess whether they were accurate. As a result, we
have redone the calculations to reflect ages with and without erosion (as erosion from boulders was
used). The ages are now reflective of maximum ages with the erosion term and are probably still
overestimates without erosion. See Section 5.3.1 lines 290-293 and Section 6.1 line 351.
*Section 5.3.1 This section is very*
*confusing in terms of what you did and is not represented in the methods at all, thus the*
*results presented here come out of nowhere. There needs to be a clearer explanation*
*of what was done. I think the approach is really neat and valuable, but right now it just*
*isn't explained well-enough. I am also very confused upon the first and second read as*
*to what was done with what profile, as the second paragraph mixes results from sites*
*with both measurements and sites without. Section 5.3.2 Like the prior section, where*
*there are a number of inferred methodological requirements, more expansion of the*
*discussion is needed to aid the reader that may only have casual knowledge of meteoric*
*10Be knowledge as I can see many readers being most interested in the inferred*
*ice history. I think one thing that will help immensely is that this and the prior section*
*are more traditionally considered as part of the discussion and the results purely your*
*10Be and NO3- measurements. Now, if you were to present the calculation methods*
*using nitrate and the inventory vs max concentration analyses in the methods, then*
*you could keep in the results. At present, there is just a bit too much mixing and overall*
*not enough time dedicated to these important sections that you then use extensively*
*in the discussion below. Also, best I can tell Figure 8 does not show the relationship*

*between max concentration and total inventory, please investigate, or do you mean to*
*only present the max exposure ages.*
We have more clearly defined our methodology for $NO_3^-$ and $^{10}Be$ in Sections 2.1-2.3, 4.3, and 5.3. As
stated earlier in this review, we have explicitly defined our terminology for the different exposure age
estimates.
*Line 247: Please elaborate or define what the*
*model limits are, as this is not defined. Presumably just the influence of the time scale*
*to 10Be saturation given an erosion rate. I also wish there were different terminologies*
*used with regards to calculated vs estimated. Perhaps refer to one as the apparent*
*max limiting age and the other a model age?*
We mention that the maximum age the model can calculate is >6 Ma. The model limits are dependent
on erosion and initial inventory, as described in 4.3.1. We have also more clearly defined our
terminology.
*Line 260: The correspondence with in*
*situ ages is quite remarkable. What is lacking though is a clear representation of the*
*two different data sets. This is why I suggested that perhaps you determine the elevation*
*above modern ice surface and thus you can then make age vs elevation plots*
*for your data and the in situ data. I think will drive home much more clearly the*
*correspondence.*
*Or you could consider maps showing the various bits of data, but I think*
*they will get very busy very quickly. While the correspondence in many scenarios is*
*striking, one thing to consider and make sure you make clear is whether the in situ data*
*are from bedrock or from erratics, as they will have quite different exposure ages and*
*thus your soil ages might always be older than nearby in situ erratic exposure ages.*
*The fact that your meteoric ages, including nitrate corrected, agree so much with in*
*situ erratic ages suggests some mechanism for resetting and flushing of 10Be or that*
*your model is determining the pre-LGM inherited concentration quite clearly. I think*
*this needs further discussion and is important to highlight more.*
We agree that we needed to represent the data comparisons in a clearer manner. We have added two
figures, 8 and 9, that show how our data compare to those from previous studies.
*Line 272: Need a reference for exposure dating results from Beardmore Glacier.*
We have removed the discussion from the Beardmore since we are unsure if we are sampling
comparable drifts or features.
*Line 288: The arguments about the suitability seem out of place*

*and kind of come out of nowhere and seem to set up a strawman for no apparent rea-*
*son. I suggest removing and focusing on the apparent success of the nitrate correction*
*given the good agreement with in situ exposure dating.*
We have focused the discussion on the shapes of the $NO_3^-$ and $^{10}$Be depth profiles, the age estimates,
and implications for climate and glacial advance and retreat.
*Starting line 292: The first few sentences of this paragraph read too much like a conclusions*
*section. Suggest revision.*
We have revised this section, now Section 6.2.
*Line 303: As mentioned above, the nitrate regression models needs further*
*description and elaboration, particularly since this really is the first major combined use*
*of these two measures.*
We have elaborated the $NO_3^-$ regression in Sections 5.2 and 5.3.2.
*Line 306: Wouldn't a lack of correlation be expected given the*
*exponential fall off of a 10Be profiles, so that below a certain depth there will be little*
*to no variance in the 10Be concentration and presumably the same in nitrate?*
We have clarified this point in Sections 5.2 and 6.3
*Line 352: Suggest rather than saying delayed response that you*
*more generalize it and just say different response from Ross Ice Shelf confluent outlet*
*glaciers, or something to that effect.*
We have removed this text.
*Line 358: This conclusion is spot on and is a major*
*finding of the paper, however its use, the details, etc. are not elaborated on enough*
*earlier in the manuscript.*
As stated previously, we have elaborated on the relationship between $NO_3^-$ and $^{10}$Be throughout the text,
especially in Section 6.2.
*Line 365: The broader question then becomes, how do we*
*differentiate between a site with inherited meteoric 10Be that was covered by LGM ice*
*from a site that was never covered during the LGM and more recent glaciations. This*
*is a question that the in situ community has struggled with. We are only starting to get*
*clarity from a focus on erratic exposure dating with long-lived nuclides or application of*

*in situ 14C to erratics and bedrock. Recent work in the Weddell Embayment with very*
*old erratic and bedrock in situ ages were clearly covered by LGM ice as shown by in*
*situ 14C, including preservation of delicate features like moraines (e.g., Nichols et al.,*
*2019). Thus, during a say 10 kyr long ice cover period, how much of a reduction in the*
*meteoric 10Be signal can be expected? What about reduction in nitrate? Presumably*
*unless the ice is wet based, neither will be mobilized and then you need the correct*
*pH conditions. These thoughts are briefly touched on, but the manuscript could use*
*a bit more elaboration on the long-term interpretation of the signal recorded by your*
*methods and what its implications are for interpreting surface processes in Antarctica.*
*Thus, it could be useful to elaborate on the presence of polythermal moraines, why are*
*some areas reset for the meteoric and in situ methods.*
Dr. Goehring brings up some very important questions. However, the answers to many of these
questions are unknown. Since we do not initially know which, if any, sites were disturbed by repeated
glaciations, we cannot correct for inheritance. Additionally, our profiles could not reach background
concentrations of $^{10}$Be for an initial inventory correction. We can only rely on the data we've collected.
Due to uncertainties with sediment transport, both modern and in the past, it is unclear how meteoric
$^{10}$Be and $NO_3^-$ would be affected over extended periods of time. We mentioned in the text (e.g. Lines
440-451) that some locations may actually be accumulating particles with $^{10}$Be instead of erosion.
However, under persistent arid conditions, we expect both $^{10}$Be and $NO_3^-$ to be largely conserved. We
have described this throughout the text.
*Figure 1: Not sure if this is supposed to be this way of if some strange PDF artifact, but the*
*exposed rock areas are banded. I also think you could make the overview map larger scale to*
*give readers a better context of the Shackleton Glacier.*
The exposed rock areas where we samples are indeed banded, hashed, and checkered in the figure to
indicate lithology as per the key. We made the overview map larger.
*Figure 3: A similar figure thinking about the fate of nitrate during ice cover would be*
*informative.*
We hope that the expanded text will suffice instead.
*Figure 4: Add panel labels please. Also, it is confusing that in the Shackleton glacier map, the*
*coloring represents concentration, but you then use the same colors for the different sites, or is*
*it only the arrows? This is somewhat confusing, and I suggest not using colored arrows that are*
*the same as the color scaled points for concentration. Here the figure is trying to show*
*too much.*
We have updated this figure.

   *Figure 5: This figure and all figures. Are uncertainties shown, but smaller*
   *than the symbol? Please note this or add uncertainties if need be.*
Due to the log scale, the measurement uncertainties are small, as indicated in Table 3.
   *Figure 6: Suggest removing the lines connecting the points, as it implies that there is a trend in*
   *grain size % between the points. The measurements are point measurements.*
We have kept the lines the help the reader connect the points.
   *Figure 7c: Please provide equations for the fits along with uncertainties on the fit parameters.*
   *These uncertainties then need to be used for error analysis on the resulting ages.*
We have removed the regressions.
   *Table 2: I suggest presenting uncertainties using the same exponent for the measured value and*
   *Uncertainty.*
Normally we would agree, but if we change to the same exponent, there are too many zeroes.

**Anonymous Referee #2**
**I. Summary.**
*The summary of this review is that the data collected in this paper are useful, interesting,*
*and valuable to publish. In general, the idea that accumulation of atmospheric*
*constituents in Antarctic soils is useful for estimating soil ages and residence times is*
*important from many perspectives, including glacier change, paleoclimate, and biology,*
*and this paper contains a lot of data that are relevant to this topic.*
**II. Overall motivation of paper.**
*II.1. The way the paper is motivated makes the experimental design look bad when, in*
*fact, it is not.*
*The experimental design of this study is very well designed from the perspective of*
*a biological survey. The use of atmospheric fallout constituents of soils to rapidly*
*get an approximate idea of the soil age, and distinguish soils that were ice-covered*
*during the LGM from soils that have not been ice-covered for millions of years, is a*
*smart, well-designed approach that is likely to be effective for its intended purpose. On*
*the other hand, the study is not well designed for the purpose of reconstructing past*
*glacier change.*
*The point here is that if the present study was motivated by the original objectives of*
*collecting geological information needed to study ecosystem succession, it would be*
*perceived by readers as well-conceived and well-designed. If motivated as a study of*
*glacier change as in this paper, on the other hand, the experimental design appears*
*weak and inadequate by comparison to other studies.*
*I very strongly urge the authors to change this emphasis. They should clearly explain*
*the purpose of the overall project that led them to the experimental design used here.*
*It is true that the data collected for this purpose also have value in quantifying glacier*
*change, so there is nothing wrong with focusing additional discussion on that later in*
*the paper, but motivating the entire paper from this perspective makes the paper much*
*weaker than it should be.*
Referee #2 is indeed correct that the samples collected for this study and for this analysis were for a
larger study on ecosystem succession following changes in climate – in this case, glacial advance and
retreat. The goal of this smaller study remains the same. We sought to determine relative surface
exposure ages of ice-free areas along the Shackleton Glacier. Though these data can be useful in
understanding glacial change, we agreed that the introduction and discussion should be refocused to
emphasize our broader goals and significance to ecological refugia. As such, much of these sections
have been rewritten to include these points.
*II.2. The way the paper is motivated leads the paper off into vague theories that can't*
*be addressed by the data.*
*The most problematic part of the paper from this perspective is the first two paragraphs*

 *of the introduction (lines 33-45) and section 2.1 ("Stability of the EAIS"), lines 55-76.*
*The introduction discusses the fact that the Antarctic ice sheets are proposed to have*
*been a lot smaller during some warm periods in the past. While it is certainly true that*
*this has been hypothesized and that in a very general sense this is a strong motivation*
*for studying past changes in the size of the Antarctic ice sheets, there is almost no connection*
*between this overall idea and the specific observations described in this paper.*
*As discussed above, if this is the motivation for the work, the work looks inadequate.*

*Section 2.1 is much more problematic.*
*It would be clearer to simply state that it is not yet known whether or not the East*
*Antarctic Ice Sheet was significantly smaller during past warm climates. The second*
*problem in this section has to do with confusion between ice sheet change and climate*
*change.*
*The discussion of how long polar desert conditions have prevailed in the TAM is important in*
*this paper because it gives context for one potential application of salt deposition in soils, i.e.*
*the idea of a "wetting age" in which the amount of salt that has accumulated can give*
*information on when liquid water was last present. However, this important implication*
*of the idea is not at all mentioned here.*

We have changed the focus of the introduction to discuss ecological dispersal and refugia during glacial periods, the overall glacial and climate history of Antarctica, the need to understand exposure ages in this region, the goals of this study to understand soil ages, and the applications both to ecology and geomorphology. We have removed much of the text on East Antarctic Ice Sheet stability and instead shift the focus to persistent arid conditions, as the desert climate is particularly important for salt accumulation and the development of our $NO_3^-$ proxy.

**III. Oversimplified explanation of atmospherically produced Be-10.**

*With regard to section 2.2, the main thing the authors need to get across here is that*
*meteoric Be-10 builds up in soils, so the total amount of Be-10 present in a soil profile is*
*related to the age of the soil. This information is here, but it is missing some important*
*context and mixed up with other confusing things. One, the*
*authors should clearly state that meteoric Be-10 is mobile in the soil, so it is not the*
*concentration at any particular location that is proportional to the exposure age, but*
*instead the total inventory in the entire soil profile. Two, the behaviour of meteoric Be-*
*10 and salts in soils may be quite different, for example because Be-10 remains bound*
*to particles even when the soil is wet, whereas salts are mostly mobile in water.*

While we do discuss meteoric [10]Be systematics later in the text, we agreed that it would be beneficial to better describe the system in more detail here and have done so. We also added in Section 2.2 on $NO_3^-$ systematics.

   *The other important area here that needs to be either here or in the section on study*
   *sites is a discussion of exactly what landforms were sampled and how that relates*
   *to meteoric Be-10 systematics.*
We have added Table 2, which describes the landforms and features we sampled at each location and
any notable features, such as nearby ponds, polygonal ground, etc. Mapped geomorphologic features,
such as drifts and moraines, are poorly documented in this region. The only published data are from
Roberts Massif and Bennett Platform. We made sure to mention any constructional landforms in Table
2.
   *Section 4.3 is about how to quantitatively interpret Be-10 concentrations as an exposure*
   *age of the soil. This section would benefit from several improvements. Specifically,*
   *Equation (1) seems to be missing important elements.*
   *A common approach in the meteoric Be-10 literature to simplify this relationship and*
   *make it more useful is to write the governing equation for the soil inventory I (atoms*
   *per cm2, vertically integrated) instead of the concentration, like:*
   *dI/dt = Q − _I − ENs (2)*
   *where Ns is the surface concentration (atoms/g) and E is the erosion rate in mass per*
   *area units. Using this equation instead of Equation (1) would make this paper much clearer.*
   *Alternatively, this paper could simply refer to other literature that describes meteoric Be-10*
   *systematics in detail – it is not necessary to reinvent the wheel here.*
We understand that the simplicity of Eq. 1 was misleading. We have removed the equation and replaced
it with a more comprehensive equation, per Referee #2's suggestions.
   *Finally, an important point for these sites is that it is not even clear that erosion is*
   *taking place throughout the ice-free at areas all. Perhaps the only process that*
   *can bring new sediment to the surface and permit deflation would be periglacial disturbance*
   *of the soil. This issue reminds me that an important thing that needs to be*
   *added to section 3 is some discussion of the surface characteristics of each site, including*
   *presence or absence of boulder pavements and periglacial features like cracks*
   *and polygons, because these features are relevant to interpreting the Be-10 data.*
   *The overall point of this section is that it is not at all clear to me*
   *that erosion should even be included in the relationship between inventory and age for*
   *these sites. For this paper, I think it might make the most sense to simply relate inventory*
   *to exposure age by dI/dt = Q − _I, i.e. disregarding erosion and deposition, and*
   *accept that this approach might be either under- or over-estimating exposure ages.*
As mentioned previously, we added a table describing the surface features of each sample location,
including whether the samples were collected on valley floors or hillslopes. While we did not sample features such as polygons and boulder pavements, it is crucial to indicate such. With the sample
locations further described, we believe the inclusion of erosion rates is clearer, though we acknowledge
that true soil erosion rates are unknown for these features. We also calculated the exposure ages without
erosion and report both throughout the text.
*[T]his section has to clearly explain how one measures the Be-10 inventory. As*
*already discussed in the paper, this can be done in two ways, either by measuring a*
*complete depth profile and integrating, or using an empirical relation between surface*
*concentration and inventory as in the Graly paper.*
*An additional problem with this section is that "inheritance" is not clearly defined, which*
*is confusing.*
*Finally, a clear definition of "background" in the context of a depth profile is needed*
*here. The basic concept (that the concentration is supposed to decrease with depth*
*until you reach a depth where the concentration becomes invariant with depth) is correctly*
*described near line 182, but what is missing is a clear statement of how one*
*knows that one has observed this. Overall, what I suggest doing here is*
*noting that in principle the depth profile method is one possible way to estimate I, but*
*it can't be used in this application because insufficient data were collected – and then*
*move on to discussing the approach of using an empirical correlation between N and*
*I to estimate I.*
We have clearly defined both inheritance and background (initial inventory) in the context of our study
on Lines 203-217. We also note that we are unable to correct for either of the two with the data we
measured. Referee #2 correctly mentions that we have not satisfied the typically criteria for attaining
background measurements of meteoric $^{10}$Be. We have communicated this in the text.
**IV. Data analysis.**
*I did not understand what the purpose of these regressions is [Fig. 5].*
*Because I don't see any basic physical relationship that would support linear regression*
*of concentration against elevation/distance, as a reader I am left with the impression*
*that the authors simply felt that there should be some linear regressions in the paper. I*
*am not sure this is the impression that the authors want to give the reader. It makes the*
*paper seem weak and confused, and I urge them to remove this section of the paper.*
We have removed this figure and the associated text.
*The second area that seems problematic to me in this section of the paper is how the*
*authors approach estimating the Be-10 inventories in section 5.2.*
*What I suggest doing here is removing section 5.2, noting that the depth profile data*
*do not allow estimating I accurately, and rely entirely on the empirical-correlationbetween-*
*I-and-N approach for estimating I, which is already clearly covered in section*

	*5.3.2. This is not really a major substantive change to the paper, because at most of*
*the sites there are only surface data in any case.*

We have decided to keep the inventory calculations for Roberts Massif, Bennett Platform, and
Thanksgiving Valley since we have [10]Be depth profiles for these locations. We also calculated inventory
using the $NO_3^-$ and [10]Be power-law relationship (Sections 4.3 and 5.3). We did not attempt correct for
initial inventory (background) or inheritance. This is stated in those sections. We mainly kept the I and
N calculations as written.

*The third area that I think needs additional discussion in this section is the discussion*
*of the relation between Be-10 and nitrate concentrations. To summarize, this*
*section needs to be made much more clear so that the reader can understand when*
*concentrations, surface concentrations, and inventories are being discussed, and what*
*differences in behaviour of Be and NO3 could lead to positive or negative correlation.*
*This may require making this section substantially longer in order to explain the reasoning step*
*by step so that the reader can follow it.*

We agree with Referee #2 that this section can and should be greatly expanded upon. We added
additional text describing the relationship between [10]Be and $NO_3^-$ for each of the three soil profiles and
the factors which have likely contributed to the observed concentration behavior in Sections 5.2, 5.3.2,
6.2, and 6.3.

**V. Discussion and interpretation areas.**

*The first aspect of the discussion that needs additional work is that the most basic prediction of*
*the experimental design is that, first, Be-10 inventories and/or concentrations*
*should increase with distance from the ice margin at each site, and, second, Be-10*
*inventories/concentrations for the ice-proximal samples that are supposed to have been*
*exposed after the LGM should have magnitudes that are appropriate to post-LGM exposure, i.e.*
*10-15,000 years of surface exposure.*
*I would do this with a figure for each site showing distance from the*
*nearest ice margin on the x-axis, and Be-10 and NO3 concentrations on the y-axis.*

We agree that an additional figure showing [10]Be concentration versus distance from glacier would be
beneficial in supporting the overall experimental design. However, some samples were collected on
ridges and we would only be able to estimate aerial distance, which is not very helpful from a
glaciological context. Instead, we have added Figures 8 and 9 which show the ages with elevation and
on maps.

*The second aspect of the discussion that is incomplete/too abbreviated is the section*
*beginning on line 260 that compares the results to existing exposure-age data from*
*glacially transported boulders. Personally, what I would view as minimally adequate*

*here is a map view of each site where there are existing/published exposure age data,*
*showing the location of the soil pits described here, the location of any moraines or*
*drift boundaries including any hypothesized LGM ice limit, and also the location of the*
*independent exposure-age data, which will be mostly boulders dated by some in-situ produced*
*nuclide. Alternatively, instead of maps, these could take the form of plots*
*with distance from the ice margin on the x-axis, and exposure ages calculated from the*
*various data on the y-axis.*

*A second issue here is that some of the other exposure-age data (e.g., Thanksgiving*
*Point, Mt. Franke) appear to be available in online databases but not yet published in*
*journal articles. I am sure the data are fine, but this may cause some citation problems.*
*I refer that issue to the editors.*

Though there are only published data from Roberts Massif, we agree that it is helpful to plot the *in-situ*
data from previous studies and ICE-D alongside our data to support our comparisons. This is done in
Figures 8 and 9. Confident estimates of the LGM trimline and mapped drifts for the other sites and
features we sampled in the Shackleton Glacier region do not currently exist. Regarding the citations, we
cite Balco, 2020, which includes the ICE-D dataset.

*In addition, some of the text in this section gives the impression that the authors have*
*a misunderstanding of the existing exposure-age data set. For example, consider the*
*remark in line 273-ish about exposure ages from the Beardmore Glacier region, which*
*states that exposure ages become younger downglacier for Shackleton and Beardmore*
*Glaciers. In principle, it is possible that pre-LGM deposits are*
*less common at low elevations, but that would have to be established via systematic*
*mapping of these deposits. Thus, this section of the paper needs to be significantly*
*reworked to focus on a comparison between specific mapped deposits of known or*
*estimated ages, and not on a broad geographic analysis of a set of ages that is probably*
*the result of selection bias.*

Considering the concerns Referee #2 raised regarding this section, we decided to largely remove it.

*The third aspect of this part of the review is that I could not understand the paragraph*
*in lines 292-302. This mixes observations that the relationship between Be-10 and*
*NO3 concentrations in depth profiles is complicated (which is true) with statements that*
*have no clear connection to this observation such as "through a coupled approach...we*
*developed a useful model for estimating soil exposure ages."*
*I suggest starting again with this paragraph and trying to lead more clearly from observations*
*to conclusions.*

With the overall manuscript reframing and editing of the discussion, clarity has improved throughout. In
particular, we outline our methodology in Sections 4.3, 5.2, and 5.3.2.

*Finally, the last important thing here is that I found the disconnect between observations*
*and conclusions to be most serious in section 6.3 ('Implications for ice sheet*
*dynamics.'). This section contains several very broad statements. Only one of them*
*(the discussion of the Sirius Fm.) is clearly related to the observations.*
*The other conclusions here are not related to the observations, and I think this area of*
*the paper needs work. For example, "Our data support models...suggesting that EAIS*
*advance and retreat was not synchronous..." (line 321). The fact that higher-Be-10*
*concentration soils are only found at more inland sites only shows that the authors were*
*able to locate older deposits at inland sites, but did not find them at lower-elevation*
*sites.*
*The discussion around line 333 also appears oversimplified and to not take into account*
*basic glaciological principles. To conclude that one site has a younger exposure age than*
*another should involve showing that the difference between measured concentrations*
*is significantly larger than we expect based on the scatter of the data used in the*
*concentration-inventory transfer function. My overall point is that the oversimplified*
*nature of this discussion gives the impression that the authors have not thought very*
*hard about this. To get from the actual observations in this paper to a conclusion*
*about glacier change, I would expect to the following steps: first, clearly describe, map,*
*and identify glacial deposits that have been sampled; second, show whether or not*
*samples from the same deposits are the same age, and then, third, conclude whether*
*or not each mapped deposit is synchronous or time-transgressive. Many of these steps*
*are absent here.*

These are all very valid points. Given the other suggestions and changes throughout the manuscript, the
revisions we made rectified much of Referee #2's concerns. Instead of focusing on EAIS behavior, the
revised manuscript focuses on the relationship between meteoric $^{10}$Be and $NO_3^-$ concentrations to
estimate relative ages and understand landscape disturbance from wetting events. Since there are few, if
any, data from many of the ice-free areas we sampled, we believe our data and measurements are
important and have emphasized this. Additionally, by focusing on smaller-scale processes, we were able
to make inferences regarding arid conditions in the CTAM (see Section 6.3). As we and Referee #2
point out, the shape of $NO_3^-$ and $^{10}$Be profiles should appear similar in persistent arid conditions since
both constituents are atmospherically derived. Deviations from this expected relationship can indicate
wetting or $^{10}$Be erosion/deposition, which have particularly important implications for ecological
succession. These points primarily constitute the discussion and conclusions.

**VI. Suggested reorganization.**
*This section makes some suggestions for how I would rewrite this paper to make it*
*better. Mainly, I suggest significantly simplifying the paper, focusing much more on the*
*data that were actually collected in this study and not on broader topics that may seem*
*more important but lack a clear relation to the data, and also being much more clear*
*on the chain of reasoning between observations and conclusions. I suggest an outline*

*that looks like the following:*

*1. Begin the paper by describing why the study was designed and conducted in the*
*way that it was – as a means of estimating surface age for biological survey purposes*
*– and then pointing out that the purpose of this paper is to describe the soil age data,*
*which may also be useful for understanding geomorphology and glacier change in this*
*area. I would remove the claim in the introduction that these data are likely to provide*
*significant information as to the stability of the Antarctic ice sheets in warm periods.*
*2. Describe the sample sites and the approach of sampling a likely-post-LGM and*
*likely-pre-LGM site in each area. Discuss in detail the physical and geomorphic characteristics*
*of the site as well as any evidence for the mode of deposition of the parent*
*material and also whether the soil is inflationary or deflationary.*
*3. Explain how meteoric Be-10 in soils works in a way that is simpler and clearer than*
*it is in the present paper, by removing Equation 1 and focusing on the relationship*
*between inventory and age and the need to relate concentration to inventory to make*
*an estimate of the age from one surface sample. Explain both ways of relating N to I.*
*Be clear about what "inheritance" is.*
*4. Explain the expected relationship between Be-10 and NO3.*
*5. In the data analysis section, begin by establishing whether the basic premises of*
*the study (ice-distal sites should have more Be-10, and LGM-age sites should have*
*the amount of Be-10 expected to have accumulated since the LGM) are true. Note*
*that the depth profile data are not adequate to estimate background concentrations,*
*and remove this section of the discussion. After addressing the basic validation of*
*the approach, move on to secondary questions such as whether presumed LGM-age*
*sites have similar Be-10/NO3 inventories up and down the glacier, and differences in*
*Be-10/NO3 inventories among pre-LGM sites.*
*6. Convert concentrations to exposure ages and compare these to the expected distribution*
*of LGM deposits as well as other exposure age data for the sites where there*
*are some data. Use maps of these sites to clearly show the geographic relationship*
*between your and other data.*
*7. With regard to the implications of these results for larger-scale issues having to*
*do with ice sheet change during warm periods, I don't think the exposure age aspect*
*of these results significantly changes the overall picture that previous research has*
*derived from the existing several thousand exposure ages from Antarctica. On the*
*other hand, the idea that salt accumulations can give some information on past warm*
*climates (was it warm enough for liquid water to be present in soils, and if so, when?)*
*could be very significant. Unfortunately, there is very little discussion of this in the*
*paper. From first principles, I would expect NO3 and Be-10 to be correlated in dry*
*soils, because both would accumulate and not be removed. But as soon as water*
*is present and leaching of NO3 can occur, one would expect a lack of correlation.*
*Thus, the relationship between these two soil age proxies could be quite valuable for*
*paleoclimate. I would give this more attention in a revised paper.*
*In general, in rewriting this paper, I very strongly urge the authors to focus much more*

*on the specific things that they measured and observed.*
We are grateful to Referee #2 for such a detailed review and have used their suggested organization as a
guide for our revisions.
**VII. Minor comments, by line number.**
*Line 37 (The WAIS has been drastically reduced in size) and line 52 (A growing body of*
*work that suggests...susceptible....). These areas incompletely describe the evidence*
*for ice sheet change during warm periods. There exist model simulations that show*
*that deglaciation of very large marine-based areas of the ice sheets is possible during*
*warm climates. These are not evidence, but hypotheses that the model simulations*
*show are physically possible. There is some indirect evidence (e.g., marine oxygen*
*isotope data) that, given several assumptions, may be consistent with this hypothesis,*
*but is also consistent with the hypothesis that minimal deglaciation occurred. There*
*is one piece of direct evidence (Be-10 in Siple Coast subglacial till; see Scherer and*
*others) showing that the WAIS was smaller by an unknown amount sometime during*
*the later Pleistocene. There is no direct evidence that hypothetical collapses simulated*
*by ice sheet models took place. In fact, the best effort so far to test this hypothesis by*
*subglacial bedrock recovery drilling in West Antarctica (Stone and others, recent WAIS*
*meeting abstracts describing bedrock recovery drilling at Pirrit Hills) did not show any*
*evidence for WAIS collapse. Thus, ice sheet collapses during warm periods need to*
*be presented as a hypothesis and not as an accepted fact.*
*Note that the text around line 75 is much more clear in this regard and correctly distinguishes*
*evidence and model predictions.*
We have considered these comments and made the changes in our introduction.
*Near Line 100 . The authors should not mix up evidence for sustained aridity in icefree*
*areas with evidence for changes in the size of the ice sheet. Aridity does not*
*necessarily require a large ice sheet, and ice sheet collapses due to marine ice margin*
*instabilities could have occurred during cold, arid conditions. These two lines of*
*reasoning should be kept separate.*
We have separated these lines of reasoning in the introduction and discussion.
*Line 101-102. I did not understand these sentences.*
We have removed these sentences.
*Line 117. "High rates" is incorrect. Because this area is extremely arid by global*
*standards, salt is delivered at a very low rate when compared to normal places. What*
*is different here is not a high rate of supply but a low or zero rate of removal.*

We have made this correction.
*Line 122-3. This discussion gives the impression of not being well founded in glacial geological*
*observations. The critical difference between moraines deposited by frozenbased*
*and wet-based ice is not their size, but rather their sedimentology. I looked at*
*imagery of the Bennett Platform moraines and although they are large, they appear*
*to be mostly composed of large boulders. No evidence is given in this paper that*
*they include a fine-grained, matrix-supported till with striated clasts that would indicate*
*formation by wet-based ice. If the authors did observe this, they should certainly describe*
*it, with pictures, because matrix-supported tills near the ice margin in this region*
*would be very surprising. It seems more likely that these moraines are typical boulder*
*moraines deposited by frozen-based ice, and their anomalous size may simply be*
*related to the supply of boulders from large overhanging cliffs.*
We agree with Referee #2 and have made this correction (see Line 135).
*Line 140-ish. I think this could be stated more clearly simply by saying "We collected*
*surface samples at all sites and 3-sample depth profiles at three sites."*
We have clarified the sampling methodology.
*Line 198ish. Because the sites you are sampling are soils and not rocks, I don't think*
*these rock surface erosion rates are relevant. I suggest looking at papers by Dan*
*Morgan and Jaakko Putkonen about the Dry Valleys to get an idea of the expected*
*range for erosion rates of unconsolidated material. However, as noted above, most*
*of these data are from hillslopes (although not all) and it's very possible that sediment*
*deposition, rather than erosion, is taking place at some of the sites in the present paper.*
It is documented that ash layers and hillslopes have relatively high erosion rates, likely much higher
than the rates expected for soils in the CTAM. We do not think these erosion rates are applicable for the
Shackleton Glacier region as a whole. We are explicit in saying we are using a rock erosion rate, note
the limitations, and provide ages without erosion terms (Sections 4.3, 5.3, and 6.3).
*line 204. What is the "coast"? It appears that the "coast" here is where the glacier*
*flows into the ice shelf, but that makes very little sense in this context if one is thinking*
*of the ocean as the source of salts. Open ocean is much farther away.*
Coast in this context represents the point where the glacier is no longer constrained by the TAM and
flows into the ice shelf. We do not rely on distance to open ocean due to seasonal and yearly changes in
this distance from sea ice extent, and to be consistent with biological literature. We have clarified in the
text (Line 143).

*Line 269. The amount of time that soils are ice free must be longer for sites that are*
*farther away from the glacier simply because of geometry. The ice sheet cannot cover*
*more ice-distal sites unless it has already covered the ice-proximal sites. Thus, for any*
*ice advance-retreat history, ice-distal sites will always be exposed longer. My point is*
*that this is not a conclusion of the study (which is what this text sounds like), but it must*
*be true under any circumstances no matter what the results.*
We agree and have removed this statement from the conclusions.

[revised manuscript text omitted]

These data suggest that the samples furthest inland at Roberts Massif and Thanksgiving Valley have been undisturbed since at least the middle to late Pleistocene. Although meteoric $^{10}$Be and $NO_3^-$ at Bennett Platform are mirrored with a negative power-law slope, we argue that the difference is not due to $NO_3^-$ mobility, but instead $^{10}$Be deposition. Bennett Platform was the only location we sampled on a large moraine (Fig. 2c), and as such, we would expect minimal inheritance with $^{10}$Be decreasing at depth. This is generally the observed behavior, with significantly higher surface concentrations. The $NO_3^-$

profile behavior is similar to those throughout the Shackleton Glacier region, though the concentrations continue to increase with depth, possibly indicating minor percolation of $NO_3^-$ rich brine. What may be considered the "anomalous" data point is the surface concentration of meteoric $^{10}$Be. Even though we sampled a constructional landform, the sample was collected between two boulder lines in a small, local depression (~1 m) (Table 2). It is probably no coincidence that this location also has the greatest proportion of fine-grained material in the soil profile. The two boulder lines impede wind flow and act as a sediment and snow trap, resulting in a higher concentration of meteoric $^{10}$Be than expected simply from atmospheric deposition. In this case, an additional deposition term (superseding any erosion) needs to be considered to accurately date the moraine, and the current exposure age we measured is may be an overestimate.

While we were not able to identify and sample common drifts at each location for comparison, the youngest surfaces we sampled are those from the lowest elevations and closest to the Ross Ice Shelf (Fig. 10). This is generally consistent with pervious glacial modeling studies which show that the greatest fluctuations in glacier height during the LGM were along outlet glacier and ice shelf margins (Golledge et al., 2012; MacKintosh et al., 2011; Mackintosh et al., 2014). We have emphasized throughout this paper that erosion and inheritance/initial inventory could not be assessed in this study. However, erosion rates are low throughout Antarctica (Balter-Kennedy et al., 2020; Ivy-Ochs et al., 1995; Morgan et al., 2010) and would not drastically impact our relatively young inferred ages (Fig. 10). Additionally, background concentrations of meteoric $^{10}$Be in other Antarctic soil profiles are often approximately one to two order of magnitude lower than surface concentrations (Fig. 6). With these considerations, the Mt. Speed, Mt. Wasko, and Mt. Franke samples were all likely covered by the Shackleton Glacier during the LGM, as well as the lower elevation, closest to the glacier samples from Mt. Heekin, Bennett Platform, and Mt. Augustana may have also been covered. The samples we collected near the head of Shackleton Glacier encompass a range of ages, where lower elevation soils are relatively younger, though the soils from Schroeder Hill and Roberts Massif have likely been exposed since the early Pleistocene (Fig. 10).

Lastly, while we cannot directly evaluate the overall stability of the EAIS during changes in climate, 
[revised manuscript text omitted]

---

## Referee Report (RR1)

General Comments
I would first like to thank the authors on their extensive revisions. The manuscript is better in all regards, except for uncertainty analysis (see below). The aims/goals, results, and interpretations are all presented quite cleanly and generally easy to follow. All figures are improved and generally I have small comments and suggestions. While the manuscript reads better and is easier to follow there are still some issues of awkward wording in a number of places and the three varieties of calculation methods I think could be simplified. Finally, I noted numerous typos/grammatical issues. I think this is just a result of revision under a deadline.

There is still a lack of uncertainty analysis and quantification. This is a major issue in my mind. The authors presented results with and without erosion, and the differences are sometimes large, sometimes not; this screams for a full analysis of the sensitivity to erosion. You also rely on regressions to assess $^{10}$Be concentrations/inventories, any regression model has uncertainty and must be incorporated. The fact that uncertainties are discussed in other papers, means that those authors discuss the possible conceptual uncertainties in the application of the method. I was not referring to this, and rather referring to the resulting uncertainties on your results. I do agree that there is not a single number that results, but rather a distribution of values. I really do think this manuscript, and methodology, would benefit from this particularly since there is a new method, it relies on a regression, and all regressions by definition have some measure of uncertainty.

Detailed Comments
Line 20: Sentence starting this line seems incomplete. I feel it needs some sort of comparison to regions elsewhere. Also, the statement about largest changes in the TAM is not true for the entirety of Antarctica, but really only applicable to EAIS.
Line 25: How can you calculate a measured age? I think this sentences just needs to say very clearly what was done and remove the parenthetical aspects. Maybe rephrase as "We measured meteoric $^{10}$Be and NO3- concentrations to calculate exposure ages using the total $^{10}$Be inventory, the NO3 concentration, and infer exposure duration from the $^{10}$Be surface concentration."
Line 27: Swap lower and relatively
Line 29: Change to indicate
Line 51: All evidence points to WAIS and EAIS max extent not synchronous with the canonical LGM (26-19 ka) and likely were largest approx. 14ka.
Line 80: Delete "it"
Line 81: You is spelled Yiou
Line 87: Insert "of" after "measurement"
Line 116: Maybe instead of "measured" say "as measured". That being said, the explanation here is better than in the abstract. Can you try to clarify the abstract a bit more?
Line 163: Many readers wont know what UVM stands for, suggest spelling out even though totally inconsequential for the manuscript.
Line 182: replace the comma with and.

Line 185: Expressing E as length per time in the text and then adding with respect to density is confusing, as the function really works with mass depth per time for erosion, E. Consider revising or rewording the text.

Line 285: Delete "regressed"

Line 292: Suggest presenting model ages in same order, first no erosion and then with erosion as above.

Line 295: Here and throughout there are some inconsistencies with tense and passive voice. Strongly suggest going through and editing for this and some other grammatical issues I noticed. I tried to point out many of them but skipped over many.

Line 377: Delete "study" before second clause.

Line 426: Should be stable

Line 438: Insert "of" after "most"

Line 460: MacKintosh is the same person as Mackintosh. The latter is correct. I noticed this elsewhere and suggest fixing before copy editing.

Figure 8: The black triangles are hard to see (even with the outline) on the imagery as the rock is so dark, and combined with ribbons of snow makes the two hard to differentiate. The symbols would also greatly benefit from displaying the ages on the maps.

Figure 9: Age-elevations plots are usually presented as elevation on the y-axis and age on the x-axis. I suggest flipping axes.

Figure 10: Swap axes like in Figure 9 for Figure 10b. For figure 10b, since you are comparing age vs elevation and showing along the length of Shackelton Glacier, where elevations span 1000 m, I suggest presenting as elevations relative to the ice surface. This will remove the slope effect on the absolute elevation.

Table 3: Uncertainties here and for all other columns should be presented using the same exponential as the concentration, otherwise just adds confusion. I know this was mentioned in my first review, there would only be one leading zero for most of the samples. One other note is that PRIME Lab reports values at two decimal places, I suggest only reporting to this precision. There is unlikely a need to present non-background corrected ratios, they are so high that it is unlikely that many will be have any significant correction.

---

## Referee Report (RR2)

**Second review, Diaz and others, Esurf**
* * *
This paper is greatly improved from the first draft. I very much appreciate the authors' effort in seriously revising the paper. Most of the major problems from the first draft have been corrected. Nice job.

From the perspective of this review, though, what that means is that it is now possible to actually understand the scientific content of the paper and review it in detail. So in this review I've actually done that. The result is that, although the paper is really, massively, improved in many ways, there's still one problem, which is the discussion of the relationship between Be-10 and NO3 concentrations. This still has some elements that are misleading because they are oversimplified and in part appear to be incorrect. So that part isn't acceptable for publication yet and needs to be fixed. I discuss this in way too much detail later.

So, going through the paper in order,

1. Introduction. This is enormously improved. It is basically fine now, although it would be helpful to better orient the reader (who perhaps has skipped over the abstract) by beginning the introduction with a sentence describing the actual research in the study, like "This study reports concentrations of atmospheric fallout constituents in soils in the southern TAM. These data can be used to understand soil age and disturbance frequency, which are biogeographically important because one of the most intriguing questions..."

2. Background section. Also much improved. Great. Minor points:

In lines 98-99, do you mean lower elevations, like water is running downhill for a significant distance, or greater depths in the soil? Clarify.

In line 111. "Considerably fewer" sounds strange to the reader here because you are saying there are fewer studies in the CTAM than in NTAM/NVL, but you cite more studies for CTAM. I would just remove the "considerably fewer" and note that there have been scattered exposure-dating studies all over the Transantarctic Mountains.

3. Methods section. This is much better. Could use minor clarification in a couple of places, as follows:

Line 151. Perhaps clearer to say "...to represent soils likely to have been covered during the LGM and exposed by more recent ice margin retreat."

Line 189. The usage of 'dz' is mathematically strange here. dz is just a generic differential in depth, it is not a parameter needed to evaluate the equation, so it is unclear what 'highly dependent on dz' means. I think what the authors are trying to say here is that the concentration gradient dn/dz depends on what D is and also varies with time, and D is unknown, so it is not really possible to calculate dn/dz. In addition, in line 187, the authors use 'concentration gradient' to describe

d2n/dz2, which is confusing because 'gradient' usually means the first derivative of something (dn/dz). The second derivative (d2n/dz2) would typically be described as 'curvature.' In any case, both of these points give the impression of carelessness and this section needs to be carefully checked to make sure it makes mathematical sense.

Lines 211-12. This doesn't seem to make sense, because if Be-10 is supplied from the surface, the concentration has to decrease with depth at some point, no matter what. I believe the difference the authors are trying to point out is that in a normal soil one would expect a fairly smooth decrease, but in a periglacial soil one might expect a well-mixed active layer with constant concentration abruptly overlying a frozen layer with much lower concentration. However, this is not what this sentence says. In any case this sentence is oversimplified to the point of causing confusion, and it's not really very important, so I would remove it.

4. Results. The basic description of the results, up to section 3.2, is good. At this point, however, we get into the subject of the relationship between meteoric Be-10 and NO3 concentrations, which needs work.

Basically, the approach in this paper as far as I can tell is to assume that Be-10 and NO3 are both supplied by fallout, and both are supplied at a fairly constant rate, which in turn means that the ratio of their depositional fluxes is also fairly constant. Call this assumption 1. OK, this makes sense, and if this is true and the soil profile is a closed system (assumption 2), they can both be considered conservative tracers. So, up to now, assumptions 1 and 2 predict that total inventories of Be-10 and NO3 in the soils should both increase monotonically with exposure time and will be highly correlated.

Correlated inventories of Be-10 and NO3 that both increase monotonically with exposure age is, of course, what we expect, not only from first principles but due to the Graly study at Mt. Achernar, which showed these amazing correlations between exposure age and salt concentrations in sediments in a blue ice moraine. That is exactly how it is supposed to work and, like the authors, I want it to work that way in this study. Unfortunately it doesn't. I thought about this issue a lot in putting together this review – because, as noted, I would like this to work – but when you look at the actual observations in this study, the only possible conclusion is that it does not work this way. The following section of this review explains why in probably too much detail.

So, more specifically, the Be-10/NO3 relationship should be close to linear for relatively young soil ages, but as soil age increases enough that Be-10 decay is important, the slope of the relationship will change as the Be-10 inventory asymptotically approaches an equilibrium value where deposition is balanced by radioactive decay. Stated in math, this means that the relationship between Be-10 and NO3 inventories is given by parametric equations in $t$ (time, yr) for the NO3 inventory $I_{NO3}$ (mol/cm2):

$$I_{NO3} = Q_{NO3}t \tag{1}$$

where $Q_{NO3}$ is the depositional NO3 flux (mol/cm2/yr),

and for the Be-10 inventory $I_{10}$ (let's also use mol/cm2):

$$I_{10} = \frac{Q_{10}}{\lambda_{10}} \left[ 1 - exp(-\lambda_{10}t) \right] \tag{2}$$

where $Q_{10}$ is the depositional flux of Be-10 (mol/cm2/yr) and $\lambda_{10}$ is the decay constant for Be-10 (4.99 e-7 /yr).

So what just happened there was we went from two simple assumptions to a quantitative prediction for how measured inventories of Be-10 and NO3 should be related. There are some additional side predictions that will be important later. One is that the Be-10/NO3 ratio should be constant and equal to the deposition flux ratio for young soils, and will be lower than the depositional ratio for old soils because of radioactive decay. The Be-10/NO3 ratio can't be higher than the depositional ratio with these assumptions.

Continuing, the authors then make a third assumption, which is that Be-10 and NO3, once deposited, are transported together. If this is true, then not only the inventories, but also the concentrations, will be highly correlated. The slope of the relationship could vary in old soils, or in parts of the soil profile that have not exchanged Be-10 with the atmosphere for a while, because of Be-10 decay, but all three assumptions together predict a positive correlation between measured concentrations. This assertion is the basis for what the authors do next, which is to further assert that if they can establish a correlation between measured Be-10 and NO3 concentrations, they can then use this correlation to estimate Be-10 concentrations and therefore inventories in samples where only NO3 was measured. They go on and do this, and many of the apparent exposure ages that are eventually presented in the paper are from estimated Be-10 concentrations.

So far, the only problem with the paper is that the authors have not actually clearly stated the assumptions that led to their assertion that they can use NO3 concentrations to predict unmeasured Be-10 concentrations. However, I am now going to point out a lot of other problems.

Problem 1 is that assumptions 1 and 2 predict a specific quantitative relationship between Be-10 and NO3 inventories. Both inventories were measured at three sites. The following figure compares these inventories to the predicted relationship from the equations above.

[Figure]

The blue lines are the predicted relationships from the equations above. The different blue lines use different values for the NO3 deposition rate from the Graly and Adams papers. Of course these papers all use different units so I hope I got all the conversions to mol correct, although the order of magnitude variation indicates that I might have made a conversion error. Regardless, the curves show the shape of the relationship and hopefully at least the right order of magnitude.

The important thing here is that the observations look nothing like the predictions. One sample has lots of Be-10 but almost no NO3. Two samples have about the same amount of NO3 but wildly different amounts of Be-10. Not even any two of the observations can be fit to any one growth curve. These observations are impossible if assumptions 1 and 2 are true, which implies that, in fact, these assumptions are not true. Of course, there are some reasons that you could have both the assumptions and these results – you could have incomplete measurements of the inventories, or you could have very large and wildly different inherited concentrations of either or both...if you can have completely arbitrary inheritance you can have whatever you want. But no matter what, these results clearly show that in sharp contrast to our expectations and to the Graly study, observed Be-10 and NO3 inventories are NOT correlated between sites, no matter whether we think they ought to be or not.

Now consider Be-10 and NO3 concentrations. This plot shows NO3 and Be-10 concentrations in surface samples where both were measured.

[Figure]

These are also not correlated. Again, it could be possible to have these results and all the above assumptions if you can have inherited concentrations be whatever you want. But if the existing set of measurements of NO3 and Be-10 in surface samples are not correlated, then any theory that predicts that they are correlated can clearly not be used to predict Be-10 concentrations where they were not measured.

The point of all this is that the data clearly show that whatever we think ought to be happening, Be-10 and NO3 inventories are not correlated, and Be-10 and NO3 concentrations in surface samples are not correlated. Therefore, there is zero reason to believe that the authors' attempt to predict Be-10 concentrations or inventories in samples where they were not measured is correct. Of course it might be correct by accident, but this seems unlikely.

Note that the authors actually tried to do this in a more complicated way. They showed that within each of the three depth profiles where both Be-10 and NO3 were measured, they could be related by a power-law relationship. These relationships were different for all thres soils. Then they used these relationships to predict Be-10 concentrations at unmeasured sites simply by asserting which unmeasured soil was most like which measured soil.

This procedure includes several unjustified and unacceptable leaps of faith. First, there is no physical basis that would lead us to expect a power-law relationship. Thus, fitting a power law to the data makes no sense. Second, there is definitely no physical reasoning that the authors have presented that would predict both positive (as in two soils) and negative (as in one) slopes. In contrast, basic assumptions, as discussed above, predict a linear relationship in young soils and asymptotic in old soils, always with positive slopes. As far as I can tell, there is especially zero physical explanation for why there should be an inverse power law relationship. Thus, the power law fits just make no physical sense, and there is no reason to believe they have any physical significance. In fact, there are many reasons to believe that they don't have any physical significance. I conclude that they

don't have any physical significance, so predicted Be-10 concentrations based on these fits are also unlikely to be correct except by accident.

A further point is that if you just correlate the two things for each depth profile linearly without log-transforming, you get basically the same r-squared. So in addition to there being no apparent physical reason that I could discern for using a log-log fit, there is also not really any empirical reason to think that a log-log fit is any better than a linear fit.

Finally, consider the observed Be-10/NO3 ratios in the depth profiles. Those are shown in this figure.

[Figure]

The vertical gray lines are various guesses at the production ratio from the Graly and Adams results. The ratios from Roberts and Bennett are somewhere in the expected range of production ratios, which is more or less consistent with what we expect for dry soils that are closed systems for both things. The increase with depth, however, does seem to indicate that even in dry soils Be-10 and NO3 don't move together, NO3 is more easily moved down.

On the other hand, remember in the discussion above we showed that Be-10/NO3 ratios above the deposition ratio, which is the same as NO3/Be-10 ratios below the deposition ratio, are impossible in a closed system without inheritance. However, we see such impossible ratios at Thanksgiving Point. There are two obvious ways to explain this. One, nearly all the Be-10 and none of the NO3 could be inherited. This might be possible given the comparison with the in-situ Be-10 exposure ages. Two, water leaching of NO3 has taken place in the relatively warmer and wetter soil at Thanksgiving Point. Both of these possibilities invalidate pretty much all the assumptions used to argue for a correlation between NO3 and Be-10. If most of the Be-10 is inherited and different sites can have different inheritance, there is no expectation of a cross-site correlation between Be-10 and NO3. If water leaching of NO3 has taken place at some sites but not others, then, likewise, clearly it is not possible to apply any fixed relationship between NO3 and Be-10 concentrations

across different sites. Either one of these cases highlights that the authors' approach to predicting Be-10 concentrations at unmeasured locations does not have merit and is nearly certain to be incorrect.

The summary of this rather long discussion is that there is exactly zero observational support for the authors' scheme for predicting Be-10 concentrations from NO3 concentrations, and, in addition, zero theoretical support for some aspects of the scheme like the decision to use a power-law fit. In fact, comparison of theory to observations indicate that this should not work, except by accident. Even if the basic assumptions are correct, the expected correlation is not present due to some combination of background effects and NO3 leaching. Thus, the predicted Be-10 concentrations for sites where only NO3 was measured are incorrect. The authors must remove this aspect of the paper and consider only Be-10 and NO3 concentrations that were actually measured.

Correcting this problem will involve:

– entirely removing section 5.3.2.

– Removing the results of the calculations in 5.3.2. from all tables and figures, so that only measured data are shown throughout the paper. That includes removing Table 4.

— Removing and rethinking any discussion that relied on the estimated Be-10 concentrations.

– Removing the discussion of this subject in section 6.2. Basically, as detailed at length above, the statement that "...we conclude that NO3 appears suitable for relative age dating..." is not supported by the observations in this study. I agree that NO3 ought to be suitable for relative dating, but unfortunately the evidence in this paper rather indicates the opposite. There are many similar remarks in this section to the effect that "the correlation between Be-10 and NO3 is widely applicable in hyper-arid soils", etc. These might be true, but they are not proven by anything in this paper.

– Removing discussion of this subject from the abstract and conclusions.

This material could be replaced by a figure or two showing that there is no discernable correlation between NO3 and Be-10 inventories or concentrations, and a couple of sentences speculating on why.

A couple of final notes on this aspect of the paper here. First, I want to assure the authors that the paper will be just as good, in fact, better, without this element of the paper. I get the idea that the authors have an interesting data set that is difficult to easily interpret, so they are kind of struggling to make the paper have some elements that they perceive as more significant than just a set of empirical observations. However, the authors need to keep things in perspective here. These are a set of totally new observations from a part of Antarctica and a type of setting that no one has looked at from this perspective before. Most (possibly all) of the existing meteoric Be-10 data in Antarctica are from really old dry soils that have been ice-free for a really long time. The data in this paper are from much more complex sites with a complicated history of ice cover and exposure. If this paper presents the data, points out that they are much more complicated than we expect from

simple relationships found at more simple sites, and then stops, that is a big contribution. Instead de-emphasizing the complexities and simply asserting without cause that we know what is going on with these data doesn't make the paper better, it makes it worse. A clear and comprehensive observational study is extremely valuable. It is not necessary to try to explain everything, or to add speculative material, to increase the perceived interest of the study.

Second, if you had asked me before I read this paper whether there would be a strong correlation between Be-10 and NO3 in these samples, I definitely would have said yes. The results from the Graly paper about Mt. Achernar are extraordinarily clear in this regard, and I would have expected a similar situation here. The actual observations that show no correlation between sites are amazingly different from the Graly results. This clearly indicates that there is something that we are missing and the setting is much more complicated than we thought, and it is extremely clear from these results that the expected behaviour is wildly oversimplified. At present, this paper just asserts that the expected behaviour is true even if the observations don't agree with it. It would be much more valuable for this paper to highlight that apparently the expected behaviour is very oversimplified relative to reality and that we are missing important things, most likely having to do with variable inheritance and/or NO3 mobility in water.

Coming back to the overall paper, the rest of the discussion having to do with the apparent exposure ages is in pretty good shape. A few comments on the discussion:

The sentence in lines 336-339 doesn't contribute anything and should be removed. Start with "The Shackleton Glacier region..." It seems like the authors had some trouble getting started in this section...they should just simplify things by starting right into the observations they want to highlight.

Area around line 364. It seems to me the easiest explanation for the greater-than-LGM apparent meteoric Be-10 ages for the lower-elevation covered-at-LGM sites is just that the inherited Be-10 inventory is large compared to the relatively short exposure time at this site. Inherited Be-10 equivalent to 10,000-100,000-ish years exposure seems unsurprising.

Area around line 425. As discussed in the first review, the authors need to be more careful not to mix up evidence for aridity with evidence for ice sheet change or lack thereof. These are not at all the same thing.

Line 436. I don't understand why a shallow active layer implies that Be-10 was able to migrate deeper into the soil in the past. How do we know that it hasn't been the opposite – the active layer was shallower in the past and has been thickening over time?

Line 448. I don't understand this argument. You can get to an increasing NO3/Be-10 ratio lots of ways. As is evident in many otherwise dry soils in Antarctica, there is commonly a subsurface maximum in salt deposition just because of brief wetting by snow events moving salts below the surface and depositing them when the water sublimates. As NO3 has more pathways for mobility than Be-10, it seems much easier to explain this by enhanced NO3 transport instead of by some

complex inheritance effect.

5. Conclusions. Besides the need to remove discussion of estimating Be-10 from NO3 in lines 489-492, I only have one comment here. As noted above, I don't understand why the presence of soil ice requires a past warmer climate. Soil ice can easily form and remain under equilibrium conditions. Explain?

6. Other items:

As discussed in my request from the editor for a revised table, it was unnecessarily difficult to get all the data from the tables that were provided, because it was hard to connect which exact samples from which depths did or did not have Be-10 and/or NO3 measurements. The table that the authors provided in response to my request is much clearer. The authors should use that table instead of the existing Table 1, and leave the details of the Be-10 measurements in a separate table, which would be basically the current Table 3.

7. Finally. OK, that's it. As you can see from the fact that most of this review is devoted to explaining why the NO3/Be-10 correlation approach appears to be incorrect, I thought about this for some time and tried to convince myself that this would work. Again, based on first principles and the Graly paper, I would expect a strong relationship. However, it just isn't there. Of course this is interesting because it might show that NO3 seems to be behaving conservatively in some soils but not others, in which case these are not all dry soils and the local microclimate matters. But it makes the approach of predicting Be-10 from NO3 untenable. It's just not right and it needs to be removed before the paper is suitable for publication.

Otherwise, I want to make sure to come back to the main point that the authors have done a great job of revising this paper. It is enormously better than the first draft, which is admirable. Most authors do not display this level of commitment to revising a paper, and I appreciate this because I would like to see this data set published. But the fact is that the paper does still have this one problem that needs to be corrected.

---

## Author Response (AR2)

Dear Dr. Stroeven,

We are pleased to submit our revised manuscript, now titled "Relationship between meteoric $^{10}$Be and $NO_3^-$ concentrations in soils along the Shackleton Glacier, Antarctica." We have updated the title to reflect changes to the manuscript, which are aligned with critiques from Dr. Goehring and Referee #2. Dr. Goehring and Referee #2 have once again provided suggestions and comments which have greatly improved and streamlined our work. We are very grateful for their help. We have addressed specific comments in the pages below.

After considering the reviews and conversations amongst the authors, we have decided to shift the focus of this manuscript towards process. Specifically, we centered on the relationship of meteoric $^{10}$Be and $NO_3^-$ with depth. This change does not significantly change the writing itself, but instead mainly the order and manner in which we frame and present the data. The most substantial change was moving the $^{10}$Be – $NO_3^-$ dating and the $^{10}$Be inventory dating (formally measured and estimated ages, respectively) to the supplement. The inferred ages using a previously established relationship between maximum $^{10}$Be concentration and inventory are located at the end of the discussion (Section 6.2). While Referee #2's assumptions about our $^{10}$Be – $NO_3^-$ method are misguided due to our initial lack of detail in the text (see comments below) and the technique holds promise for future work, $^{10}$Be systematics in CTAM soils still needs further interrogation before exposure ages can be accurately determined.

We have three different soil profiles: Roberts Massif – a hyper-arid site with long exposure, Bennett Platform – a site recently uncovered by glacial retreat with large moraines, and Thanksgiving Valley – a site with a nearby active hydrologic system. We hypothesized that the relationship between the concentrations of $^{10}$Be and $NO_3^-$ for these soil profiles would be different, and they were. This was to be expected given our understanding of the mobility of $NO_3^-$ and $^{10}$Be in soils with different wetting and glacial histories. While this makes for an interesting comparison between the sites and informs landscape disturbance (either by glaciers or wetting), the fact that all the sites are different makes evaluating the technique difficult. For example, Dr. Goehring encouraged us to pursue a sensitivity analysis for both the $^{10}$Be – $NO_3^-$ regression and the $^{10}$Be inventory method. While this is relatively straightforward for evaluating the sensitivity of erosion in Eq. 4, the small number (3) of regressions makes the Monte Carlo statistical analyses questionable.

Our exposure duration estimations are comparable to cosmogenic exposure ages of respective nearby features in the Shackleton Glacier region (see supplementary figures S2 and S3), supporting this technique. However, with only three profiles that represent the complex soil environment of the region, we first need to describe the soil depositional environment and demonstrate that geochemical relationships exist before the ages can be verified.

The revision to our manuscript positions this work as a foundation to build upon for understanding landscape development, disturbance, and exposure age dating for Antarctica soils using meteoric $^{10}$Be and $NO_3^-$. Most ages terrestrial ages from Antarctica are ages of boulders and moraines, not soils. It appears that only one other study has measured salts and meteoric $^{10}$Be in soil from the Central Transantarctic Mountains (CTAM) (see Graham et al., 1997). Graham et al. conclude that meteoric $^{10}$Be systematics needs further study as an exposure proxy in the CTAM, but there has been no progress until our study. We show that meteoric $^{10}$Be inventories in the CTAM are similar to other hyper-arid soils in Antarctica, though interestingly the depth profiles themselves are variable. This is true for $NO_3^-$ as well. Since it is expected that $NO_3^-$ and $^{10}$Be would have similar concentration depth profile patterns in hyper-arid soils, deviations from this relationship help us understand if/when the soils were disturbed.

By focusing on the concentrations, patterns, and relationship of meteoric $^{10}$Be and $NO_3^-$, have taken the first step towards determining accurate soil exposure ages for the CTAM. Our work is not only critical for glaciologist and geomorphologists seeking to understand glacial advance and retreat as well as paleoclimate, but also for biologist searching for Antarctic refugia.

To summarize the contents in the following pages, we have made the suggested changes by both Dr. Goehring and Referee #2, which mainly entailed some reorganization and moving the ages to both the end of the discussion and the supplement. We hope to do a full sensitivity analysis of our exposure methods in a future study once we have collected sufficient measurements.

Best regards,

Melisa Diaz (on behalf of all authors)

Postdoctoral Scholar
Woods Hole Oceanographic Institution

**Brent Goehring (Referee)**

bgoehrin@tulane.edu

General Comments

> *There is still a lack of uncertainty analysis and quantification. This is a major issue in my mind. The authors presented results with and without erosion, and the differences are sometimes large, sometimes not; this screams for a full analysis of the sensitivity to erosion. You also rely on regressions to assess 10Be concentrations/inventories, any regression model has uncertainty and must be incorporated. The fact that uncertainties are discussed in other papers, means that those authors discuss the possible conceptual uncertainties in the application of the method. I was not referring to this, and rather referring to the resulting uncertainties on your results. I do agree that there is not a single number that results, but rather a distribution of values. I really do think this manuscript, and methodology, would benefit from this particularly since there is a new method, it relies on a regression, and all regressions by definition have some measure of uncertainty.*

We are very grateful to Dr. Goehring for explaining the need for the sensitivity analysis. We agree that our surface exposure age method and calculations appear sensitive to the erosion term. Additionally, we acknowledge that the uncertainty in our regressions needs to be considered. Given the comments from Reviewer 2 among other concerns, we have decided to move the ages calculated using the $^{10}$Be inventories and the ages using the $^{10}$Be – $NO_3^-$ regression to the supplementary materials, and have moved the inferred ages to the end of the discussion. With our three sediment profiles from Roberts Massif, Bennett Platform, and Thanksgiving Valley, we have demonstrated that there appears to be a relationship between $^{10}$Be and $NO_3^-$ with depth in hyper-arid TAM soils and this relationship can help inform wetting history, landscape disturbance/development, and possibly exposure age. These sites were selected because we hypothesized that they had different wetting and glacial histories, which is what our analyses ultimately showed to be true. However, since they are all different cross comparison is difficult and we ultimately would like to collect more and analyze more data for a future study specifically dedicated to interrogating our exposure age proxies. In this future study, we will certainly run Monte Carlo simulations for both the $^{10}$Be inventory and $^{10}$Be – $NO_3^-$ ages.

Detailed Comments

> *Line 20: Sentence starting this line seems incomplete. I feel it needs some sort of comparison to regions elsewhere. Also, the statement about largest changes in the TAM is not true for the entirety of Antarctica, but really only applicable to EAIS.*

We have corrected this sentence.

> *Line 25: How can you calculate a measured age? I think this sentences just needs to say very clearly what was done and remove the parenthetical aspects. Maybe rephrase as "We measured meteoric 10Be and NO3- concentrations to calculate exposure ages using the total 10Be inventory, the NO3 concentration, and infer exposure duration from the 10Be surface concentration."*

We have edited the abstract and removed this sentence since the $^{10}$Be ages have moved to the supplement.

> *Line 27: Swap lower and relatively*

This sentence has been removed.

> *Line 29: Change to indicate*

We have changed the tenses in the abstract.

> *Line 51: All evidence points to WAIS and EAIS max extent not synchronous with the*
> *canonical LGM (26-19 ka) and likely were largest approx. 14ka.*

We have made this correction.

> *Line 80: Delete "it"*

We have made this correction.

> *Line 81: You is spelled Yiou*

We checked the original publication and the author's name is indeed You.

> *Line 87: Insert "of" after "measurement"*

We have made this correction.

> *Line 116: Maybe instead of "measured" say "as measured". That being said, the*
> *explanation here is better than in the abstract. Can you try to clarify the abstract a bit*
> *more?*

We have moved the measured ages to the supplement.

> *Line 163: Many readers wont know what UVM stands for, suggest spelling out even*
> *though totally inconsequential for the manuscript.*

We have defined this acronym.

> *Line 182: replace the comma with and.*

We have made this correction.

> *Line 185: Expressing E as length per time in the text and then adding with respect to*
> *density is confusing, as the function really works with mass depth per time for erosion, E.*
> *Consider revising or rewording the text.*

We have reworded the text. It now reads, "The concentration of meteoric $^{10}$Be at the surface ($N$, atoms g$^{-1}$) per unit of time ($dt$) is expressed as a function, where the addition of $^{10}$Be is represented as the atmospheric flux to the surface ($Q$, atoms cm$^{-2}$ yr$^{-1}$), and removal is due to both radioactive decay, which is represented by a disintegration constant ($\lambda$, yr$^{-1}$), and erosion ($E$, cm yr$^{-1}$) (Eq. 1). Particle mobility into the soil column is represented by a diffusion constant ($D$, cm$^2$ yr$^{-1}$). The differential in depth is represented by $dz$."

*Line 285: Delete "regressed"*

We have deleted this.

*Line 292: Suggest presenting model ages in same order, first no erosion and then with erosion as above.*

We have taken this suggestion for the ages, now in the supplement. See Section S2 in the supplement.

*Line 295: Here and throughout there are some inconsistencies with tense and passive voice. Strongly suggest going through and editing for this and some other grammatical issues I noticed. I tried to point out many of them but skipped over many.*

We thank Dr. Goehring for bringing some of these errors to our attention and have corrected our grammar throughout.

*Line 377: Delete "study" before second clause.*

We have edited this section.

*Line 426: Should be stable*

We have made this correction.

*Line 438: Insert "of" after "most"*

We have made this correction.

*Line 460: MacKintosh is the same person as Mackintosh. The latter is correct. I noticed this elsewhere and suggest fixing before copy editing.*

We have made this correction.

*Figure 8: The black triangles are hard to see (even with the outline) on the imagery as the rock is so dark, and combined with ribbons of snow makes the two hard to differentiate. The symbols would also greatly benefit from displaying the ages on the maps.*

We have updated the symbol colors for Figure 8 (now Figure S2). We tried adding ages to the figure, but the figure quickly became difficult to interpret considering the wide range of ages from the literature. The ages are still included in the text and in Tables 3 and S3.

*Figure 9: Age-elevations plots are usually presented as elevation on the y-axis and age on the xaxis. I suggest flipping axes.*
*Figure 10: Swap axes like in Figure 9 for Figure 10b. For figure 10b, since you are comparing age vs elevation and showing along the length of Shackelton Glacier, where elevations span 1000 m, I suggest presenting as elevations relative to the ice surface. This will remove the slope effect on the absolute elevation.*

We have decided to keep elevation and distance from coast on the x-axes for what is now Figure 8.

*Table 3: Uncertainties here and for all other columns should be presented using the same exponential as the concentration, otherwise just adds confusion. I know this was mentioned in my first review, there would only be one leading zero for most of the samples. One other note is that PRIME Lab reports values at two decimal places, I suggest only reporting to this precision. There is unlikely a need to present non-background corrected ratios, they are so high that it is unlikely that many will be have any significant correction.*

We have changed the uncertainties to the same exponent as the concentration data and have moved the non-background corrected ratios to Table S2. Table 3 has now combined with Table 1 and includes the NO3 data.

**Anonymous Referee #2**

Second review, Diaz and others, Esurf

> *1. Introduction. This is enormously improved. It is basically fine now, although it would be helpful to better orient the reader (who perhaps has skipped over the abstract) by beginning the introduction with a sentence describing the actual research in the study, like "This study reports concentrations of atmospheric fallout constituents in soils in the southern TAM. These data can be used to understand soil age and disturbance frequency, which are biogeographically important because one of the most intriguing questions..."*

We thank Referee #2 for the suggestions on the introduction. We have rearranged the Introduction and Background sections to be more streamlined. The study goals are now in the second paragraph of the Introduction.

> *2. Background section. Also much improved. Great. Minor points:*
> *In lines 98-99, do you mean lower elevations, like water is running downhill for a significant distance, or greater depths in the soil? Clarify.*

We've edited this sentence on lines 115-116. It reads, "Once deposited on the surface, nitrate salts can be dissolved and transported down gradient or eluted to depth when wetted…"

> *In line 111. "Considerably fewer" sounds strange to the reader here because you are saying there are fewer studies in the CTAM than in NTAM/NVL, but you cite more studies for CTAM. I would just remove the "considerably fewer" and note that there have been scattered exposure-dating studies all over the Transantarctic Mountains.*

The sentence on lines 98-99 now reads, "There are scattered exposure age studies from across the CTAM using a variety of in-situ produced cosmogenic nuclides…"

> *3. Methods section. This is much better. Could use minor clarification in a couple of places, as follows:*
> *Line 151. Perhaps clearer to say "...to represent soils likely to have been covered during the LGM and exposed by more recent ice margin retreat."*

The sentence on line 157-158 now reads, "A second sample was collected closer to the glacier (between ~1,500 and 200 m from the first sample) to represent soils likely to have been covered during the LGM and exposed by more recent ice margin retreat."

> *Line 189. The usage of 'dz' is mathematically strange here. dz is just a generic differential in depth, it is not a parameter needed to evaluate the equation, so it is unclear what 'highly dependent on dz' means. I think what the authors are trying to say here is that the concentration gradient dn/dz depends on what D is and also varies with time, and D is unknown, so it is not really possible to calculate dn/dz. In addition, in line 187, the authors use 'concentration gradient' to describe d2n/dz2, which is confusing*

*because 'gradient' usually means the first derivative of something (dn/dz). The second derivative (d2n/dz2) would typically be described as 'curvature.' In any case, both of these points give the impression of carelessness and this section needs to be carefully checked to make sure it makes mathematical sense.*

We thank Referee #2 for identifying this confusing section. We have made modifications to clarify. See Section 4.3.

*Lines 211-12. This doesn't seem to make sense, because if Be-10 is supplied from the surface, the concentration has to decrease with depth at some point, no matter what. I believe the difference the authors are trying to point out is that in a normal soil one would expect a fairly smooth decrease, but in a perigacial soil one might expect a well-mixed active layer with constant concentration abruptly overlying a frozen layer with much lower concentration. However, this is not what this sentence says. In any case this sentence is oversimplified to the point of causing confusion, and it's not really very important, so I would remove it.*

We have modified lines 215 - 217 for clarity. The text now reads, "However, an accurate initial inventory can only be determined for soil profiles that are deep enough to capture background concentrations. This may not be the case in areas of permafrost where 10Be is restricted to the active layer."

*4. Results. The basic description of the results, up to section 3.2, is good. At this point, however, we get into the subject of the relationship between meteoric Be-10 and NO3 concentrations, which needs work.*
*Correlated inventories of Be-10 and NO3 that both increase monotonically with exposure age is, of course, what we expect, not only from first principles but due to the Graly study at Mt. Achernar, which showed these amazing correlations between exposure age and salt concentrations in sediments in a blue ice moraine. That is exactly how it is supposed to work and, like the authors, I want it to work that way in this study. Unfortunately it doesn't. I thought about this issue a lot in putting together this review – because, as noted, I would like this to work – but when you look at the actual observations in this study, the only possible conclusion is that it does not work this way.*

Before we address specific comments on the $^{10}$Be – NO$_3^-$ dating method, we want to emphasize that it was never our assumption that $^{10}$Be and NO$_3^-$ would have the same relationship across the Shackleton Glacier region. It is clear through satellite imagery that landscape development and evolution is variable for our sampling locations. As such, we intentionally selected three sites that we thought represented some of this variability. We apologize this was not clearly mentioned before. We have added the following sentences to Section 4.1, "We selected Roberts Massif, Bennett Platform, and Thanksgiving Valley as locations for the most in-depth analysis for the depth profiles. These locations were chosen to maximize variability in landscape development: Roberts Massif represented an older, likely minimally disturbed landscape; Thanksgiving Valley represented a landscape with possible hydrologic activity, as evidenced by

nearby ponds; Bennett Platform represented a landscape with evidence of recent glacial advance and retreat, and substantial topographic highs and lows (Table 2)."

> *The following section of this review explains why in probably too much detail.*
> *So, more specifically, the Be-10/NO3 relationship should be close to linear for relatively young soil ages, but as soil age increases enough that Be-10 decay is important, the slope of the relationship will change as the Be-10 inventory asymptotically approaches an equilibrium value where deposition is balanced by radioactive decay. Stated in math, this means that the relationship between Be-10 and NO3 inventories is given by parametric equations in t (time, yr) for the NO3 inventory*
> *So what just happened there was we went from two simple assumptions to a quantitative prediction for how measured inventories of Be-10 and NO3 should be related. There are some additional side predictions that will be important later. One is that the Be-10/NO3 ratio should be constant and equal to the deposition flux ratio for young soils, and will be lower than the depositional ratio for old soils because of radioactive decay. The Be-10/NO3 ratio can't be higher than the depositional ratio with these assumptions. Continuing, the authors then make a third assumption, which is that Be-10 and NO3, once deposited, are transported together. If this is true, then not only the inventories, but also the concentrations, will be highly correlated. The slope of the relationship could vary in old soils, or in parts of the soil profile that have not exchanged Be-10 with the atmosphere for a while, because of Be-10 decay, but all three assumptions together predict a positive correlation between measured concentrations.*
> *This assertion is the basis for what the authors do next, which is to further assert that if they can establish a correlation between measured Be-10 and NO3 concentrations, they can then use this correlation to estimate Be-10 concentrations and therefore inventories in samples where only NO3 was measured. They go on and do this, and many of the apparent exposure ages that are eventually presented in the paper are from estimated Be-10 concentrations.*
> *So far, the only problem with the paper is that the authors have not actually clearly stated the assumptions that led to their assertion that they can use NO3 concentrations to predict unmeasured Be-10 concentrations. However, I am now going to point out a lot of other problems.*

As we stated earlier and throughout our response to Referee #2, we have moved the $^{10}$Be – NO$_3^-$ dating approach to the supplementary materials (see Section S2.2). We agree that we did not clearly state our assumptions before interpreting the relationship between $^{10}$Be and NO$_3^-$. With the focus shift in the narrative from dating towards process, we expand upon our assumptions in greater detail in Section 6.1.

Specifically, lines 304-315 read, "Given sustained hyper-arid conditions, minimal landscape disturbance, and negligible biologic activity, one can expect meteoric 10Be and NO3- to be correlated throughout a depth profile given the similar accumulation mechanism (Everett, 1971;

Graham et al., 1997). Further, their inventories (Eq. 2) should increase monotonically with exposure duration. Deviations from this expected relationship could be due to 1) soil wetting, either in the present or past, 2) deposition of sediment with different 10Be to NO3- ratios compared to the depositional environment, 3) changes in the flux of either 10Be or NO3- with time, and 4) additional loss of NO3- due to denitrification or volatilization. The latter two mechanisms are likely minor processes, however, NO3- deposition fluxes are known to be spatially variable (Jackson et al., 2016; Lyons et al., 1990). As described above, Roberts Massif, Bennett Platform, and Thanksgiving Valley were selected for further investigation as locations which may represent different depositional environments: hypothesized hyper-aridity, recent glacial activity with large moraines, and active hydrology, respectively. By comparing differences in the expected and observed relationship between 10Be and NO¬3-, we can infer the processes which have influenced their relationship."

> *Problem 1 is that assumptions 1 and 2 predict a specific quantitative relationship between Be-10 and NO3 inventories. Both inventories were measured at three sites. The following figure compares these inventories to the predicted relationship from the equations above.*
> *The point of all this is that the data clearly show that whatever we think ought to be happening, Be-10 and NO3 inventories are not correlated, and Be-10 and NO3 concentrations in surface samples are not correlated. Therefore, there is zero reason to believe that the authors' attempt to predict Be-10 concentrations or inventories in samples where they were not measured is correct. Of course it might be correct by accident, but this seems unlikely.*
> *Note that the authors actually tried to do this in a more complicated way. They showed that within each of the three depth profiles where both Be-10 and NO3 were measured, they could be related by a power-law relationship. These relationships were different for all three soils. Then they used these relationships to predict Be-10 concentrations at unmeasured sites simply by asserting which unmeasured soil was most like which measured soil.*

The next several paragraphs of the review suggest that Referee #2 assumed we believed that [10]Be and NO$_3^-$ would have the same relationship across the region. This was not our intention and we have clarified throughout the manuscript. In our baseline assumptions, we argue that [10]Be and NO$_3^-$ are deposited by atmospheric deposition at a fairly constant rate and at a fixed ratio. Further, assuming neutral soil pH and sustained hyperaridity, their concentrations within a soil profile will be similar since they are both conservative. Deviations in this expected relationship can help us better understand the history of the landscape.

Referee #2 argues that [10]Be and NO$_3^-$ are not correlated in the CTAM soils. While Referee #2 is correct that if you combine all measurements and plot them, there is not a clear relationship, Roberts Massif, Bennett Platform, and Thanksgiving Valley all demonstrate that there is indeed a correlation between [10]Be and NO$_3^-$, albeit a complicated one. In Fig. 6b, we show the

concentrations of $^{10}$Be and $NO_3^-$ in the depth profiles. For Roberts Massif, the pattern is the same: the concentration increases just below the surface and then starts to decrease again. The concentrations for Thanksgiving Valley are similar and do not vary significantly throughout the profile. On the other hand, the $^{10}$Be concentrations at Bennett Platform decrease with depth, while the $NO_3^-$ increase with depth. Despite each of these locations having different concentration – depth relationships and probably different inheritance, $^{10}$Be and $NO_3^-$ still have statistically significant (though we acknowledge the low number of data points) and clear positive or negative correlations. Considering each location depth profile separately was not to the "more complicated way" since we hypothesized and showed that $^{10}$Be and $NO_3^-$ varied depending on wetting history and inflation/deflation. Once again, we apologize for the confusion and that our assumptions were not clearly stated in the beginning.

> *The summary of this rather long discussion is that there is exactly zero observational support for the authors' scheme for predicting Be-10 concentrations from NO3 concentrations, and, in addition, zero theoretical support for some aspects of the scheme like the decision to use a power-law fit. In fact, comparison of theory to observations indicate that this should not work, except by accident. Even if the basic assumptions are correct, the expected correlation is not present due to some combination of background effects and NO3 leaching. Thus, the predicted Be-10 concentrations for sites where only NO3 was measured are incorrect. The authors must remove this aspect of the paper and consider only Be-10 and NO3 concentrations that were actually measured. Correcting this problem will involve:*
>
> *−entirely removing section 5.3.2.*

Removed from the main text and into the supplement.

> *-Removing the results of the calculations in 5.3.2. from all tables and figures, so that only measured data are shown throughout the paper. That includes removing Table 4.*

Removed from the main text and into the supplement.

> *—Removing and rethinking any discussion that relied on the estimated Be-10 concentrations.*

Removed from the main text and into the supplement. The discussion has been refocused.

> *−Removing the discussion of this subject in section 6.2.*

Removed from the main text and revised for the supplement.

> *−Removing discussion of this subject from the abstract and conclusions.*

Removed from the main text and into the supplement.

> *A couple of final notes on this aspect of the paper here. First, I want to assure the authors that the paper will be just as good, in fact, better, without this element of the paper. I get the idea that the authors have an interesting data set that is difficult to easily interpret, so they are kind of struggling to make the paper have some elements that they perceive as*

*more significant than just a set of empirical observations. However, the authors need to keep things in perspective here. These are a set of totally new observations from a part of Antarctica and a type of setting that no one has looked at from this perspective before. Most (possibly all) of the existing meteoric Be-10 data in Antarctica are from really old dry soils that have been ice-free for a really long time. The data in this paper are from much more complex sites with a complicated history of ice cover and exposure.*

*If this paper presents the data, points out that they are much more complicated than we expect from simple relationships found at more simple sites, and then stops, that is a big contribution. Instead de-emphasizing the complexities and simply asserting without cause that we know what is going on with these data doesn't make the paper better, it makes it worse. A clear and comprehensive observational study is extremely valuable. It is not necessary to try to explain everything, or to add speculative material, to increase the perceived interest of the study.*

We thank Referee #2 for acknowledging the value of our data and the importance of our interpretations. We are excited to continue studying the relationship between meteoric $^{10}$Be and salts for future CTAM studies.

*Second, if you had asked me before I read this paper whether there would be a strong correlation between Be-10 and NO3 in these samples, I definitely would have said yes. The results from the Graly paper about Mt. Achernar are extraordinarily clear in this regard, and I would have expected a similar situation here. The actual observations that show no correlation between sites are amazingly different from the Graly results. This clearly indicates that there is something that we are missing and the setting is much more complicated than we thought, and it is extremely clear from these results that the expected behaviour is wildly oversimplified. At present, this paper just asserts that the expected behaviour is true even if the observations don't agree with it. It would be much more valuable for this paper to highlight that apparently the expected behaviour is very oversimplified relative to reality and that we are missing important things, most likely having to do with variable inheritance and/or NO3 mobility in water.*

We have revised the text to emphasize the complexity of CTAM terrestrial environment. Much of this discussion is presented in Section 6.1.

*Coming back to the overall paper, the rest of the discussion having to do with the apparent exposure ages is in pretty good shape. A few comments on the discussion: The sentence in lines 336-339 doesn't contribute anything and should be removed. Start with "The Shackleton Glacier region..." It seems like the authors had some trouble getting started in this section...they should just simplify things by starting right into the observations they want to highlight.*

This sentence has been removed.

*Area around line 364. It seems to me the easiest explanation for the greater-than-LGM apparent meteoric Be-10 ages for the lower-elevation covered-at-LGM sites is just that the inherited Be-10 inventory is large compared to the relatively short exposure time at this site. Inherited Be-10 equivalent to 10,000-100,000-ish years exposure seems unsurprising.*

This have been corrected in the supplement on line 127 in Section S3.

*Area around line 425. As discussed in the first review, the authors need to be more careful not to mix up evidence for aridity with evidence for ice sheet change or lack thereof. These are not at all the same thing.*

We thank Referee #2 for this reminder. We were attempting to make two different statements here about the past climate and glacial history. We have ensured that our references are appropriate throughout.

*Line 436. I don't understand why a shallow active layer implies that Be-10 was able to migrate deeper into the soil in the past. How do we know that it hasn't been the opposite – the active layer was shallower in the past and has been thickening over time?*

We have updated this sentence on lines 333-334 to read, "This suggests that the active layer may have deepened and shallowed throughout time, and modern 10Be mobility is limited to the top ~20 cm for most of the Shackleton Glacier region."

*Line 448. I don't understand this argument. You can get to an increasing NO3/Be-10 ratio lots of ways. As is evident in many otherwise dry soils in Antarctica, there is commonly a subsurface maximum in salt deposition just because of brief wetting by snow events moving salts below the surface and depositing them when the water sublimates. As NO3 has more pathways for mobility than Be-10, it seems much easier to explain this by enhanced NO3 transport instead of by some complex inheritance effect.*

We have modified this portion in lines 345-356 to consider both processes.

*5. Conclusions. Besides the need to remove discussion of estimating Be-10 from NO3 in lines 489-492, I only have one comment here. As noted above, I don't understand why the presence of soil ice requires a past warmer climate. Soil ice can easily form and remain under equilibrium conditions. Explain?*

We agree this was confusing and have removed it from the conclusions.

*6. Other items:*
*As discussed in my request from the editor for a revised table, it was unnecessarily difficult to get all the data from the tables that were provided, because it was hard to connect which exact samples from which depths did or did not have Be-10 and/or NO3 measurements. The table that the authors provided in response to my request is much clearer. The authors should use that table instead of the existing Table 1, and leave the*

> *details of the Be-10 measurements in a separate table, which would be basically the current Table 3.*

We thank Referee #2 for this suggestion. Table 1 now includes both the $^{10}$Be and $NO_3^-$ data. The metadata for $^{10}$Be are now included in the supplement as Table S2.

---

## Author Response (AR3)

Dear Dr. Stroeven,

We are pleased to submit our revised manuscript, titled "Relationship between meteoric $^{10}$Be and $NO_3^-$ concentrations in soils along Shackleton Glacier, Antarctica." We are grateful for the time and effort you have taken to review our manuscript and provide suggestions to improve the writing.

In the attached documents, you will see our tracked changes for this revision and the line changes below. In brief, we have made all recommended changes and edited for spelling and grammar. We are excited to see this work published and thank you again for your time throughout the review process.

Best regards,

Melisa Diaz (on behalf of all authors)

Postdoctoral Scholar
Woods Hole Oceanographic Institution

**Line comments:**

Line 33 and throughout: We have taken care to review each usage of "suggest" and similar verbs to restrict the usage of these words to when they refer to the authors.

Line 50: We have made this correction.

Line 57: We have removed "distinct".

Lines 67-69: We have rephrased this sentence. It now reads, "Antarctica is believed to have maintained a persistent ice sheet since at least the Eocene epoch, where paleorecords indicate that the East and West Antarctic Ice Sheets (EAIS and WAIS, respectively) have waxed and waned since at least the Miocene"

Line 72: We have replaced "with a grounding line…" with "[defined by] a grounding line…" We have also replaced "rapid" with "[more] rapid".

Line 76: We have made this correction.

Line 126 and throughout: We have replaced meters (m) with meters above sea level (m.a.s.l.).

Line 133: We have added references on the stable versus dynamic EAIS debate.

**Tables and figures:**

Figure 1: We have updated this map to include an insert of the Transantarctic Mountains, centered on the Shackleton and Beardmore regions. All abbreviations are defined in the figure caption and the scale bar was updated.

Figure 3: All terms are now defined in the caption.

Figure 4: "i, ii, and iii" are now defined in the caption.

Figure 5: "serve" has been deleted.

Figure 8: The axes labels have been updated to reflect m.a.s.l.

All edits were made for the tables.